# Multi-View Oriented GPLVM: Expressiveness and Efficiency

**Zi Yang**[*]
School of Computing and Data Science
University of Hong Kong
ziyang2023@connect.hku.hk

**Ying Li**[*]
School of Computing and Data Science
University of Hong Kong
lynnli98@connect.hku.hk

**Zhidi Lin**[†]
School of Computing and Data Science
University of Hong Kong
linzhidi017@gmail.com

**Michael Minyi Zhang**[†]
School of Computing and Data Science
University of Hong Kong
mzhang18@hku.hk

**Pablo M. Olmos**
Department of Signal Theory and Communications
Universidad Carlos III de Madrid
pamartin@ing.uc3m.es

## Abstract

The multi-view Gaussian process latent variable model (MV-GPLVM) aims to learn a unified representation from multi-view data but is hindered by challenges such as limited kernel expressiveness and low computational efficiency. To overcome these issues, we first introduce a new duality between the spectral density and the kernel function. By modeling the spectral density with a bivariate Gaussian mixture, we then derive a generic and expressive kernel termed Next-Gen Spectral Mixture (NG-SM) for MV-GPLVMs. To address the inherent computational inefficiency of the NG-SM kernel, we design a new form of random Fourier feature approximation. Combined with a tailored reparameterization trick, this approximation enables scalable variational inference for both the model and the unified latent representations. Numerical evaluations across a diverse range of multi-view datasets demonstrate that our proposed method consistently outperforms state-of-the-art models in learning meaningful latent representations.

## 1 Introduction

Multi-view representation learning aims to construct a unified latent representation by integrating multiple modalities and aspects of the observed data [1, 2]. The learned representation captures inter-view correlations within observations. By sharing information between each view of the data, we can obtain a much richer latent representation of the data compared to modeling each view independently which is crucial for handling complex datasets [3–5]. For example, modeling video data which involves both visual frames and audio signals [6], or developing clinical diagnostic systems that incorporates the patients' various medical records [7].

The paradigm of multi-view learning first emerged in early works leveraging techniques such as canonical correlation analysis (CCA) [8] and its kernelized extensions [9, 10] to jointly model

---

[*]Equal contribution
[†]Corresponding author

39th Conference on Neural Information Processing Systems (NeurIPS 2025).

**Table 1:** An overview of relevant MV-LVMs, where we extend advanced GPLVMs to the multi-view case by prefixing them with MV. Additionally, $N$ represents the total number of observations, $M$ denotes the observation dimensions, $V$ refers to the number of views, $U$ refers to the number of inducing points, $H$ indicates the number of GP layer, and $L$ indicates the dimension of random features.

| Model | Scalable model | Highly expressive kernel | Probabilistic mapping | Bayesian inference of latent variables | Computational complexity | Reference |
|---|---|---|---|---|---|---|
| MVAE | ✓ | - | ✗ | ✓ | - | [11] |
| MM-VAE | ✓ | - | ✗ | ✓ | - | [12, 16] |
| MV-GPLVM | ✗ | ✗ | ✓ | ✗ | $\mathcal{O}(N^3V)$ | [22] |
| MV-GPLVM-SVI | ✓ | ✗ | ✓ | ✓ | $\mathcal{O}(MU^3V)$ | [23] |
| MV-RFLVM | ✗ | ✗ | ✓ | ✗ | $\mathcal{O}(NL^2V)$ | [24] |
| MV-DGPLVM | ✗ | ✗ | ✓ | ✓ | $\mathcal{O}(HNU^2V)$ | [14] |
| MV-ARFLVM | ✓ | ✗ | ✓ | ✓ | $\mathcal{O}(NL^2V)$ | [21] |
| NG-MVLVM | ✓ | ✓ | ✓ | ✓ | $\mathcal{O}(NL^2V)$ | **This work** |

correlated data. However, these methods are limited in their ability to capture latent representations in complex datasets. To address this limitation, two more advanced approaches have become standard in multi-view learning: neural network-based methods, exemplified by multi-view variational autoencoders (MV-VAEs) [11, 12], and Gaussian process-based methods, represented by multi-view Gaussian process latent variable models (MV-GPLVMs) [13, 14].

MV-VAEs incorporate various view-specific variational autoencoders (VAEs) to address multi-view representation learning [11, 12, 15–17]. However, they suffer from posterior collapse—an inherent issue in VAEs [18]—where the encoder collapses to the prior distribution over the latent variables, thereby failing to capture the underlying structure of the data. Several hypotheses have been proposed to explain posterior collapse. A well-known contributing factor is the overfitting of the decoder network [19, 20]. One promising direction to mitigate this issue is to introduce regularization in the function space of the decoder, which may help the latent variable model learn more informative representations.

The implicit regularization imposed by the Gaussian process (GP) prior helps prevent MV-GPLVMs from severe overfitting [21]. However, MV-GPLVMs often lack the kernel flexibility required to model complex representations [13], and are computationally intensive to train, especially on large-scale multi-view datasets [14]. To overcome these challenges, we propose an expressive and efficient multi-view oriented GPLVM. Our contributions are:

- We establish a novel duality between the spectral density and the kernel function, deriving the expressive and generic Next-Gen Spectral Mixture (NG-SM) kernel, which, by modeling spectral density as dense Gaussian mixtures, can approximate any continuous kernel with arbitrary precision given enough mixture components. Building on this, we design a novel MV-GPLVM for multi-view scenarios, capable of capturing the unique characteristics of each view, leading to an informative unified latent representation.

- To enhance the computational efficiency, we design a unique unbiased random Fourier features (RFF) approximation for the NG-SM kernel that is differential w.r.t. kernel hyperparameters. By integrating this RFF approximation with an efficient two-step reparameterization trick, we enable efficient and scalable learning of kernel hyperparameters and unified latent representations within the variational inference framework [15], making the proposed model well-suited for multi-view scenarios.

- We validate our model on a range of cross-domain multi-view datasets, including synthetic, image, text, and wireless communication data. The results show that our model consistently outperforms various state-of-the-art (SOTA) MV-VAEs, MV-GPLVMs, and multi-view extensions of SOTA GPLVMs in terms of generating informative unified latent representations.

## 2   Background

**Gaussian Processes.** Probabilistic models like the Gaussian process latent variable model (GPLVM) [25] introduces a regularization effect via a GP-distributed prior that helps prevent overfitting and

thus improves generalization from limited samples. A GP $f(\cdot)$ is defined as a real-valued stochastic process defined over the input set $\mathcal{X} \subseteq \mathbb{R}^D$, such that for any finite subset of inputs $\mathbf{X} = \{\mathbf{x}_n\}_{n=1}^N \subset \mathcal{X}$, the random variables $\mathbf{f} = \{f(\mathbf{x}_n)\}_{n=1}^N$ follow a joint Gaussian distribution [26]. A common prior choice for a GP-distributed function is:

$$\mathbf{f} \mid \mathbf{X} = \mathcal{N}(\mathbf{f} \mid \mathbf{0}, \mathbf{K}), \tag{1}$$

where $\mathbf{K}$ denotes the covariance/kernel matrix evaluated on the finite input $\mathbf{X}$ with the kernel function $k(\mathbf{x}_1, \mathbf{x}_2)$, i.e., $[\mathbf{K}]_{i,j} = k(\mathbf{x}_i, \mathbf{x}_j), i, j \in (1, ..., N)$.

Consequently, the GPLVM has laid the groundwork for several advancements in multi-view representation learning. One of the early works by Li et al. [13] straightforwardly assumes that each view in observations is a projection from a shared latent space using a GPLVM, referred to as multi-view GPLVM (MV-GPLVM).

**Multi-View Gaussian Process Latent Variable Model.** A MV-GPLVM assumes that the relationship between each view $v \in (1, ..., V)$ of observed data $\mathbf{Y}^v = [\mathbf{y}_{:,1}^v, \mathbf{y}_{:,2}^v, ..., \mathbf{y}_{:,M_v}^v] \in \mathbb{R}^{N \times M_v}$ and the shared/unified latent variables $\mathbf{X} = [\mathbf{x}_1, \mathbf{x}_2, ..., \mathbf{x}_N]^\top \in \mathbb{R}^{N \times D}$ is modeled by a GP. That is, in each dimension $m \in (1, ..., M_v)$ and view $v$, MV-GPLVM is defined as follows:

$$\mathbf{y}_{:,m}^v \mid f_m^v(\mathbf{X}) \sim \mathcal{N}(f_m^v(\mathbf{X}), \sigma_v^2 \mathbf{I}), \quad f_m^v(\mathbf{X}) \sim \mathcal{N}(\mathbf{0}, \mathbf{K}^v), \quad \mathbf{x}_n \sim \mathcal{N}(\mathbf{0}, \mathbf{I}), \tag{2}$$

where $\sigma_v^2$ represents the noise variance and $f_m^v(\mathbf{X}) = [f_m^v(\mathbf{x}_1) \ldots f_m^v(\mathbf{x}_N)]^\top \in \mathbb{R}^N$. The stationary radial basis function (RBF) is typically the 'default' choice of the kernel function. Due to the conjugacy between the Gaussian likelihood[3] and the GP, we can integrate out each $f_m^v(\cdot)$ and get the marginal likelihood formed as :

$$\mathbf{y}_{:,m}^v \sim \mathcal{N}(\mathbf{0}, \mathbf{K}^v + \sigma_v^2 \mathbf{I}). \tag{3}$$

Based on the marginal likelihood and the prior of $\mathbf{X}$, the *maximum a posteriori* (MAP) estimation of $\mathbf{X}$ can be obtained, with $\mathcal{O}(N^3)$ computational complexity due to the inversion of the kernel matrix. Eleftheriadis et al. [27] extend the MV-GPLVM with a discriminative prior over the shared latent space for adapted to classification tasks. Later, Sun et al. [14] incorporate deep GPs [28] into MV-GPLVM for model flexibility, named MV-DGPLVM. However, existing MV-GPLVMs still fall short in handling practical multi-view datasets, that are often large-scale and exhibit diverse patterns across views. This limitation arises from either (1) the high computational complexity of fitting a deep GP or (2) limited kernel expressiveness caused by the stationary assumption. Particularly, a stationary kernel may fail to model input-varying correlations [29], especially in domains like video analysis and clinical diagnosis that exhibit complex, time-varying dynamics.

To address those issues, we review the recent work of GPLVM [21, 23, 24], that we may extend to deal with the multi-view scenario (See detailed comparisons in Table 1). One potential solution is the *advised* RFLVM (named ARFLVM for short) [21]. This method integrates the stationary spectral mixture (SM) kernel to enhance kernel flexibility in conjunction with a scalable random Fourier feature (RFF) kernel approximation. However, it remains limited by the stationary assumption.

**Random Fourier Features.** Bochner's theorem [30] states that any continuous stationary kernel and its spectral density $p(\mathbf{w})$ are Fourier duals, i.e., $k(\mathbf{x}_1 - \mathbf{x}_2) = \int p(\mathbf{w}) \exp(i\mathbf{w}^\top(\mathbf{x}_1 - \mathbf{x}_2)) \, d\mathbf{w}$. Built upon this duality, Rahimi and Recht [31] approximate the stationary kernel $k(\mathbf{x}_1 - \mathbf{x}_2)$ using an unbiased Monte Carlo (MC) estimator with $L/2$ spectral points $\{\mathbf{w}^{(l)}\}_{l=1}^{L/2}$ sampled from $p(\mathbf{w})$, given by

$$k(\mathbf{x}_1 - \mathbf{x}_2) \approx \phi(\mathbf{x}_1)^\top \phi(\mathbf{x}_2), \tag{4}$$

where the random feature

$$\phi(\mathbf{x}) = \sqrt{\frac{2}{L}} \left[ \cos\left(2\pi \mathbf{x}^\top \mathbf{w}^{1:L/2}\right), \sin\left(2\pi \mathbf{x}^\top \mathbf{w}^{1:L/2}\right) \right]^\top \in \mathbb{R}^L.$$

Here the superscript $1 : L/2$ indicates that the cosine or sine function is repeated $L/2$ times, with each element corresponding to the one entry of $\{\mathbf{w}^{(l)}\}_{l=1}^{L/2}$. Consequently, the kernel matrix $\mathbf{K}$ can be approximated by $\Phi(\mathbf{x})\Phi(\mathbf{x})^\top$, with $\Phi(\mathbf{x}) = [\phi(\mathbf{x}_1); \ldots; \phi(\mathbf{x}_N)]^\top$. By employing the RFF approximation, the kernel matrix inversion can be computed using the Woodbury matrix identity [32], thereby reducing the computational complexity to $\mathcal{O}(NL^2)$.

---

[3]Extending to other likelihoods is straightforward, as guided by the GP literature [23, 24].

# 3 Methodology

In § 3.1, we introduce our expressive kernel function for multi-view representation learning. In § 3.2, we propose a sampling-based variational inference algorithm to efficiently learn both kernel hyperparameters and latent representations. The scalable RFF approximation and the associated efficient reparameterization trick are detailed in § 3.3.

## 3.1 Next-Gen SM (NG-SM) Kernel

As mentioned in § 2, limited kernel expressiveness in the MV-GPLVM may hinder its ability to capture informative latent representations, potentially neglecting crucial view-specific characteristics like time-varying correlations in the data[4]. This problem necessitates a more flexible kernel design. The bivariate SM kernel (BSM) is one notable development for improving kernel flexibility [29, 33, 34]. Grounded in the duality that relates any continuous kernel function to a dual density from the generalized Bochner's theorem [35], this method first models the dual density using a mixture of eight bivariate Gaussian components and then transforms it into the BSM kernel. By removing the restrictions of stationarity, the generalized Bochner's theorem enables the BSM kernel to capture time-varying correlations [29].

However, the BSM kernel faces several limitations imposed by the generalized Bochner's theorem: (1) To guarantee the positive semi-definite (PSD) spectral density, the two variables in the bivariate Gaussian must have equal variances, which limits the flexibility of the Gaussian mixture and consequently reduces the kernel's expressiveness. (2) According to the duality in the generalized Bochner's theorem, it is not possible to derive a closed form expression of random features[5], such as $\phi(\mathbf{x})$ in Eq. (4). Thus, the inversion of the BSM kernel matrix retains high computational complexity, rendering it unsuitable for multi-view datasets. To address the limitations of the BSM kernel, we propose the new NG-SM kernel—a more flexible and RFF-admissible alternative. Its formulation is enabled by a novel duality, stated in Theorem 1.

**Theorem 1** (Universal Bochner's Theorem). *A complex-valued bounded continuous kernel $k(\mathbf{x}_1, \mathbf{x}_2)$ on $\mathbb{R}^D$ is the covariance function of a mean square continuous complex-valued random process on $\mathbb{R}^D$ if and only if*

$$k(\mathbf{x}_1, \mathbf{x}_2) = \frac{1}{4} \int \exp(i\mathbf{w}_1^\top \mathbf{x}_1 - i\mathbf{w}_2^\top \mathbf{x}_2) + \exp(i\mathbf{w}_2^\top \mathbf{x}_1 - i\mathbf{w}_1^\top \mathbf{x}_2) +$$
$$\exp(i\mathbf{w}_1^\top \mathbf{x}_1 - i\mathbf{w}_1^\top \mathbf{x}_2) + \exp(i\mathbf{w}_2^\top \mathbf{x}_1 - i\mathbf{w}_2^\top \mathbf{x}_2) u(\mathrm{d}\mathbf{w}_1, \mathrm{d}\mathbf{w}_2)$$

*where $u$ is the Lebesgue-Stieltjes measure associated with some function $p(\mathbf{w}_1, \mathbf{w}_2)$. When $\mathbf{w}_1 = \mathbf{w}_2$, this theorem reduces to Bochner's theorem.*

*Proof.* The proof can be found in App. B.2. $\qquad\square$

The duality established in Theorem 1 implies that a bivariate spectral density entirely determines the properties of a continuous kernel function. In this sense, we propose the underlying bivariate spectral density by a Gaussian mixture:

$$p_{\mathrm{ngsm}}(\mathbf{w}_1, \mathbf{w}_2) = \sum_{q=1}^{Q} \alpha_q s_q(\mathbf{w}_1, \mathbf{w}_2) \tag{5}$$

with each symmetric density $s_q(\mathbf{w}_1, \mathbf{w}_2)$

$$s_q(\mathbf{w}_1, \mathbf{w}_2) = \frac{1}{2} \mathcal{N}\left(\begin{pmatrix}\mathbf{w}_1 \\ \mathbf{w}_2\end{pmatrix} \middle| \begin{pmatrix}\boldsymbol{\mu}_{q1} \\ \boldsymbol{\mu}_{q2}\end{pmatrix}, \begin{bmatrix}\boldsymbol{\Sigma}_1 & \boldsymbol{\Sigma}_{\mathrm{c}}^\top \\ \boldsymbol{\Sigma}_{\mathrm{c}} & \boldsymbol{\Sigma}_2\end{bmatrix}\right) + \frac{1}{2} \mathcal{N}\left(\begin{pmatrix}-\mathbf{w}_1 \\ -\mathbf{w}_2\end{pmatrix} \middle| \begin{pmatrix}\boldsymbol{\mu}_{q1} \\ \boldsymbol{\mu}_{q2}\end{pmatrix}, \begin{bmatrix}\boldsymbol{\Sigma}_1 & \boldsymbol{\Sigma}_{\mathrm{c}}^\top \\ \boldsymbol{\Sigma}_{\mathrm{c}} & \boldsymbol{\Sigma}_2\end{bmatrix}\right) \tag{6}$$

in order to explore the space of continuous kernels. To simplify the notation, we omit the index $q$ from the sub-matrices $\boldsymbol{\Sigma}_1 = \mathrm{diag}(\boldsymbol{\sigma}_{q1}^2)$, $\boldsymbol{\Sigma}_2 = \mathrm{diag}(\boldsymbol{\sigma}_{q2}^2)$, and $\boldsymbol{\Sigma}_{\mathrm{c}} = \rho_q \mathrm{diag}(\boldsymbol{\sigma}_{q1}) \mathrm{diag}(\boldsymbol{\sigma}_{q2})$, where $\boldsymbol{\sigma}_{q1}^2, \boldsymbol{\sigma}_{q2}^2 \in \mathbb{R}^D$ and the scalar $\rho_q$ denotes the correlation between $\mathbf{w}_1$ and $\mathbf{w}_2$. These components

---

[4]For an exploration of the impact of limited kernel expressiveness in manifold learning, see § 4.1.

[5]See App. B.1 for a more detailed discussion.

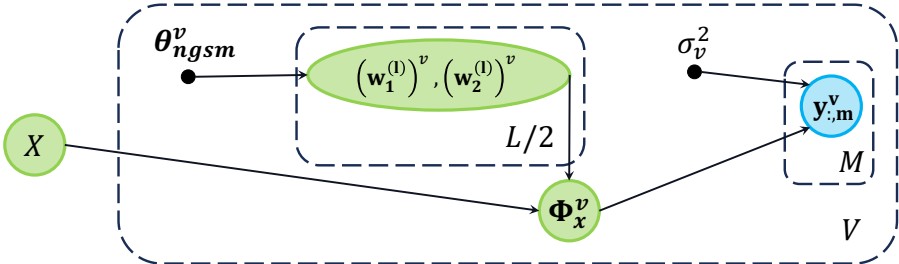

**Figure 1:** Probabilistic graphical model of our proposed method. Here, $\boldsymbol{\theta}_{ngsm}^{v}$ denotes the parameters of $p_{\text{ngsm}}$.

collectively form the covariance matrix of the $q$-th bivariate Gaussian component. Furthermore, the vectors $\boldsymbol{\mu}_{q1}$ and $\boldsymbol{\mu}_{q2} \in \mathbb{R}^D$ constitute the mean of the $q$-th bivariate Gaussian component.

Based on Theorem 1, we derive the NG-SM kernel, $k_{\text{ngsm}}(\mathbf{x}_1, \mathbf{x}_2)$ (see Eq. (40) in App. B.3), with kernel hyperparameters

$$\boldsymbol{\theta}_{\text{ngsm}} = \{\alpha_q, \boldsymbol{\mu}_{q1}, \boldsymbol{\mu}_{q2}, \boldsymbol{\sigma}_{q1}^2, \boldsymbol{\sigma}_{q2}^2, \rho_q\}_{q=1}^Q.$$

As the PSD assumption is relaxed and the mixtures of Gaussians become dense [36], the duality ensures that the NG-SM kernel can approximate any continuous kernel arbitrary well. Next, we will demonstrate that a general closed-form RFF approximation can be derived for all kernels based on our duality, which we specify for the NG-SM kernel below.

**Theorem 2.** *Let $\phi(\mathbf{x})$ be the randomized feature map of $k_{ngsm}(\mathbf{x}_1, \mathbf{x}_2)$, defined as follows:*

$$\sqrt{\frac{1}{2L}} \begin{bmatrix} \cos\left(\mathbf{w}_1^{(1:L/2)\top}\mathbf{x}\right) + \cos\left(\mathbf{w}_2^{(1:L/2)\top}\mathbf{x}\right) \\ \sin\left(\mathbf{w}_1^{(1:L/2)\top}\mathbf{x}\right) + \sin\left(\mathbf{w}_2^{(1:L/2)\top}\mathbf{x}\right) \end{bmatrix} \in \mathbb{R}^L, \tag{7}$$

*where the vertically stacked vectors $\{[\mathbf{w}_1^{(l)}; \mathbf{w}_2^{(l)}]\}_{l=1}^{L/2} \in \mathbb{R}^{L/2 \times 2D}$ are independent and identically distributed (i.i.d) random vectors drawn from the spectral density $p_{ngsm}(\mathbf{w}_1, \mathbf{w}_2)$. Then, the unbiased estimator of the kernel $k_{ngsm}(\mathbf{x}_1, \mathbf{x}_2)$ using RFFs is given by:*

$$k_{ngsm}(\mathbf{x}_1, \mathbf{x}_2) \approx \phi(\mathbf{x}_1)^\top \phi(\mathbf{x}_2). \tag{8}$$

*Proof.* The proof can be found in App. C.1. $\square$

By integrating this unbiased estimator into the framework of MV-GPLVM, we derive the next-gen multi-view GPLVM:

$$\mathbf{y}_{:,m}^v \sim \mathcal{N}(\mathbf{0}, \boldsymbol{\Phi}_x^v \boldsymbol{\Phi}_x^{v\top} + \sigma_v^2 \mathbf{I}), \quad (\mathbf{w}_1^{(l)})^v, (\mathbf{w}_2^{(l)})^v \sim p_{\text{ngsm}}^v(\mathbf{w}_1^v, \mathbf{w}_2^v), \quad \mathbf{x}_n \sim \mathcal{N}(\mathbf{0}, \mathbf{I}), \tag{9}$$

where the random feature matrix for each view is denoted as $\boldsymbol{\Phi}_x^v = [\phi(\mathbf{x}_1); \dots; \phi(\mathbf{x}_N)]^\top$, with the superscript of each feature map omitted for simplicity of notation. The probabilistic graphical model is shown in Figure 1. Furthermore, we collectively denote $\boldsymbol{\sigma}^2 = \{\sigma_v^2\}_{v=1}^V$ and spectral points $\mathbf{W} \triangleq \{\mathbf{W}^v\}_{v=1}^V$, with each

$$\mathbf{W}^v \triangleq \{[(\mathbf{w}_1^{(l)})^v; (\mathbf{w}_2^{(l)})^v]\}_{l=1}^{L/2} \in \mathbb{R}^{L/2 \times 2D}. \tag{10}$$

The following subsections will illustrate how to infer the parameters of the NG-MVLVM through a unified sampling-based variational inference framework [15].

## 3.2 Sampling-based Variational Inference

We employ variational inference [37] to jointly and efficiently estimate the posterior and hyperparameters $\boldsymbol{\theta} = \{\boldsymbol{\theta}_{\text{ngsm}}, \boldsymbol{\sigma}^2\}$. Variational inference reformulates Bayesian inference task as a deterministic optimization problem by approximating the true posterior $p(\mathbf{W}, \mathbf{X}|\mathbf{Y})$ using a surrogate distribution, $q(\mathbf{W}, \mathbf{X})$, indexed by variational parameters $\boldsymbol{\xi}$. The variational parameters are typically estimated by maximizing the evidence lower bound (ELBO) which is equivalent to minimizing the

Kullback-Leibler (KL) divergence $\text{KL}(q(\mathbf{W}, \mathbf{X}) \| p(\mathbf{W}, \mathbf{X}|\mathbf{Y}))$ between the surrogate and the true posterior.

To construct the ELBO for NG-MVLVM, we first obtain joint distribution of the model:

$$p(\mathbf{Y}, \mathbf{X}, \mathbf{W}) = p(\mathbf{X}) \prod_{v=1}^{V} p(\mathbf{W}^v) p(\mathbf{Y}^v | \mathbf{X}, \mathbf{W}^v) \tag{11}$$

and then define the variational distributions $q(\mathbf{W}, \mathbf{X})$ as

$$q(\mathbf{X}, \mathbf{W}) \triangleq q(\mathbf{X}) p(\mathbf{W}) = q(\mathbf{X}) \prod_{v=1}^{V} p(\mathbf{W}^v), \tag{12}$$

where $q(\mathbf{X}) = \prod_{n=1}^{N} \mathcal{N}(\mathbf{x}_n | \boldsymbol{\mu}_n, \mathbf{S}_n)$, and $\boldsymbol{\xi} \triangleq \{\boldsymbol{\mu}_n \in \mathbb{R}^D, \mathbf{S}_n \in \mathbb{R}^{D \times D}\}_{n=1}^{N}$ are the free variational parameters. The variational distribution $q(\mathbf{W})$ is set to its prior $p(\mathbf{W})$ as this is essentially equal to assuming that $q(\mathbf{W})$ follows a Gaussian mixtures (the justification for this choice of approximation is in App. C.3). Consequently, the optimization problem is:

$$\max_{\boldsymbol{\theta}, \boldsymbol{\xi}} \mathbb{E}_{q(\mathbf{X}, \mathbf{W})} \left[ \log \frac{p(\mathbf{X}) \prod_{v=1}^{V} p(\mathbf{W}^v) p(\mathbf{Y}^v | \mathbf{X}, \mathbf{W}^v)}{q(\mathbf{X}) \prod_{v=1}^{V} p(\mathbf{W}^v)} \right]$$

$$= \sum_{v=1}^{V} \underbrace{\mathbb{E}_{q(\cdot, \cdot)} [\log p(\mathbf{Y}^v | \mathbf{X}, \mathbf{W}^v)]}_{\text{(a): reconstruction}} - \underbrace{\text{KL}(q(\mathbf{X}) \| p(\mathbf{X}))}_{\text{(b): regularization}}$$

which jointly learns the model hyperparameters $\boldsymbol{\theta}$ and infers the variational parameters $\boldsymbol{\xi}$. Term (a) encourages reconstructing the observations using any samples drawn from $q(\mathbf{X}, \mathbf{W})$ and term (b) serves as a regularization to prevent $\mathbf{X}$ from deviating significantly from the prior distribution.

To address this optimization problem using the sampling-based variational inference [15], we first derive the analytical form of term (b), the KL divergence between two Gaussian distributions, and expand term (a), approximating the expectation via MC estimation,

$$\sum_{v=1}^{V} \sum_{m=1}^{M_v} \frac{1}{I} \sum_{i=1}^{I} \log \mathcal{N} \left( \mathbf{y}_{:,m}^v | \mathbf{0}, (\boldsymbol{\Phi}_x^v \boldsymbol{\Phi}_x^{v\top})^{(i)} + \sigma_v^2 \mathbf{I} \right), \tag{13}$$

where $I$ denotes the number of differentiable MC samples drawn from $q(\mathbf{X})$ and $p(\mathbf{W})$ with respect to $\boldsymbol{\theta}$ and $\boldsymbol{\xi}$ (see the further computational details in App. C.2). Then, modern optimization techniques, such as Adam [38], can be directly applied to solve the problem. However, this raises a question: *How can differentiable MC samples be efficiently generated from the mixture bivariate Gaussian distribution, $p(\mathbf{W}^v)$, that implicitly involves discrete variables?*

### 3.3 Sampling in Mixture Bivariate Gaussians

In other words, it is essential to both generate differentiable samples from the mixture bivariate Gaussian and ensure high sampling efficiency, which is particularly beneficial in the multi-view case. However, the typical sampling process hinders us from achieving this goal [39, 40]. A primary difficulty stems from first generating an index $q$ from the discrete distribution, $P(q) = \alpha_q / \sum_{j=1}^{Q} \alpha_j, q = 1, ..., Q$, as it is inherently non-reparameterizable w.r.t. the mixture weights. Although the Gumbel-Softmax method provides an approximation, it remains unstable and highly sensitive to hyperparameter choices [41]. Additionally, directly performing joint sampling from $s_q(\mathbf{w}_1, \mathbf{w}_2)$ (see Eq. (6)) incurs a computational complexity of $\mathcal{O}(8D^3)$, further complicating the sampling process. Next, we will demonstrate how to address both issues. To simplify the notation, we omit the superscript $v$ in this section.

### 1) Two-step reparameterization trick

Applying the reparameterization trick on a multivariate normal distribution requires computing the Cholesky decomposition of the full covariance matrix, incurring a computational cost of $\mathcal{O}(8D^3)$ [39]. To alleviate this computational burden, we propose the two-step reparameterization trick as follows, which reducing sampling complexity to $\mathcal{O}(D)$.

**Proposition 1** (Two-step reparameterization trick)**.** *We can sample* $\mathbf{w}_1^{(l)}, \mathbf{w}_2^{(l)}$ *from a bivariate Gaussian distribution* $s_q(\mathbf{w}_1, \mathbf{w}_2)$ *using the following steps:*

*1.* $\mathbf{w}_1^{(l)} = \boldsymbol{\mu}_{q1} + \boldsymbol{\sigma}_{q1} \circ \boldsymbol{\epsilon}_1,$

*2.* $\mathbf{w}_2^{(l)} = \boldsymbol{\mu}_{q2} + \rho_q(\boldsymbol{\sigma}_{q2} \backslash \boldsymbol{\sigma}_{q1}) \circ (\mathbf{w}_1^{(l)} - \boldsymbol{\mu}_{q1}) + \sqrt{1 - \rho_q^2} \boldsymbol{\sigma}_{q2} \circ \boldsymbol{\epsilon}_2,$

*where* $\boldsymbol{\epsilon}_2, \boldsymbol{\epsilon}_1$ *are standard normal random variables and* $\circ$ *and* $\backslash$ *represents element-wise multiplication and division, respectively.*

*Proof.* The proof and complexity analysis can be found in App. C.4. $\qquad\square$

## 2) Unbiased differential RFF estimator

Let spectral points $\mathbf{W} \triangleq \{[\mathbf{w}_{q1}^{(l)}; \mathbf{w}_{q2}^{(l)}]\}_{q=1, l=1}^{Q, L/2}$, be sampled from $p(\mathbf{W}) = \prod_{q=1}^{Q} \prod_{l=1}^{L/2} s_q(\mathbf{w}_1, \mathbf{w}_2)$ using the two-step reparameterization trick. Inspired by previous work [21, 42], we first use the spectral points from the $q$-th mixture component to construct the following feature map:

$$\varphi_q(\mathbf{x}) \triangleq \sqrt{\alpha_q} \cdot \phi(\mathbf{x}; \{\mathbf{w}_{q1}^{(l)}\}_{l=1}^{L/2}, \{\mathbf{w}_{q2}^{(l)}\}_{l=1}^{L/2}). \tag{14}$$

Based on the feature maps $\{\varphi_q(\mathbf{x})\}_{q=1}^{Q}$, we can formulate our RFF estimator for the NG-SM kernel as follows.

**Theorem 3.** *Stacking the feature maps* $\varphi_q(\mathbf{x}), q = 1, \ldots, Q$, *yields the final form of the RFF approximation for the NG-SM kernel,* $\varphi(\mathbf{x})$:

$$\varphi(\mathbf{x}) = [\varphi_1(\mathbf{x})^\top, \varphi_2(\mathbf{x})^\top, \ldots, \varphi_Q(\mathbf{x})^\top]^\top \in \mathbb{R}^{QL}. \tag{15}$$

*Built upon this mapping, our unbiased RFF estimator for the NG-SM kernel is reformulated as follows:*

$$\mathbb{E}_{p(\mathbf{W})}[\varphi(\mathbf{x}_1)^\top \varphi(\mathbf{x}_2)] = k_{\text{ngsm}}(\mathbf{x}_1, \mathbf{x}_2). \tag{16}$$

*Proof.* The proof can be found in App. C.5. $\qquad\square$

Moreover, given inputs $\mathbf{X}$ and the RFF feature map $\varphi(\mathbf{x})$, we can establish the approximation error bound for the NG-SM kernel gram matrix approximation, $\hat{\mathbf{K}}_{\text{ngsm}} \triangleq \Phi_{\text{ngsm}}(\mathbf{X})\Phi_{\text{ngsm}}(\mathbf{X})^\top$ below, where $\Phi_{\text{ngsm}}(\mathbf{X}) = [\varphi(\mathbf{x}_1), \ldots, \varphi(\mathbf{x}_N)]^\top \in \mathbb{R}^{N \times QL}$.

**Theorem 4.** *Let* $C = (\sum_{q=1}^{Q} \alpha_q^2)^{1/2}$, *then for a small* $\epsilon > 0$, *the approximation error between the true NG-SM kernel matrix* $\mathbf{K}_{\text{ngsm}}$ *and its RFF approximation* $\hat{\mathbf{K}}_{\text{ngsm}}$ *is bounded as follows:*

$$P\left(\left\|\hat{\mathbf{K}}_{\text{ngsm}} - \mathbf{K}_{\text{ngsm}}\right\|_2 \geq \epsilon\right) \leq N \exp\left(\frac{-3\epsilon^2 L}{2NC\left(6\left\|\mathbf{K}_{\text{ngsm}}\right\|_2 + 3NC\sqrt{Q} + 8\epsilon\right)}\right),$$

*where* $\|\cdot\|_2$ *is the matrix spectral norm.*

*Proof.* The proof can be found in App. C.6. $\qquad\square$

The feature map $\varphi(\mathbf{x})$ not only provides theoretical guarantees for the approximation but also eliminates the need to generate differential samples from the mixture bivariate Gaussian distribution by directly introducing differentiability into the optimization objective w.r.t. the mixture weights $\alpha_q$.

Furthermore, the two-step reparameterization trick uses the correlations between $\mathbf{w}_1$ and $\mathbf{w}_2$, for efficient sampling. Consequently, the aforementioned optimization problem is solvable with standard algorithms like Adam [15]. We summarize the pseudocode in Algorithm 1 which illustrates the implementation of our proposed method, named Next-Gen Multi-View Latent Variable Model (NG-MVLVM).

**Algorithm 1:** NG-MVLVM: Next-Gen Multi-View Latent Variable Model

---

1 **Input:** Dataset $\mathbf{Y}$; Maximum iterations $T$.
2 **Initialize:** Iteration count $t = 0$, model hyperparameters $\boldsymbol{\theta}$, and variational parameters $\boldsymbol{\xi}$.
3 **while** $t < T$ *or Not Converged* **do**
4     Sample $\mathbf{X}$ from $q(\mathbf{X}) = \prod_{n=1}^{N} \mathcal{N}(\mathbf{x}_n | \boldsymbol{\mu}_n, \boldsymbol{S}_n)$ using the reparameterization trick.
5     For each view, sample $\mathbf{W}^v$ from $p(\mathbf{W}^v) = \prod_{q=1}^{Q} \prod_{l=1}^{L/2} s_q(\mathbf{w}_1^v, \mathbf{w}_2^v)$ via the two-step
      reparameterization trick.
6     For each view, construct $\boldsymbol{\Phi}_{\text{ngsm}}^v(\mathbf{X})$ using the sampled $\mathbf{X}$ and $\mathbf{W}^v$.
7     Evaluate data reconstruction term of ELBO through Eq. (13).
8     Evaluate regularized term of ELBO analytically.
9     Maximize ELBO and update $\boldsymbol{\theta}, \boldsymbol{\xi}$ using Adam [38].
10     Increment $t = t + 1$.
11 **Output:** $\boldsymbol{\theta}, \boldsymbol{\xi}$.

---

## 4 Experiments

We demonstrate the superior performance of our model[6] across multiple cross-domain multi-view datasets, including synthetic data (§ 4.1), image-text data (§ 4.2), and wireless communication data (§ 4.3). Additional experimental details are provided in App. D, including benchmark implementations (App. D.2) and hyperparameter selection (App. D.3). We further evaluate the representation learning capability on single-view real-world datasets (App. D.4.1), and demonstrate robustness in multi-view settings (App. D.4) and data reconstruction tasks (App. D.5), through supplementary simulations.

### 4.1 Synthetic Data

We firstly demonstrate the impact of kernel expressiveness on manifold learning and highlight the expressive power of the proposed NG-SM kernel. To this end, we synthesized two datasets using a two-view MV-GPLVM with the $S$-shape $\mathbf{X}$, based on different kernel configurations: (1) both views using the stationary RBF kernel, and (2) one view using the RBF kernel and the other using the non-stationary Gibbs kernel [26] (see further details in App. D.1.1). For benchmark methods, we use the MV-DGPLVM [14] and the multi-view extension of the SOTA ARFLVM [21], namely, MV-ARFLVM.

The manifold learning and kernel learning results across different methods for the two datasets are presented in Figure 2. The results for MV-ARFLVM indicate that if the model fails to capture the non-stationary features of one view, then significant distortions arise in the unified latent variables. In turn this degrades the model's ability to learn stationary features from other views. In contrast, both the latent variables and kernel matrices learned by our method are closer to the ground truth compared to the benchmark methods, especially in non-stationary kernel setting. Thus, these results demonstrate the importance of kernel flexibility for the MV-GPLVM.

For the MV-DGPLVM, we select the best latent representations from all GP layers and plot the corresponding kernel Gram matrix. However, the kernel learning result may be less meaningful, as the flexibility of model stems from using deep GPs rather than the kernel choice. The deep architecture of MV-DGPLVM exhibits a notable capability to capture non-stationary components, yielding a relatively "high-quality" latent representations. However, due to the large number of parameters and the complex hierarchical structure [43], MV-DGPLVM often suffers from overfitting and computational inefficiency, still resulting in inferior performance compared to our method.

### 4.2 Multi-View Image and Text Data

We further demonstrate our model's ability to learn unified latent representations on various multi-view image and text datasets, while maintaining high computational efficiency. Specifically, following the setting of Wu and Goodman [11], we construct multi-view settings by pairing each single-view dataset—images (MNIST, YALE, CIFAR), text (NEWSGROUPS), and structured data

---

[6]Code is publicly available at `https://github.com/ziyang18/NG-MVLVM`.

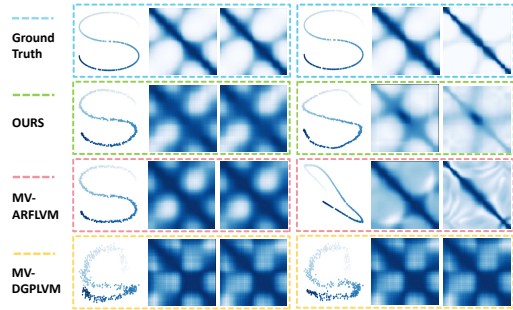 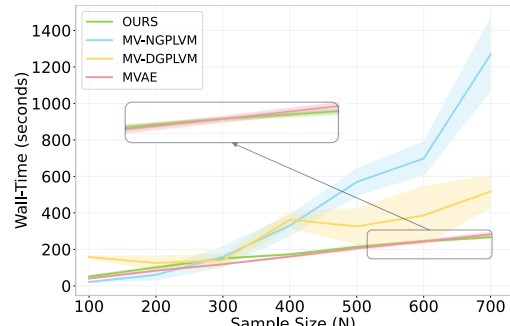

**Figure 2:** Comparison of learned latent variables and kernel matrices with the ground truth. Each dashed box shows latent variables (left) and kernel matrices for the two views (middle and right). **Left**: Both views use RBF. **Right**: View 1 uses RBF, view 2 uses Gibbs.

**Figure 3:** The wall-time (in seconds) for training on MNIST across different benchmark methods with varying dataset size $N$. Solid lines indicate the average over five runs; shaded regions represent the standard deviation.

**Table 2:** Classification accuracy, expressed as a percentage (%), is evaluated by using KNN and SVM classifiers with five-fold cross-validation. **Mean and standard deviation of the accuracy** is computed over five experiments and the top-performing result for each metric is in bold.

| DATASET | BRIDGES | | CIFAR | | MNIST | | NEWGROUPS | | YALE | |
|---|---|---|---|---|---|---|---|---|---|---|
| METRIC | KNN | SVM | KNN | SVM | KNN | SVM | KNN | SVM | KNN | SVM |
| **OURS** | **85.31 ± 1.47** | **87.49 ± 1.76** | **36.02 ± 0.34** | 32.47 ± 0.96 | **82.72 ± 0.61** | **60.14 ± 1.11** | **100.0 ± 0.00** | **100.0 ± 0.00** | **83.87 ± 1.44** | 64.81 ± 2.50 |
| MV-ARFLVM | 83.42 ± 1.53 | 86.21 ± 7.59 | 35.71 ± 0.59 | 31.58 ± 1.13 | 81.85 ± 1.58 | 59.51 ± 2.43 | **100.0 ± 0.00** | **100.0 ± 0.00** | 78.86 ± 2.23 | 56.72 ± 3.41 |
| MV-DGPLVM | 78.03 ± 11.96 | 73.48 ± 3.08 | 22.15 ± 10.54 | 24.31 ± 8.07 | 26.83 ± 12.25 | 27.04 ± 18.25 | 60.14 ± 2.31 | 36.02 ± 5.354 | 53.97 ± 6.37 | 34.58 ± 12.67 |
| MVAE | 73.38 ± 2.83 | 71.26 ± 3.22 | 25.83 ± 7.97 | 31.74 ± 2.73 | 46.41 ± 0.88 | 55.18 ± 1.91 | 33.81 ± 4.18 | 35.97 ± 3.25 | 74.92 ± 3.31 | **65.13 ± 2.34** |
| MMVAE | 73.01 ± 3.01 | 72.04 ± 2.43 | 29.81 ± 2.24 | **34.73 ± 1.64** | 39.07 ± 2.58 | 40.85 ± 7.90 | 95.21 ± 1.98 | 38.86 ± 2.70 | 41.35 ± 4.94 | 22.12 ± 5.97 |
| MV-NGPLVM | 80.72 ± 3.89 | 71.58 ± 3.59 | 29.14 ± 2.07 | 21.92 ± 1.72 | 52.74 ± 5.10 | 13.63 ± 1.96 | 99.25 ± 4.84 | 35.28 ± 1.41 | 30.37 ± 1.36 | 23.91 ± 4.09 |

(BRIDGES)[7]—with its corresponding label as an additional view. In addition to SOTA MV-GPLVM variants used in § 4.1, we also compare MV-GPLVM with the Gibbs kernel (MV-NGPLVM) and the SOTA MV-VAE variants: MVAE [11] and MMVAE [12]. After inferring the unified latent variable **X**, we perform five-fold cross-validation using two types of classifiers: K-nearest neighbor (KNN) and support vector machine (SVM). As summarized in Table 2, we report the mean and standard deviation of classification accuracy of various classifiers across multiple datasets. Additionally, Figure 3 illustrates the model fitting wall-time on the MNIST dataset with respect to dataset size.

From those results, we can see that our method demonstrates superior performance in estimating the latent represntations while maintaining computational scalability. For the MV-VAE variants, the inferior performance can be attributed to: (1) optimizing the huge number of neural network parameters leads to model instability, and (2) the posterior collapse issue in the VAE results in an uninformative latent space. Both factors are partly due to overfitting, a problem that can be naturally addressed by the MV-GPLVM variants.

Consequently, the performance of the MV-GPLVM variants is generally superior to that of the MV-VAE, though this often comes with increased computational cost—an issue our method effectively mitigates. Specifically, the performance improvement of our method compared to MV-ARFLVM and MV-NGPLVM is due to our proposed NG-SM kernel. MV-DGPLVM not only exhibits instability but also incurs extremely high computational costs, which can be attributed to its large number of parameters.

### 4.3 Wireless Communication Data

For further evaluations, we test the model in a channel compression task. Specifically, we generate wireless communication channel datasets using a high-fidelity channel simulator—QUAsi Deterministic RadIo channel GenerAtor (QUADRIGA)[8]. The environment we considered consists of a base station (BS) with 32 antennas serving 10 single-antenna user equipments (UEs), each moving at a speed of 30 km/h. We sample $1,000$ complex-valued channel vectors for each UE at intervals of 2.5

---

[7]See detailed dataset descriptions in App. D.1.2.

[8]https://quadriga-channel-model.de

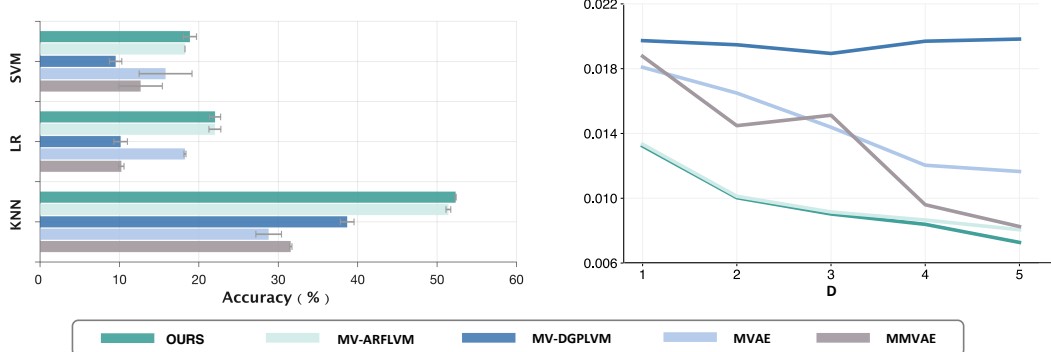

**Figure 4:** (**Left**) Classification accuracy , expressed as a percentage (%), is evaluated using KNN, Logistic Regression (LR) and SVM classifiers with five-fold cross-validation. Mean and standard deviation are computed over five experiments. (**Right**) The average mean squared error (MSE) of the reconstructed channel data compared to the original channel data across different latent variable dimensions in five experiments.

ms. The real and imaginary parts of each complex channel vector are treated as two distinct views, with the identifier of the UE as the label, resulting in a two-view dataset[9] with $N = 10,000$ and $M_v = 32$ for $v = 1, 2$.

To reduce communication overhead, the UE typically transmits a compressed channel vector to the BS instead of the full vector. This compressed vector must retain sufficient information to accurately reconstruct the ground-truth channel vectors. We use the unified latent variables as the compressed channel vector and evaluate both their classification performance and reconstruction capability. The mean and standard error of the classification accuracy, along with the mean square error (MSE) between the reconstructed and ground-truth channel vectors, are shown in Figure 4. The results indicate that the KNN accuracy achieved by our method consistently surpasses that of competing methods, while the MSE is consistently lower. These findings highlight the superior performance of our model in channel compression tasks compared to other benchmark approaches.

## 5 Conclusion

This paper introduces NG-MVLVM, a novel model designed to address two critical challenges in multi-view representation learning with MV-GPLVMs: limited kernel expressiveness and computational inefficiency. Specifically, we first establish a duality between spectral density and kernel function, yielding a versatile kernel capable of modeling the non-stationarity in multi-view datasets. We then show that with the proposed RFF approximation and efficient sampling method, the inference of model and latent representations can be effectively performed within a variational inference framework. Experimental validations demonstrate that our model, NG-MVLVM, outperforms state-of-the-art methods such as MV-VAE and MV-GPLVM in providing informative unified latent representations across diverse cross-domain multi-view datasets.

## Acknowledgments

The contribution of Michael Zhang was supported by the HKU Seed Fund for PI Research – Basic Research #2402101367.

Pablo M. Olmos was supported by the Comunidad de Madrid IND2024/TIC-34728, IDEA-CM project (TEC-2024/COM-89), the ELLIS Unit Madrid (European Laboratory for Learning and Intelligent Systems), the 2024 Leonardo Grant for Scientific Research and Cultural Creation from the BBVA Foundation, and by projects MICIU/AEI/10.13039/501100011033/FEDER and UE (PID2024-157856NB-I00 CARTESIAN, PID2021-123182OB-I00; EPiCENTER).

---

[9]See App. D.1.3 for more details regarding the wireless communication scenario settings.

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

# APPENDICES

# A  Notation Table

We summarize key symbols and their corresponding dimensions used throughout the paper to improve clarity.

**Table 3:** Summary of Notations

| Symbol | Description | Dimension |
|---|---|---|
| $N$ | Number of observations | scalar |
| $M$ | Dimensionality of observation | scalar |
| $V$ | Number of views | scalar |
| $D$ | Dimensionality of latent space | scalar |
| $U$ | Number of inducing points | scalar |
| $H$ | Number of GP layers (for DGPLVM) | scalar |
| $L$ | Dimensionality of random features | scalar |
| $X$ | Latent representation | $N \times D$ |
| $Y^{(v)}$ | Observation in view $v$ | $N \times M_v$ |
| $\mathbf{y}^v_{:,m}$ | Observation vector in view $v$, dimension $m$ | $N \times 1$ |
| $f^{(v)}_m(X)$ | Latent function for view $v$, dimension $m$ | $N \times 1$ |
| $K^{(v)}$ | Kernel matrix for view $v$ | $N \times N$ |
| $\mathbf{w}^{(v)}$ | Spectral points of view $v$ | $L/2 \times 2D$ |
| $(\mathbf{w}^{(l)}_1)^v$ | $l$-th sampled spectral point for view $v$ (first component) | $D$ |
| $\sigma^2_v$ | Noise variance for view $v$ | scalar |
| $\alpha_q$ | Mixture weight of $q$-th Gaussian component | scalar |
| $\mu_{q1}, \mu_{q2}$ | Mean vectors of $q$-th bivariate Gaussian | $D$ |
| $\sigma^2_{q1}, \sigma^2_{q2}$ | Variance vectors of $q$-th bivariate Gaussian | $D$ |
| $\rho_q$ | Correlation coefficient in $q$-th component | scalar |
| $\theta$ | Set of kernel hyperparameters | - |
| $\xi$ | Set of variational parameters | - |

# B  Next-Gen SM kernel and Universal Bochner's Theorem

## B.1  Bivariate SM kernel

The development of the spectral mixture (SM) kernel is based on the fundamental Bochner's theorem [44], which suggests that any stationary kernel and its spectral density are Fourier duals. However, the stationarity assumption limits the kernel's learning capacity, especially when dealing with multi-view datasets, where some views may exhibit non-stationary characteristics. To model the non-stationarity, the Bivariate SM (BSM) kernel was introduced in [29, 45], based on the following generalized Bochner's theorem:

**Theorem 5** (Generalized Bochner's Theorem). *A complex-valued bounded continuous kernel $k(\mathbf{x}_1, \mathbf{x}_2)$ on $\mathbb{R}^D$ is the covariance function of a mean square continuous complex-valued random process on $\mathbb{R}^D$ if and only if*

$$k(\mathbf{x}_1, \mathbf{x}_2) = \int \exp\big(i\big(\mathbf{w}_1^\top \mathbf{x}_1 - \mathbf{w}_2^\top \mathbf{x}_2\big)\big) m(\mathrm{d}\mathbf{w}_1, \mathrm{d}\mathbf{w}_2), \tag{17}$$

*where $m$ is the Lebesgue-Stieltjes measure associated with some positive semi-definite (PSD) function $S(\mathbf{w}_1, \mathbf{w}_2)$.*

According to Theorem 5, one can approximate the function $S(\mathbf{w}_1, \mathbf{w}_2)$, and implement the inverse Fourier transform shown in Eq. (17) to obtain the associated kernel function. The bivariate spectral mixture (BSM) kernel adopts this concept and approximates $S(\mathbf{w}_1, \mathbf{w}_2)$ as a bivariate Gaussian mixture as follows:

$$S(\mathbf{w}_1, \mathbf{w}_2) = \sum_{q=1}^{Q} \alpha_q s_q(\mathbf{w}_{q1}, \mathbf{w}_{q2}), \tag{18}$$

where $s_q\left(\mathbf{w}_{q1}, \mathbf{w}_{q2}\right)$ represents the $q$-th component of a bivariate Gaussian distribution, formally defined as [29]:

$$\frac{1}{8} \sum_{\boldsymbol{\mu}_q \in \pm\{\boldsymbol{\mu}_{q1}, \boldsymbol{\mu}_{q2}\}^2} \mathcal{N}\left[\begin{pmatrix}\mathbf{w}_{q1} \\ \mathbf{w}_{q2}\end{pmatrix}\bigg| \begin{pmatrix}\boldsymbol{\mu}_{q1} \\ \boldsymbol{\mu}_{q2}\end{pmatrix}, \underbrace{\begin{pmatrix}\operatorname{diag}(\boldsymbol{\sigma}_{q1}^2) & \rho_q \operatorname{diag}(\boldsymbol{\sigma}_{q1}) \operatorname{diag}(\boldsymbol{\sigma}_{q2}) \\ \rho_q \operatorname{diag}(\boldsymbol{\sigma}_{q1}) \operatorname{diag}(\boldsymbol{\sigma}_{q2}) & \operatorname{diag}(\boldsymbol{\sigma}_{q2}^2)\end{pmatrix}}_{\triangleq \boldsymbol{\Sigma}_q}\right].$$
(19)

Here $\pm\{\boldsymbol{\mu}_{q1}, \boldsymbol{\mu}_{q2}\}^2$ represents:

$$\{(\boldsymbol{\mu}_{q1}, \boldsymbol{\mu}_{q2}), (\boldsymbol{\mu}_{q1}, \boldsymbol{\mu}_{q1}), (\boldsymbol{\mu}_{q2}, \boldsymbol{\mu}_{q2}), (\boldsymbol{\mu}_{q2}, \boldsymbol{\mu}_{q1}),$$
$$(-\boldsymbol{\mu}_{q1}, -\boldsymbol{\mu}_{q2}), (-\boldsymbol{\mu}_{q1}, -\boldsymbol{\mu}_{q1}), (-\boldsymbol{\mu}_{q2}, -\boldsymbol{\mu}_{q2}), (-\boldsymbol{\mu}_{q2}, -\boldsymbol{\mu}_{q1})\}.$$
(20)

The terms $\boldsymbol{\sigma}_{q1}^2$ and $\boldsymbol{\sigma}_{q2}^2 \in \mathbb{R}^D$ represent the variances of the $q$-th bivariate Gaussian component, while $\rho_q$ denotes the correlation between $\mathbf{w}_{q1}$ and $\mathbf{w}_{q2}$; the vectors $\boldsymbol{\mu}_{q1}$ and $\boldsymbol{\mu}_{q2} \in \mathbb{R}^D$ specify the means of the $q$-th Gaussian component. The mixture weight is denoted by $\alpha_q$, and $Q$ is the total number of mixture components. Taking the inverse Fourier transform, we can obtain the following kernel function:

$$k\left(\mathbf{x}_1, \mathbf{x}_2\right) = \sum_{q=1}^{Q} \alpha_q \exp\left(-\frac{1}{2} \tilde{\boldsymbol{x}}^\top \boldsymbol{\Sigma}_q \tilde{\boldsymbol{x}}\right) \Psi_q(\mathbf{x}_1)^\top \Psi_q\left(\mathbf{x}_2\right),$$
(21)

where

$$\Psi_q(\mathbf{x}) = \begin{bmatrix}\cos\left(\boldsymbol{\mu}_{q1}^\top \mathbf{x}\right) + \cos\left(\boldsymbol{\mu}_{q2}^\top \mathbf{x}\right) \\ \sin\left(\boldsymbol{\mu}_{q1}^\top \mathbf{x}\right) + \sin\left(\boldsymbol{\mu}_{q2}^\top \mathbf{x}\right)\end{bmatrix} \in \mathbb{R}^2,$$

and $\tilde{\mathbf{x}} = (\mathbf{x}_1, -\mathbf{x}_2) \in \mathbb{R}^{2D}$ is a stacked vector.

The BSM kernel overcomes the stationarity limitation in the SM kernel. However, it still has two major issues.

- First, the assumption of identical variance of $\mathbf{w}_1$ and $\mathbf{w}_2$ limits the approximation flexibility of the mixture of Gaussian, which in turn diminishes the generalization capacity of the kernel.
- Second, the RFF kernel approximation technique cannot be directly applied, as the closed-form expression of the feature map is hard to derive (see explaination below).

The first limitation arises from the requirement that the spectral density must be PSD. To ensure the symmetry of the BSM kernel function (i.e., $k(\mathbf{x}_1, \mathbf{x}_2) = k(\mathbf{x}_2, \mathbf{x}_1)$), the BSM kernel assumes that $\boldsymbol{\sigma}_{q1} = \boldsymbol{\sigma}_{q2}, \forall q$.

In addition, we highlight the challenges of deriving a closed-form feature map for RFF when directly applying Theorem 5. According to Theorem 5, the kernel can be approximated via MC as follows:

$$\begin{aligned}
k\left(\mathbf{x}_1, \mathbf{x}_2\right) &= \frac{1}{4} \int \exp\left(i(\mathbf{w}_1^\top \mathbf{x}_1 - \mathbf{w}_2^\top \mathbf{x}_2)\right) u(\mathrm{d}\mathbf{w}_1, \mathrm{d}\mathbf{w}_2) \\
&= \frac{1}{4} \mathbb{E}_u\left(\exp\left(i(\mathbf{w}_1^\top \mathbf{x}_1 - \mathbf{w}_2^\top \mathbf{x}_2)\right)\right) \\
&\approx \frac{1}{2L} \sum_{l=1}^{L/2} \exp\left(i(\mathbf{w}_1^{(l)\top} \mathbf{x}_1 - \mathbf{w}_2^{(l)\top} \mathbf{x}_2)\right) \\
&= \frac{1}{2L} \sum_{l=1}^{L/2} \left(\cos\left(\mathbf{w}_1^{(l)\top} \mathbf{x}_1\right) \cos\left(\mathbf{w}_2^{(l)\top} \mathbf{x}_2\right) + \sin\left(\mathbf{w}_1^{(l)\top} \mathbf{x}_1\right) \sin\left(\mathbf{w}_2^{(l)\top} \mathbf{x}_2\right)\right).
\end{aligned}$$
(22)

Since $\mathbf{w}_1$ and $\mathbf{w}_2$ are distinct, it is hard to derive a closed-form RFF approximation for $k(\mathbf{x}_1, \mathbf{x}_2)$. More specifically, it is challenging to explicitly define $\phi_{\mathbf{w}_1}(\cdot), \phi_{\mathbf{w}_2}(\cdot)$, the feature maps of the kernel function, such that

$$k(\mathbf{x}_1, \mathbf{x}_2) \approx \phi_{\mathbf{w}_1}(\mathbf{x}_1)^\top \phi_{\mathbf{w}_2}(\mathbf{x}_2) = \phi_{\mathbf{w}_1}(\mathbf{x}_2)^\top \phi_{\mathbf{w}_2}(\mathbf{x}_1)$$
(23)

Thus, the inversion of the BSM kernel matrix retains high computational complexity, rendering it unsuitable for multi-view data.

**Remark 1.** *To enhance the kernel capacity, this paper proposes the Universal Bochner's Theorem (Theorem 1) and the NG-SM kernel. The main contribution of Theorem 1 is that it relaxes the PSD assumption of the spectral density, thus the induced NG-SM kernel can mitigate the constraint of identical spectral variance.*

**Remark 2.** *To derive a closed-form feature map for any kernel, one potential approach is to decompose the spectral density $S(\mathbf{w}_1, \mathbf{w}_2)$ into some density functions $p(\mathbf{w}_1, \mathbf{w}_2)$ [45, 46], such as:*

$$S(\mathbf{w}_1, \mathbf{w}_2) = \frac{1}{4}(p(\mathbf{w}_1, \mathbf{w}_2) + p(\mathbf{w}_2, \mathbf{w}_1) + p(\mathbf{w}_1)\delta(\mathbf{w}_2 - \mathbf{w}_1) + p(\mathbf{w}_2)\delta(\mathbf{w}_1 - \mathbf{w}_2)), \quad (24)$$

*where $p(\mathbf{w}_1)$ and $p(\mathbf{w}_2)$ are the marginal distributions of $p(\mathbf{w}_1, \mathbf{w}_2)$, and $\delta(x)$ denotes the Dirac delta function. Subsequently, MC integration can be applied to $S(\mathbf{w}_1, \mathbf{w}_2)$ to derive the closed-form feature map, see details in Appendix C.1.*

## B.2 Proof of Theorem 1

($\Longrightarrow$) Suppose there exists a continuous kernel $k(\mathbf{x}_1, \mathbf{x}_2)$ on $\mathbb{R}^D$. By the Theorem 5, this kernel can be represented as:

$$k(\mathbf{x}_1, \mathbf{x}_2) = \int \exp\left(i(\mathbf{w}_1^\top \mathbf{x}_1 - \mathbf{w}_2^\top \mathbf{x}_2)\right) m(\mathrm{d}\mathbf{w}_1, \mathrm{d}\mathbf{w}_2),$$

where $m$ is the Lebesgue-Stieltjes measure associated with some PSD function $S(\mathbf{w}_1, \mathbf{w}_2)$ of bounded variation.

To ensure that the kernel function is exchangeable and PSD, we design the spectral density $S(\mathbf{w}_1, \mathbf{w}_2)$ as follows:

$$S(\mathbf{w}_1, \mathbf{w}_2) = \frac{1}{4}(p(\mathbf{w}_1, \mathbf{w}_2) + p(\mathbf{w}_2, \mathbf{w}_1) + p(\mathbf{w}_1)\delta(\mathbf{w}_2 - \mathbf{w}_1) + p(\mathbf{w}_2)\delta(\mathbf{w}_1 - \mathbf{w}_2)), \quad (25)$$

where $\delta(x)$ represents the Dirac delta function, and $p$ is a certain density function that can be decomposed from $S$ as Eq. (25). Additionally, $p(\mathbf{w}_1)$ and $p(\mathbf{w}_2)$ are the marginal distributions of $p(\mathbf{w}_1, \mathbf{w}_2)$.

The resulting kernel $k(\mathbf{x}_1, \mathbf{x}_2) =$

$$\frac{1}{4} \left( \int \exp\left(i\left(\mathbf{w}_1^\top \mathbf{x}_1 - \mathbf{w}_2^\top \mathbf{x}_2\right)\right) p(\mathbf{w}_1, \mathbf{w}_2) \, \mathrm{d}\mathbf{w}_1 \, \mathrm{d}\mathbf{w}_2 + \int \exp\left(i\left(\mathbf{w}_2^\top \mathbf{x}_1 - \mathbf{w}_1^\top \mathbf{x}_2\right)\right) p(\mathbf{w}_2, \mathbf{w}_1) \, \mathrm{d}\mathbf{w}_2 \, \mathrm{d}\mathbf{w}_1 \right.$$

$$\left. + \int \exp\left(i\mathbf{w}_1^\top (\mathbf{x}_1 - \mathbf{x}_2)\right) p(\mathbf{w}_1) \delta(\mathbf{w}_2 - \mathbf{w}_1) \, \mathrm{d}\mathbf{w}_1 + \int \exp\left(i\mathbf{w}_2^\top (\mathbf{x}_1 - \mathbf{x}_2)\right) p(\mathbf{w}_2) \delta(\mathbf{w}_1 - \mathbf{w}_2) \, \mathrm{d}\mathbf{w}_2 \right)$$

$$= \frac{1}{4} \left( \int \exp\left(i\left(\mathbf{w}_1^\top \mathbf{x}_1 - \mathbf{w}_2^\top \mathbf{x}_2\right)\right) u(\mathrm{d}\mathbf{w}_1, \mathrm{d}\mathbf{w}_2) + \int \exp\left(i\left(\mathbf{w}_2^\top \mathbf{x}_1 - \mathbf{w}_1^\top \mathbf{x}_2\right)\right) u(\mathrm{d}\mathbf{w}_1, \mathrm{d}\mathbf{w}_2) \right.$$

$$\left. + \int \exp\left(i\left(\mathbf{w}_1^\top \mathbf{x}_1 - \mathbf{w}_1^\top \mathbf{x}_2\right)\right) u(\mathrm{d}\mathbf{w}_1) + \int \exp\left(i\left(\mathbf{w}_2^\top \mathbf{x}_1 - \mathbf{w}_2^\top \mathbf{x}_2\right)\right) u(\mathrm{d}\mathbf{w}_2) \right),$$

$$(26)$$

where $u$ is the Lebesgue-Stieltjes measure associated with the density function $p(\mathbf{w}_1, \mathbf{w}_2)$.

Note that we can disregard the $\delta$ functions in the last two terms of the expression, as the integrands in these terms depend solely on $\mathbf{w}_1$ or $\mathbf{w}_2$. Consequently, we can only integrate over the single variable, while setting the other variable to be equal to the one being integrated.

Finally, we can express the kernel:

$$k(\mathbf{x}_1, \mathbf{x}_2) = \frac{1}{4} \int \left( \exp\left(i\left(\mathbf{w}_1^\top \mathbf{x}_1 - \mathbf{w}_2^\top \mathbf{x}_2\right)\right) + \exp\left(i\left(\mathbf{w}_2^\top \mathbf{x}_1 - \mathbf{w}_1^\top \mathbf{x}_2\right)\right) \right.$$

$$\left. + \exp\left(i\mathbf{w}_1^\top (\mathbf{x}_1 - \mathbf{x}_2)\right) + \exp\left(i\mathbf{w}_2^\top (\mathbf{x}_1 - \mathbf{x}_2)\right) \right) u(\mathrm{d}\mathbf{w}_1, \mathrm{d}\mathbf{w}_2), \quad (27)$$

which is the expression shown in Theorem 1.

($\Longleftarrow$) Given the function:

$$k(\mathbf{x}_1, \mathbf{x}_2) = \frac{1}{4} \int \left( \exp\left(i\mathbf{w}_1^\top \mathbf{x}_1 - i\mathbf{w}_2^\top \mathbf{x}_2\right) + \exp\left(i\mathbf{w}_2^\top \mathbf{x}_1 - i\mathbf{w}_1^\top \mathbf{x}_2\right) \right.$$

$$\left. + \exp\left(i\mathbf{w}_1^\top (\mathbf{x}_1 - \mathbf{x}_2)\right) + \exp\left(i\mathbf{w}_2^\top (\mathbf{x}_1 - \mathbf{x}_2)\right) \right) u(\mathrm{d}\mathbf{w}_1, \mathrm{d}\mathbf{w}_2), \quad (28)$$

we have the condition that:

$$S(\mathbf{w}_1, \mathbf{w}_2) = \frac{1}{4}(p(\mathbf{w}_1, \mathbf{w}_2) + p(\mathbf{w}_2, \mathbf{w}_1) + p(\mathbf{w}_1)\delta(\mathbf{w}_2 - \mathbf{w}_1) + p(\mathbf{w}_2)\delta(\mathbf{w}_1 - \mathbf{w}_2)), \quad (29)$$

is a PSD function. We aim to demonstrate that $k(\mathbf{x}_1, \mathbf{x}_2)$ is a valid kernel function. Specifically, we need to show that $k$ is both symmetric and PSD.

First, it is straightforward to show that $k(\mathbf{x}_1, \mathbf{x}_2) = k(\mathbf{x}_2, \mathbf{x}_1)$, confirming that $k$ is symmetric.

The next step is to establish that $k$ is PSD. Consider that $k(\mathbf{x}_1, \mathbf{x}_2) =$

$$\frac{1}{4}\left(\int \exp\bigl(i(\mathbf{w}_1^\top \mathbf{x}_1 - \mathbf{w}_2^\top \mathbf{x}_2))\bigr)p(\mathbf{w}_1, \mathbf{w}_2)\,\mathrm{d}\mathbf{w}_1\mathrm{d}\mathbf{w}_2 + \int \exp\bigl(i(\mathbf{w}_1^\top \mathbf{x}_1 - \mathbf{w}_1^\top \mathbf{x}_2))\bigr)p(\mathbf{w}_1, \mathbf{w}_2)\,\mathrm{d}\mathbf{w}_1\mathrm{d}\mathbf{w}_2\right.$$

$$\left. + \int \exp\bigl(i(\mathbf{w}_2^\top \mathbf{x}_1 - \mathbf{w}_1^\top \mathbf{x}_2))\bigr)p(\mathbf{w}_1, \mathbf{w}_2)\,\mathrm{d}\mathbf{w}_1\mathrm{d}\mathbf{w}_2 + \int \exp\bigl(i(\mathbf{w}_2^\top \mathbf{x}_1 - \mathbf{w}_2^\top \mathbf{x}_2))\bigr)p(\mathbf{w}_1, \mathbf{w}_2)\,\mathrm{d}\mathbf{w}_1\mathrm{d}\mathbf{w}_2\right)$$

$$= \frac{1}{4}\left(\int \exp\bigl(i(\mathbf{w}_1^\top \mathbf{x}_1 - \mathbf{w}_2^\top \mathbf{x}_2))\bigr)p(\mathbf{w}_1, \mathbf{w}_2)\,\mathrm{d}\mathbf{w}_1\mathrm{d}\mathbf{w}_2 + \int \exp\bigl(i(\mathbf{w}_1^\top \mathbf{x}_1 - \mathbf{w}_2^\top \mathbf{x}_2))\bigr)p(\mathbf{w}_1)\delta(\mathbf{w}_2 - \mathbf{w}_1)\,\mathrm{d}\mathbf{w}_1\mathrm{d}\mathbf{w}_2\right.$$

$$\left. + \int \exp\bigl(i(\mathbf{w}_1^\top \mathbf{x}_1 - \mathbf{w}_2^\top \mathbf{x}_2))\bigr)p(\mathbf{w}_2, \mathbf{w}_1)\,\mathrm{d}\mathbf{w}_1\mathrm{d}\mathbf{w}_2 + \int \exp\bigl(i(\mathbf{w}_1^\top \mathbf{x}_1 - \mathbf{w}_2^\top \mathbf{x}_2))\bigr)p(\mathbf{w}_2)\delta(\mathbf{w}_1 - \mathbf{w}_2)\,\mathrm{d}\mathbf{w}_1\mathrm{d}\mathbf{w}_2\right)$$

$$= \int \exp\bigl(i(\mathbf{w}_1^\top \mathbf{x}_1 - \mathbf{w}_2^\top \mathbf{x}_2))\bigr)\frac{1}{4}(p(\mathbf{w}_1, \mathbf{w}_2) + p(\mathbf{w}_2, \mathbf{w}_1) + p(\mathbf{w}_1)\delta(\mathbf{w}_2 - \mathbf{w}_1) + p(\mathbf{w}_2)\delta(\mathbf{w}_1 - \mathbf{w}_2))\,\mathrm{d}\mathbf{w}_1\mathrm{d}\mathbf{w}_2$$

$$= \int \exp\bigl(i(\mathbf{w}_1^\top \mathbf{x}_1 - \mathbf{w}_2^\top \mathbf{x}_2))\bigr)m(\mathrm{d}\mathbf{w}_1\mathrm{d}\mathbf{w}_2),$$

(30)

where $m$ is the Lebesgue-Stieltjes measure associated with the PSD density function $S(\mathbf{w}_1, \mathbf{w}_2)$. Thus, by Theorem 5, $k(\mathbf{w}_1, \mathbf{w}_2)$ is PSD. $\square$

## B.3 Derivation of Next-Gen SM kernel

The BSM kernel based on Theorem 5 is constrained by the requirement of identical variances for $\mathbf{w}_1$ and $\mathbf{w}_2$, and is incompatible with the RFF approximation technique. In this section, we derive the NG-SM kernel based on Theorem 1, which effectively resolves these limitations.

The spectral density of the NG-SM kernel is designed as:

$$p_{\text{ngsm}}(\mathbf{w}_1, \mathbf{w}_2) = \sum_{q=1}^{Q} \alpha_q s_q(\mathbf{w}_1, \mathbf{w}_2), \quad (31)$$

with each

$$s_q(\mathbf{w}_1, \mathbf{w}_2) = \frac{1}{2}\mathcal{N}\left(\begin{pmatrix}\mathbf{w}_1\\\mathbf{w}_2\end{pmatrix}\middle|\begin{pmatrix}\boldsymbol{\mu}_{q1}\\\boldsymbol{\mu}_{q2}\end{pmatrix}, \begin{bmatrix}\boldsymbol{\Sigma}_1 & \boldsymbol{\Sigma}_{\text{c}}^\top\\\boldsymbol{\Sigma}_{\text{c}} & \boldsymbol{\Sigma}_2\end{bmatrix}\right) + \frac{1}{2}\mathcal{N}\left(\begin{pmatrix}-\mathbf{w}_1\\-\mathbf{w}_2\end{pmatrix}\middle|\begin{pmatrix}\boldsymbol{\mu}_{q1}\\\boldsymbol{\mu}_{q2}\end{pmatrix}, \begin{bmatrix}\boldsymbol{\Sigma}_1 & \boldsymbol{\Sigma}_{\text{c}}^\top\\\boldsymbol{\Sigma}_{\text{c}} & \boldsymbol{\Sigma}_2\end{bmatrix}\right).$$

(32)

We simplify the notation by omitting the index $q$ from the sub-matrices $\boldsymbol{\Sigma}_1 = \text{diag}(\boldsymbol{\sigma}_{q1}^2)$, $\boldsymbol{\Sigma}_2 = \text{diag}(\boldsymbol{\sigma}_{q2}^2)$, and $\boldsymbol{\Sigma}_{\text{c}} = \rho_q \text{diag}(\boldsymbol{\sigma}_{q1})\text{diag}(\boldsymbol{\sigma}_{q2})$, where $\boldsymbol{\sigma}_{q1}^2, \boldsymbol{\sigma}_{q2}^2 \in \mathbb{R}^D$ and $\rho_q$ represents the correlation between $\mathbf{w}_1$ and $\mathbf{w}_2$. These terms together define the covariance matrix for the $q$-th bivariate Gaussian component. Additionally, the vectors $\boldsymbol{\mu}_{q1}$ and $\boldsymbol{\mu}_{q2} \in \mathbb{R}^D$ serve as the mean of the $q$-th bivariate Gaussian component.

Relying on Theorem 1, we derive the NG-SM kernel $k_{\text{ngsm}}(\mathbf{x}_1, \mathbf{x}_2) =$

$$\frac{1}{4}\int p_{\text{ngsm}}\left(\mathbf{w}_1,\mathbf{w}_2\right)\Big(\exp\!\big(i\mathbf{w}_1^\top\mathbf{x}_1 - i\mathbf{w}_2^\top\mathbf{x}_2\big) + \exp\!\big(i\mathbf{w}_2^\top\mathbf{x}_1 - i\mathbf{w}_1^\top\mathbf{x}_2\big)$$

$$+ \exp\!\big(i\mathbf{w}_1^\top\mathbf{x}_1 - i\mathbf{w}_1^\top\mathbf{x}_2\big) + \exp\!\big(i\mathbf{w}_2^\top\mathbf{x}_1 - i\mathbf{w}_2^\top\mathbf{x}_2\big)\Big)\mathrm{d}\mathbf{w}_1\mathrm{d}\mathbf{w}_2$$

$$= \frac{1}{4}\sum_{q=1}^{Q}\alpha_q\int\frac{1}{2}\big(\Phi_q(\mathbf{w}_1,\mathbf{w}_2) + \Phi_q(-\mathbf{w}_1,-\mathbf{w}_2)\big)\Big(\exp\!\big(i\mathbf{w}_1^\top\mathbf{x}_1 - i\mathbf{w}_2^\top\mathbf{x}_2\big)$$

$$+\exp\!\big(i\mathbf{w}_2^\top\mathbf{x}_1 - i\mathbf{w}_1^\top\mathbf{x}_2\big) + \exp\!\big(i\mathbf{w}_1^\top\mathbf{x}_1 - i\mathbf{w}_1^\top\mathbf{x}_2\big) + \exp\!\big(i\mathbf{w}_2^\top\mathbf{x}_1 - i\mathbf{w}_2^\top\mathbf{x}_2\big)\Big)\mathrm{d}\mathbf{w}_1\mathrm{d}\mathbf{w}_2,$$

where

$$\Phi_q(\mathbf{w}_1,\mathbf{w}_2) = \mathcal{N}\left(\begin{pmatrix}\mathbf{w}_1\\\mathbf{w}_2\end{pmatrix}\middle|\ \begin{pmatrix}\boldsymbol{\mu}_{q1}\\\boldsymbol{\mu}_{q2}\end{pmatrix}, \begin{bmatrix}\boldsymbol{\Sigma}_1 & \boldsymbol{\Sigma}_{\mathrm{c}}^\top\\\boldsymbol{\Sigma}_{\mathrm{c}} & \boldsymbol{\Sigma}_2\end{bmatrix}\right). \tag{33}$$

We focus solely on the real part of the kernel function. Since the real part of the integrand is a cosine function, and both $\Phi_q$ and the cosine function are even functions, we can therefore simplify the expression as follows.

$$= \frac{1}{4}\sum_{q=1}^{Q}\alpha_q\int\Phi_q(\mathbf{w}_1,\mathbf{w}_2)\Big(\exp\!\big(i\mathbf{w}_1^\top\mathbf{x}_1 - i\mathbf{w}_2^\top\mathbf{x}_2\big) + \exp\!\big(i\mathbf{w}_2^\top\mathbf{x}_1 - i\mathbf{w}_1^\top\mathbf{x}_2\big)$$

$$+ \exp\!\big(i\mathbf{w}_1^\top\mathbf{x}_1 - i\mathbf{w}_1^\top\mathbf{x}_2\big) + \exp\!\big(i\mathbf{w}_2^\top\mathbf{x}_1 - i\mathbf{w}_2^\top\mathbf{x}_2\big)\Big)\mathrm{d}\mathbf{w}_1\mathrm{d}\mathbf{w}_2$$

$$= \frac{1}{4}\sum_{q=1}^{Q}\alpha_q\Bigg(\underbrace{\int\Phi_q(\mathbf{w}_1,\mathbf{w}_2)\exp\!\big(i\mathbf{w}_1^\top\mathbf{x}_1 - i\mathbf{w}_2^\top\mathbf{x}_2\big)\mathrm{d}\mathbf{w}_1\mathrm{d}\mathbf{w}_2}_{\text{Term (1)}}$$

$$+ \underbrace{\int\Phi_q(\mathbf{w}_2,\mathbf{w}_1)\exp\!\big(i\mathbf{w}_2^\top\mathbf{x}_1 - i\mathbf{w}_1^\top\mathbf{x}_2\big)\mathrm{d}\mathbf{w}_1\mathrm{d}\mathbf{w}_2}_{\text{Term (2)}}$$

$$+ \underbrace{\int\Phi_q(\mathbf{w}_1)\exp\!\big(i\mathbf{w}_1^\top\mathbf{x}_1 - i\mathbf{w}_1^\top\mathbf{x}_2\big)\mathrm{d}\mathbf{w}_1}_{\text{Term (3)}} + \underbrace{\int\Phi_q(\mathbf{w}_2)\exp\!\big(i\mathbf{w}_2^\top\mathbf{x}_1 - i\mathbf{w}_2^\top\mathbf{x}_2\big)\mathrm{d}\mathbf{w}_2}_{\text{Term (4)}}\Bigg),$$

$$\tag{34}$$

where $\Phi_q(\mathbf{w}_1)$ and $\Phi_q(\mathbf{w}_2)$ are marginal distributions of $\Phi_q(\mathbf{w}_1,\mathbf{w}_2)$. Next, we will derive the closed forms for each term. First, Term (1) is given by:

$$\text{Term (1)} = \int\mathcal{N}\left(\mathbf{w}\mid\boldsymbol{\mu}_q,\boldsymbol{\Sigma}_q\right)e^{\mathbf{w}^\top\tilde{\mathbf{x}}}\mathrm{d}\mathbf{w}$$

$$= \frac{1}{(2\pi)^2|\boldsymbol{\Sigma}_q|}\int\exp\left(-\frac{1}{2}\left(\mathbf{w}-\boldsymbol{\mu}_q\right)^\top\boldsymbol{\Sigma}_q^{-1}\left(\mathbf{w}-\boldsymbol{\mu}_q\right) + \mathbf{w}^\top\tilde{\mathbf{x}}\right)\mathrm{d}\mathbf{w}$$

$$= \frac{1}{(2\pi)^2|\boldsymbol{\Sigma}_q|}\int\exp\left(-\frac{1}{2}\mathbf{w}^\top\boldsymbol{\Sigma}_q^{-1}\mathbf{w} + \mathbf{w}^\top\left(\tilde{\mathbf{x}} + \boldsymbol{\Sigma}_q^{-1}\boldsymbol{\mu}_q\right) - \frac{1}{2}\boldsymbol{\mu}_q^\top\boldsymbol{\Sigma}_q^{-1}\boldsymbol{\mu}_q\right)\mathrm{d}\mathbf{w} \tag{35}$$

$$= \exp\left(\frac{1}{2}\left(\tilde{\mathbf{x}} + \boldsymbol{\Sigma}_q^{-1}\boldsymbol{\mu}_q^\top\right)^\top\boldsymbol{\Sigma}_q\left(\tilde{\mathbf{x}} + \boldsymbol{\Sigma}_q^{-1}\boldsymbol{\mu}_q^\top\right)\right)\exp\left(-\frac{1}{2}\boldsymbol{\mu}_q^\top\boldsymbol{\Sigma}_q^{-1}\boldsymbol{\mu}_q\right)$$

$$= \exp\left(\frac{1}{2}\tilde{\mathbf{x}}^\top\boldsymbol{\Sigma}_q\tilde{\mathbf{x}} + \boldsymbol{\mu}_q^\top\tilde{\mathbf{x}}\right),$$

where we defined $\tilde{\mathbf{x}} = (i\mathbf{x}_1, -i\mathbf{x}_2)$ and $\mathbf{w} = (\mathbf{w}_1, \mathbf{w}_2)$. In addition, $\boldsymbol{\mu}_q = (\boldsymbol{\mu}_{q1}, \boldsymbol{\mu}_{q2})$ and

$$\boldsymbol{\Sigma}_q = \begin{bmatrix}\boldsymbol{\Sigma}_1 & \boldsymbol{\Sigma}_{\mathrm{c}}^\top\\\boldsymbol{\Sigma}_{\mathrm{c}} & \boldsymbol{\Sigma}_2\end{bmatrix}.$$

The first term of the kernel mixture is then given by:

$$
\begin{aligned}
\text{Term (2)} &= \exp\left(\frac{1}{2}\tilde{\mathbf{x}}^\top \boldsymbol{\Sigma}_q \tilde{\mathbf{x}} + \boldsymbol{\mu}_q^\top \tilde{\mathbf{x}}\right) \\
&= \exp\left(-\frac{1}{2}\left(\mathbf{x}_1^\top \boldsymbol{\Sigma}_1 \mathbf{x}_1 - 2\mathbf{x}_1^\top \boldsymbol{\Sigma}_c \mathbf{x}_2 + \mathbf{x}_2^\top \boldsymbol{\Sigma}_2 \mathbf{x}_2\right)\right) \exp\left(i\left(\boldsymbol{\mu}_{q1}^\top \mathbf{x}_1 - \boldsymbol{\mu}_{q2}^\top \mathbf{x}_2\right)\right) \quad (36) \\
&= \exp\left(-\frac{1}{2}\left(\mathbf{x}_1^\top \boldsymbol{\Sigma}_1 \mathbf{x}_1 - 2\mathbf{x}_1^\top \boldsymbol{\Sigma}_c \mathbf{x}_2 + \mathbf{x}_2^\top \boldsymbol{\Sigma}_2 \mathbf{x}_2\right)\right) \cos\left(\boldsymbol{\mu}_{q1}^\top \mathbf{x}_1 - \boldsymbol{\mu}_{q2}^\top \mathbf{x}_2\right).
\end{aligned}
$$

By swapping $\mathbf{x}_1$ and $\mathbf{x}_2$ in Term (1), the closed form of Term (2) can be easily obtained as below:

$$
\text{Term (2)} = \exp\left(-\frac{1}{2}\left(\mathbf{x}_2 \boldsymbol{\Sigma}_1 \mathbf{x}_2^\top - 2\mathbf{x}_1 \boldsymbol{\Sigma}_c \mathbf{x}_2^\top + \mathbf{x}_1 \boldsymbol{\Sigma}_2 \mathbf{x}_1^\top\right)\right) \cos\left(\boldsymbol{\mu}_{q1}\mathbf{x}_2^\top - \boldsymbol{\mu}_{q2}\mathbf{x}_1^\top\right). \quad (37)
$$

Term (3) of the kernel is then given by:

$$
\begin{aligned}
\text{Term (3)} &= \int \mathcal{N}\left(\mathbf{w}_1 \mid \boldsymbol{\mu}_{q1}, \boldsymbol{\Sigma}_1\right) \exp\left(i\mathbf{w}_1^\top(\mathbf{x}_1 - \mathbf{x}_2)\right) \mathrm{d}\mathbf{w}_1 \\
&= \frac{1}{(2\pi)^2 |\boldsymbol{\Sigma}_1|} \int \exp\left(-\frac{1}{2}(\mathbf{w}_1 - \boldsymbol{\mu}_{q1})^\top \boldsymbol{\Sigma}_1^{-1}(\mathbf{w}_1 - \boldsymbol{\mu}_{q1}) + i\mathbf{w}_1^\top(\mathbf{x}_1 - \mathbf{x}_2)\right)\mathrm{d}\mathbf{w}_1 \\
&= \frac{1}{(2\pi)^2 |\boldsymbol{\Sigma}_1|} \int \exp\left(-\frac{1}{2}\mathbf{w}_1^\top \boldsymbol{\Sigma}_1^{-1}\mathbf{w}_1 + \mathbf{w}_1^\top\left(i(\mathbf{x}_1 - \mathbf{x}_2) + \boldsymbol{\Sigma}_1^{-1}\boldsymbol{\mu}_{q1}\right) - \frac{1}{2}\boldsymbol{\mu}_{q1}^\top \boldsymbol{\Sigma}_1^{-1}\boldsymbol{\mu}_{q1}\right)\mathrm{d}\mathbf{w}_1 \\
&= \exp\left(\frac{1}{2}\left(i(\mathbf{x}_1 - \mathbf{x}_2) + \boldsymbol{\Sigma}_1^{-1}\boldsymbol{\mu}_{q1}\right)^\top \boldsymbol{\Sigma}_1\left(i(\mathbf{x}_1 - \mathbf{x}_2) + \boldsymbol{\Sigma}_1^{-1}\boldsymbol{\mu}_{q1}\right)\right)\exp\left(-\frac{1}{2}\boldsymbol{\mu}_{q1}^\top \boldsymbol{\Sigma}_1^{-1}\boldsymbol{\mu}_{q1}\right) \\
&= \exp\left(\frac{1}{2}i(\mathbf{x}_1 - \mathbf{x}_2)^\top \boldsymbol{\Sigma}_1 i(\mathbf{x}_1 - \mathbf{x}_2) + \boldsymbol{\mu}_{q1}^\top i(\mathbf{x}_1 - \mathbf{x}_2)\right) \\
&= \exp\left(-\frac{1}{2}(\mathbf{x}_1 - \mathbf{x}_2)^\top \boldsymbol{\Sigma}_1(\mathbf{x}_1 - \mathbf{x}_2)\right)\exp\left(i\boldsymbol{\mu}_{q1}^\top(\mathbf{x}_1 - \mathbf{x}_2)\right) \\
&= \exp\left(-\frac{1}{2}(\mathbf{x}_1 - \mathbf{x}_2)^\top \boldsymbol{\Sigma}_1(\mathbf{x}_1 - \mathbf{x}_2)\right)\cos\left(\boldsymbol{\mu}_{q1}^\top(\mathbf{x}_1 - \mathbf{x}_2)\right).
\end{aligned}
$$
$$(38)$$

The 4'th term of the kernel can be derived in a manner similar to the 3'rd term.

$$
\text{Term (4)} = \exp\left(-\frac{1}{2}(\mathbf{x}_1 - \mathbf{x}_2)^\top \boldsymbol{\Sigma}_2(\mathbf{x}_1 - \mathbf{x}_2)\right)\cos\left(\boldsymbol{\mu}_{q2}^\top(\mathbf{x}_1 - \mathbf{x}_2)\right). \quad (39)
$$

Thus, NG-SM kernel takes the form:

$$
\begin{aligned}
k(\mathbf{x}_1, \mathbf{x}_2) = \frac{1}{4}\sum_{q=1}^{Q} \alpha_q \Big[ & \exp\left(-\frac{1}{2}\left(\mathbf{x}_1^\top \boldsymbol{\Sigma}_1 \mathbf{x}_1 - 2\mathbf{x}_1^\top \boldsymbol{\Sigma}_c \mathbf{x}_2 + \mathbf{x}_2^\top \boldsymbol{\Sigma}_2 \mathbf{x}_2\right)\right)\cos\left(\boldsymbol{\mu}_{q1}^\top \mathbf{x}_1 - \boldsymbol{\mu}_{q2}^\top \mathbf{x}_2\right) \\
& + \exp\left(-\frac{1}{2}\left(\mathbf{x}_2^\top \boldsymbol{\Sigma}_1 \mathbf{x}_2 - 2\mathbf{x}_1^\top \boldsymbol{\Sigma}_c \mathbf{x}_2 + \mathbf{x}_1^\top \boldsymbol{\Sigma}_2 \mathbf{x}_1\right)\right)\cos\left(\boldsymbol{\mu}_{q1}^\top \mathbf{x}_2 - \boldsymbol{\mu}_{q2}^\top \mathbf{x}_1\right) \\
& + \exp\left(-\frac{1}{2}(\mathbf{x}_1 - \mathbf{x}_2)^\top \boldsymbol{\Sigma}_1(\mathbf{x}_1 - \mathbf{x}_2)\right)\cos\left(\boldsymbol{\mu}_{q1}^\top(\mathbf{x}_1 - \mathbf{x}_2)\right) \\
& + \exp\left(-\frac{1}{2}(\mathbf{x}_1 - \mathbf{x}_2)^\top \boldsymbol{\Sigma}_2(\mathbf{x}_1 - \mathbf{x}_2)\right)\cos\left(\boldsymbol{\mu}_{q2}^\top(\mathbf{x}_1 - \mathbf{x}_2)\right) \Big].
\end{aligned}
$$
$$(40)$$

## C  Auto-differentiable Next-Gen SM Kernel using RFF Approximation

### C.1  Random Fourier feature for Next-Gen SM Kernel

Our proposed Theorem 1 establishes the following duality: $k(\mathbf{x}_1, \mathbf{x}_2) =$

$$
\begin{aligned}
\frac{1}{4}\mathbb{E}_u \Big( & \exp\left(i(\mathbf{w}_1^\top \mathbf{x}_1 - \mathbf{w}_2^\top \mathbf{x}_2)\right) + \exp\left(i(\mathbf{w}_2^\top \mathbf{x}_1 - \mathbf{w}_1^\top \mathbf{x}_2)\right) \\
& + \exp\left(i(\mathbf{w}_1^\top \mathbf{x}_1 - \mathbf{w}_1^\top \mathbf{x}_2)\right) + \exp\left(i(\mathbf{w}_2^\top \mathbf{x}_1 - \mathbf{w}_2^\top \mathbf{x}_2)\right)\Big).
\end{aligned}
$$

By estimating this expectation with the MC estimator using spectral points $\{\mathbf{w}_1^{(l)}; \mathbf{w}_2^{(l)}\}_{l=1}^{L/2}$ sampled from $p(\mathbf{w}_1, \mathbf{w}_2)$, we can drive

$$
\begin{aligned}
k\left(\mathbf{x}_1, \mathbf{x}_2\right) \approx{} & \frac{1}{2L} \sum_{l=1}^{L/2}\left(\exp\left(i(\mathbf{w}_1^{(l)\top}\mathbf{x}_1 - \mathbf{w}_2^{(l)\top}\mathbf{x}_2)\right) + \exp\left(i(\mathbf{w}_2^{(l)\top}\mathbf{x}_1 - \mathbf{w}_1^{(l)\top}\mathbf{x}_2)\right)\right. \\
& \left. + \exp\left(i(\mathbf{w}_1^{(l)\top}\mathbf{x}_1 - \mathbf{w}_1^{(l)\top}\mathbf{x}_2)\right) + \exp\left(i(\mathbf{w}_2^{(l)\top}\mathbf{x}_1 - \mathbf{w}_2^{(l)\top}\mathbf{x}_2)\right)\right) \\
={} & \frac{1}{2L}\sum_{l=1}^{L/2}\left(\cos\left(\mathbf{w}_1^{(l)\top}\mathbf{x}_1\right)\cos\left(\mathbf{w}_1^{(l)\top}\mathbf{x}_2\right) + \cos\left(\mathbf{w}_1^{(l)\top}\mathbf{x}_1\right)\cos\left(\mathbf{w}_2^{(l)\top}\mathbf{x}_2\right)\right. \\
& + \cos\left(\mathbf{w}_2^{(l)\top}\mathbf{x}_1\right)\cos\left(\mathbf{w}_1^{(l)\top}\mathbf{x}_2\right) + \cos\left(\mathbf{w}_2^{(l)\top}\mathbf{x}_1\right)\cos\left(\mathbf{w}_2^{(l)\top}\mathbf{x}_2\right) \\
& + \sin\left(\mathbf{w}_1^{(l)\top}\mathbf{x}_1\right)\sin\left(\mathbf{w}_1^{(l)\top}\mathbf{x}_2\right) + \sin\left(\mathbf{w}_1^{(l)\top}\mathbf{x}_1\right)\sin\left(\mathbf{w}_2^{(l)\top}\mathbf{x}_2\right) \\
& \left. + \sin\left(\mathbf{w}_2^{(l)\top}\mathbf{x}_1\right)\sin\left(\mathbf{w}_1^{(l)\top}\mathbf{x}_2\right) + \sin\left(\mathbf{w}_2^{(l)\top}\mathbf{x}_1\right)\sin\left(\mathbf{w}_2^{(l)\top}\mathbf{x}_2\right)\right) \\
={} & \phi(\mathbf{x}_1)^\top \phi(\mathbf{x}_2)
\end{aligned}
\tag{41}
$$

where

$$
\phi\left(\mathbf{x}\right) = \sqrt{\frac{1}{2L}}\begin{bmatrix} \cos\left(\mathbf{w}_1^{(1:L/2)\top}\mathbf{x}\right) + \cos\left(\mathbf{w}_2^{(1:L/2)\top}\mathbf{x}\right) \\ \sin\left(\mathbf{w}_1^{(1:L/2)\top}\mathbf{x}\right) + \sin\left(\mathbf{w}_2^{(1:L/2)\top}\mathbf{x}\right) \end{bmatrix} \in \mathbb{R}^L.
\tag{42}
$$

Here the superscript $1:L/2$ indicates that the cosine plus cosine or sine plus sine function is repeated $L/2$ times, with each element corresponding to the one entry of $\{\mathbf{w}_1^{(l)}; \mathbf{w}_2^{(l)}\}_{l=1}^{L/2}$. If we specify the spectral density as $p_{\text{ngsm}}(\mathbf{w}_1, \mathbf{w}_2)$ (Eq. (31)), the estimator of the kernel $k_{\text{ngsm}}(\mathbf{x}_1, \mathbf{x}_2)$ can be formulated as:

$$
k_{\text{ngsm}}(\mathbf{x}_1, \mathbf{x}_2) \approx \phi(\mathbf{x}_1)^\top \phi(\mathbf{x}_2),
\tag{43}
$$

where random features $\phi(\mathbf{x}_1)$ and $\phi(\mathbf{x}_2)$ are constructed using spectral points sampled from $p_{\text{ngsm}}(\mathbf{w}_1, \mathbf{w}_2)$.

### C.2 ELBO Derivation and Evaluation

The term (a) of ELBO is handled numerically with MC estimation as below:

$$
(a) = \sum_{m=1}^{M_v} \mathbb{E}_{q(\mathbf{X}, \mathbf{W})}\left[\log p(\mathbf{y}_{:,m}^v | \mathbf{X}, \mathbf{W}^v)\right]
\tag{44a}
$$

$$
\approx \sum_{m=1}^{M_v} \frac{1}{I} \sum_{i=1}^{I} \log \mathcal{N}(\mathbf{y}_{:,m}^v | \mathbf{0}, \tilde{\mathbf{K}}_{\text{ngsm}}^{v(i)} + \sigma_v^2 \mathbf{I}_N),
\tag{44b}
$$

where $I$ denotes the number of MC samples drawn from $q(\mathbf{X}, \mathbf{W})$. Additionally, $\tilde{\mathbf{K}}_{\text{ngsm}}^{v(i)} = (\mathbf{\Phi}_x^v \mathbf{\Phi}_x^{v\top})^{(i)}$ is the NG-SM kernel gram matrix approximation, where $\Phi_x^v \in \mathbb{R}^{N \times L}$.

The term (b) of ELBO can be evaluated analytically due to the Gaussian nature of the distributions. More specific, we have

$$
(b) = \sum_{n=1}^{N} \text{KL}(q(\mathbf{x}_n) \| p(\mathbf{x}_n))
\tag{45a}
$$

$$
= \frac{1}{2}\sum_{n=1}^{N}\left[\text{tr}(\mathbf{S}_n) + \boldsymbol{\mu}_n^\top \boldsymbol{\mu}_n - \log|\mathbf{S}_n| - D\right],
\tag{45b}
$$

where $D$ represents the dimensionality of $\mathbf{x}_n$, and $\mathbf{S}_n$ is commonly assumed to be a diagonal matrix. Consequently, the ELBO can be expressed as follows:

$$ELBO = \mathbb{E}_{q(\mathbf{X},\mathbf{W})}\left[\frac{p(\mathbf{Y},\mathbf{X};\mathbf{W})}{q(\mathbf{X},\mathbf{W})}\right]$$

$$= \mathbb{E}_{q(\mathbf{X},\mathbf{W})}\left[\log \frac{p(\mathbf{X})\prod_{v=1}^{V}p(\mathbf{W}^v)p(\mathbf{Y}^v|\mathbf{X},\mathbf{W}^v)}{q(\mathbf{X})\prod_{v=1}^{V}p(\mathbf{W}^v)}\right]$$

$$= \sum_{v=1}^{V}\underbrace{\mathbb{E}_{q(\cdot,\cdot)}\left[\log p(\mathbf{Y}^v|\mathbf{X},\mathbf{W}^v)\right]}_{\text{(a): reconstruction}} - \underbrace{\mathrm{KL}(q(\mathbf{X})\|p(\mathbf{X}))}_{\text{(b): regularization}}$$

$$\approx \sum_{v=1}^{V}\sum_{m=1}^{M_v}\frac{1}{I}\sum_{i=1}^{I}\log \mathcal{N}(\mathbf{y}_{:,m}^v|\mathbf{0},\tilde{\mathbf{K}}_{\text{ngsm}}^{v(i)}+\sigma^2\mathbf{I}_N) - \frac{1}{2}\sum_{n=1}^{N}\left[\mathrm{tr}(\mathbf{S}_n)+\boldsymbol{\mu}_n^\top\boldsymbol{\mu}_n - \log|\mathbf{S}_n| - D\right]$$

$$= \sum_{v=1}^{V}\sum_{m=1}^{M_v}\frac{1}{I}\sum_{i=1}^{I}\left\{-\frac{N}{2}\log 2\pi - \frac{1}{2}\log\left|\tilde{\mathbf{K}}_{\text{ngsm}}^{v(i)}+\sigma_v^2\mathbf{I}_N\right| - \frac{1}{2}\mathbf{y}_{:,m}^{v\top}\left(\tilde{\mathbf{K}}_{\text{ngsm}}^{v(i)}+\sigma_v^2\mathbf{I}_N\right)^{-1}\mathbf{y}_{:,m}^v\right\}$$

$$- \frac{1}{2}\sum_{n=1}^{N}\left[\mathrm{tr}(\mathbf{S}_n)+\boldsymbol{\mu}_n^\top\boldsymbol{\mu}_n - \log|\mathbf{S}_n| - D\right].$$

When $N \gg L$, both the determinant and the inverse of $\tilde{\mathbf{K}}_{\text{ngsm}}^{v(i)}+\sigma_v^2\mathbf{I}_N$ can be computed efficiently by the following two lemma [26].

**Lemma 1.** *Suppose $\mathbf{A}$ is an invertible $n$-by-$n$ matrix and $\mathbf{U},\mathbf{V}$ are $n$-by-$m$ matrices. Then the following determinant equality holds.*

$$\left|\mathbf{A}+\mathbf{U}\mathbf{V}^\top\right| = \left|\mathbf{I}_{\mathrm{m}}+\mathbf{V}^\top\mathbf{A}^{-1}\mathbf{U}\right||\mathbf{A}|.$$

**Lemma 2** (Woodbury matrix identity). *Suppose $\mathbf{A}$ is an invertible $n$-by-$n$ matrix and $\mathbf{U},\mathbf{V}$ are $n$-by-$m$ matrices. Then*

$$\left(\mathbf{A}+\mathbf{U}\mathbf{V}^\top\right)^{-1} = \mathbf{A}^{-1} - \mathbf{A}^{-1}\mathbf{U}(\mathbf{I}_{\mathrm{m}}+\mathbf{V}^\top\mathbf{U})^{-1}\mathbf{V}^\top.$$

We can reduce the computational complexity for evaluating the ELBO from the original $\mathcal{O}(N^3)$ to $\mathcal{O}(NL^2)$ as below:

$$\left|\tilde{\mathbf{K}}_{\text{ngsm}}^{v(i)}+\sigma_v^2\mathbf{I}_N\right| = \sigma_v^{2N}\left|\mathbf{I}_L + \frac{1}{\sigma_v^2}(\boldsymbol{\Phi}_x^v\boldsymbol{\Phi}_x^{v\top})^{(i)}\right|, \tag{46}$$

$$\left(\tilde{\mathbf{K}}_{\text{ngsm}}^{v(i)}+\sigma_v^2\mathbf{I}_N\right)^{-1} = \frac{1}{\sigma_v^2}\left[\mathbf{I}_N - \boldsymbol{\Phi}_x^{v(i)}\left(\mathbf{I}_L + (\boldsymbol{\Phi}_x^v\boldsymbol{\Phi}_x^{v\top})^{(i)}\right)^{-1}\boldsymbol{\Phi}_x^{v(i)\top}\right]. \tag{47}$$

### C.3 Treating W variationally

Alternatively, we define the variational distributions as

$$q(\mathbf{X},\mathbf{W}) = q(\mathbf{W};\boldsymbol{\eta})q(\mathbf{X}),$$

where the variational distribution $q(\mathbf{W};\boldsymbol{\eta})$ is also a bivariate Gaussian mixture that is parameterized by $\boldsymbol{\eta}$. By combining these variational distributions with the joint distribution defined in Eq. (11), we derive the following ELBO:

$$ELBO = \mathbb{E}_{q(\mathbf{X},\mathbf{W})}\left[\frac{p(\mathbf{Y},\mathbf{X},\mathbf{W})}{q(\mathbf{W};\boldsymbol{\eta})q(\mathbf{X})}\right]$$

$$= \mathbb{E}_{q(\mathbf{X},\mathbf{W})}\left[\log \frac{p(\mathbf{X})\prod_{v=1}^{V}p(\mathbf{W}^v;\boldsymbol{\theta}_{\text{ngsm}})p(\mathbf{Y}^v|\mathbf{X},\mathbf{W}^v)}{q(\mathbf{X})\prod_{v=1}^{V}p(\mathbf{W}^v;\boldsymbol{\eta})}\right]$$

$$= \sum_{v=1}^{V}\underbrace{\mathbb{E}_{q(\cdot,\cdot)}\left[\log p(\mathbf{Y}^v|\mathbf{X},\mathbf{W}^v)\right]}_{\text{(a): reconstruction}} - \underbrace{\mathrm{KL}(q(\mathbf{X})\|p(\mathbf{X}))}_{\text{(b): regularization of }\mathbf{X}} - \underbrace{\mathrm{KL}(q(\mathbf{W};\boldsymbol{\eta})\|p(\mathbf{W};\boldsymbol{\theta}_{\text{ngsm}}))}_{\text{(c): regularization of }\mathbf{W}}, \tag{48}$$

where we redefine the prior distribution $p(\mathbf{W})$ as $p(\mathbf{W};\boldsymbol{\theta}_{\text{ngsm}})$ to maintain notational consistency. When maximizing the ELBO, we obtain $q(\mathbf{W};\boldsymbol{\eta}) = p(\mathbf{W};\boldsymbol{\theta}_{\text{ngsm}})$, as the optimization variable $\boldsymbol{\theta}_{\text{ngsm}}$ only affects term $c$. Consequently, term $(c)$ becomes zero, and Eq. (48) aligns with the optimization objective outlined in the main text.

### C.4 Proof of Proposition 1

*Proof.* The proposed two-step reparameterization trick leverages sequential simulation, where $\mathbf{w}_{q1}$ is first sampled from $p(\mathbf{w}_{q1})$, followed by $\mathbf{w}_{q2}$ drawn from the conditional distribution $p(\mathbf{w}_{q2}|\mathbf{w}_{q1})$.

1) Sample from $p(\mathbf{w}_{q1})$: Given that $\mathbf{w}_{q1}$ follows a normal distribution $\mathcal{N}(\boldsymbol{\mu}_{q1}, \mathrm{diag}(\boldsymbol{\sigma}_{q1}^2))$, we can directly use the reparameterization trick to sample from it, i.e.,

$$\mathbf{w}_{q1}^{(l)} = \boldsymbol{\mu}_{q1} + \boldsymbol{\sigma}_{q1} \circ \boldsymbol{\epsilon}_1,$$

where $\boldsymbol{\epsilon}_1 \sim \mathcal{N}(\mathbf{0}, \mathbf{I})$.

2) Sample from $p(\mathbf{w}_{q2}|\mathbf{w}_{q1})$: Given $\mathbf{w}_{q1}^{(l)}$, we use the given correlation parameter $\rho_q$ and a new standard normal variable $\boldsymbol{\epsilon}_2$ to generate $\mathbf{w}_{q2}^{(l)}$:

$$\mathbf{w}_{q2}^{(l)} = \boldsymbol{\mu}_{q2} + \rho_q \boldsymbol{\sigma}_{q2} \backslash \boldsymbol{\sigma}_{q1} \circ (\mathbf{w}_{q1}^{(l)} - \boldsymbol{\mu}_{q1}) + \sqrt{1 - \rho_q^2} \boldsymbol{\sigma}_{q2} \circ \boldsymbol{\epsilon}_2.$$

Now, we need to proof that the generated $\mathbf{w}_{q1}^{(l)}$ and $\mathbf{w}_{q2}^{(l)}$ follow the bivariate Gaussian distribution $s_q(\mathbf{w}_1, \mathbf{w}_2)$. To this ends, we compute the mean and variance of $\mathbf{w}_{q2}^{(l)}$, and the covariance between $\mathbf{w}_{q1}^{(l)}$ and $\mathbf{w}_{q2}^{(l)}$ below.

- Mean of $\mathbf{w}_{q2}^{(l)}$:

$$\mathbb{E}[\mathbf{w}_{q2}^{(l)}] = \mathbb{E}\left[\boldsymbol{\mu}_{q2} + \rho_q \boldsymbol{\sigma}_{q2} \backslash \boldsymbol{\sigma}_{q1} \circ (\mathbf{w}_{q1}^{(l)} - \boldsymbol{\mu}_{q1}) + \sqrt{1 - \rho_q^2} \boldsymbol{\sigma}_{q2} \circ \boldsymbol{\epsilon}_2\right].$$

Since $\mathbb{E}[\boldsymbol{\epsilon}_2] = 0$ and $\mathbb{E}[\mathbf{w}_{q1}^{(l)}] = \boldsymbol{\mu}_{q1}$, we have:

$$\mathbb{E}[\mathbf{w}_{q2}^{(l)}] = \boldsymbol{\mu}_{q2} + \rho_q \boldsymbol{\sigma}_{q2} \backslash \boldsymbol{\sigma}_{q1} \circ (\mathbb{E}[\mathbf{w}_{q1}^{(l)}] - \boldsymbol{\mu}_{q1}) + \sqrt{1 - \rho_q^2} \boldsymbol{\sigma}_{q2} \circ \mathbb{E}[\boldsymbol{\epsilon}_2] = \boldsymbol{\mu}_{q2}.$$

- Variance of $\mathbf{w}_{q2}^{(l)}$:

$$\mathrm{Var}(\mathbf{w}_{q2}^{(l)}) = \mathrm{Var}\left(\rho_q \boldsymbol{\sigma}_{q2} \backslash \boldsymbol{\sigma}_{q1} \circ (\mathbf{w}_{q1}^{(l)} - \boldsymbol{\mu}_{q1}) + \sqrt{1 - \rho_q^2} \boldsymbol{\sigma}_{q2} \circ \boldsymbol{\epsilon}_2\right).$$

Since $\mathbf{w}_{q1}^{(l)} - \boldsymbol{\mu}_{q1}$ and $\boldsymbol{\epsilon}_2$ are independent, and $\mathrm{Var}(\boldsymbol{\epsilon}_2) = \mathbf{I}$, we have:

$$\mathrm{Var}(\mathbf{w}_{q2}^{(l)}) = \rho_q^2 (\boldsymbol{\sigma}_{q2} \backslash \boldsymbol{\sigma}_{q1})^2 \circ \mathrm{Var}(\mathbf{w}_{q1}^{(l)}) + (1 - \rho_q^2)\boldsymbol{\sigma}_{q2}^2 = \rho_q^2 \boldsymbol{\sigma}_{q2}^2 + (1 - \rho_q^2)\boldsymbol{\sigma}_{q2}^2 = \boldsymbol{\sigma}_{q2}^2.$$

- Covariance between $\mathbf{w}_{q1}^{(l)}$ and $\mathbf{w}_{q2}^{(l)}$:

$$\mathrm{Cov}(\mathbf{w}_{q1}^{(l)}, \mathbf{w}_{q2}^{(l)}) = \mathrm{Cov}\left(\boldsymbol{\mu}_{q1} + \boldsymbol{\sigma}_{q1} \circ \boldsymbol{\epsilon}_1, \boldsymbol{\mu}_{q2} + \rho_q \boldsymbol{\sigma}_{q2} \backslash \boldsymbol{\sigma}_{q1} \circ (\mathbf{w}_{q1}^{(l)} - \boldsymbol{\mu}_{q1}) + \sqrt{1 - \rho_q^2} \boldsymbol{\sigma}_{q2} \circ \boldsymbol{\epsilon}_2\right).$$
$$(49)$$

Since $\boldsymbol{\mu}_{q1}$ and $\boldsymbol{\mu}_{q2}$ are constants, the covariance only depends on $\boldsymbol{\sigma}_{q1} \circ \boldsymbol{\epsilon}_1$ and $\rho_q \boldsymbol{\sigma}_{q2} \circ \boldsymbol{\epsilon}_1 + \sqrt{1 - \rho_q^2} \boldsymbol{\sigma}_{q2} \circ \boldsymbol{\epsilon}_2$. Thus we can reformulate Eq. (49) as

$$\mathrm{Cov}(\boldsymbol{\sigma}_{q1} \circ \boldsymbol{\epsilon}_1, \rho_q \boldsymbol{\sigma}_{q2} \circ \boldsymbol{\epsilon}_1 + \sqrt{1 - \rho_q^2} \boldsymbol{\sigma}_{q2} \circ \boldsymbol{\epsilon}_2) = \boldsymbol{\sigma}_{q1} \rho_q \circ \boldsymbol{\sigma}_{q2} \mathrm{Cov}(\boldsymbol{\epsilon}_1, \boldsymbol{\epsilon}_1) + \boldsymbol{\sigma}_{q1} \sqrt{1 - \rho_q^2} \circ \boldsymbol{\sigma}_{q2} \mathrm{Cov}(\boldsymbol{\epsilon}_1, \boldsymbol{\epsilon}_2).$$

Given that $\boldsymbol{\epsilon}_1$ and $\boldsymbol{\epsilon}_2$ are independent, $\mathrm{Cov}(\boldsymbol{\epsilon}_1, \boldsymbol{\epsilon}_2) = \mathbf{0}$, and $\mathrm{Cov}(\boldsymbol{\epsilon}_1, \boldsymbol{\epsilon}_1) = \mathbf{I}$, this covariance is equal to:

$$\boldsymbol{\sigma}_{q1} \rho_q \circ \boldsymbol{\sigma}_{q2} \circ \mathbf{I} + \boldsymbol{\sigma}_{q1} \sqrt{1 - \rho_q^2} \circ \boldsymbol{\sigma}_{q2} \circ \mathbf{0} = \rho_q \boldsymbol{\sigma}_{q1} \circ \boldsymbol{\sigma}_{q2}.$$

Therefore, the generated $\mathbf{w}_{q1}^{(l)}$ and $\mathbf{w}_{q2}^{(l)}$ follow the bivariate Gaussian distribution $s_q(\mathbf{w}_1, \mathbf{w}_2)$. $\qquad \square$

The two-step reparameterization trick simplifies the sampling process of the bivariate Gaussian distribution. Specifically, the traditional sampling method [39] requires Cholesky decomposition of the $2D \times 2D$ covariance matrix, resulting in a computational complexity of $\mathcal{O}(8D^3)$. In contrast, our method achieves a computational complexity of $\mathcal{O}(D)$ for sampling from $p(\mathbf{w}_{q1})$, while sampling from $p(\mathbf{w}_{q2}|\mathbf{w}_{q1})$ also maintains a complexity of $\mathcal{O}(D)$. This results in a total computational cost of $\mathcal{O}(D)$.

## C.5 Proof of Theorem 3

*Proof.* With the RFF feature map defined in Eq. (15), we can express the inner product of the feature maps as follows:

$$\varphi\left(\mathbf{x}_1\right)^\top \varphi\left(\mathbf{x}_2\right) = \sum_{q=1}^{Q} \alpha_q \sum_{l=1}^{L/2} \frac{1}{2L} A_q^{(l)}, \tag{50}$$

where,

$$
\begin{aligned}
A_q^{(l)} = \Big( &\cos\Big(\mathbf{w}_{q1}^{(l)\top}\mathbf{x}_1\Big)\cos\Big(\mathbf{w}_{q1}^{(l)\top}\mathbf{x}_2\Big) + \cos\Big(\mathbf{w}_{q1}^{(l)\top}\mathbf{x}_1\Big)\cos\Big(\mathbf{w}_{q2}^{(l)\top}\mathbf{x}_2\Big) \\
&+ \cos\Big(\mathbf{w}_{q2}^{(l)\top}\mathbf{x}_1\Big)\cos\Big(\mathbf{w}_{q1}^{(l)\top}\mathbf{x}_2\Big) + \cos\Big(\mathbf{w}_{q2}^{(l)\top}\mathbf{x}_1\Big)\cos\Big(\mathbf{w}_{q2}^{(l)\top}\mathbf{x}_2\Big) \\
&+ \sin\Big(\mathbf{w}_{q1}^{(l)\top}\mathbf{x}_1\Big)\sin\Big(\mathbf{w}_{q1}^{(l)\top}\mathbf{x}_2\Big) + \sin\Big(\mathbf{w}_{q1}^{(l)\top}\mathbf{x}_1\Big)\sin\Big(\mathbf{w}_{q2}^{(l)\top}\mathbf{x}_2\Big) \\
&+ \sin\Big(\mathbf{w}_{q2}^{(l)\top}\mathbf{x}_1\Big)\sin\Big(\mathbf{w}_{q1}^{(l)\top}\mathbf{x}_2\Big) + \sin\Big(\mathbf{w}_{q2}^{(l)\top}\mathbf{x}_1\Big)\sin\Big(\mathbf{w}_{q2}^{(l)\top}\mathbf{x}_2\Big)\Big),
\end{aligned}
\tag{51}
$$

where $\{\mathbf{w}_1^l, \mathbf{w}_2^l\}_{l=1}^{L/2}$ are independently and identically distributed (i.i.d.) spectral points drawn from the density function $s_q(\mathbf{w}_1, \mathbf{w}_2)$ using the two step reparameterization trick. Taking the expectation with respect to $p_{\text{ngsm}}(\mathbf{w}_1, \mathbf{w}_2) = \prod_{q=1}^{Q}\prod_{l=1}^{L/2} s_q(\mathbf{w}_1, \mathbf{w}_2)$, we obtain:

$$
\mathbb{E}_{p_{\text{ngsm}}(\mathbf{w}_1, \mathbf{w}_2)}\left[\varphi\left(\mathbf{x}_1\right)^\top \varphi\left(\mathbf{x}_2\right)\right] = \mathbb{E}_{p_{\text{ngsm}}(\mathbf{w}_1, \mathbf{w}_2)}\left[\sum_{q=1}^{Q}\alpha_q\sum_{l=1}^{L/2}\frac{1}{2L}A_q^{(l)}\right]
$$

$$
= \sum_{q=1}^{Q}\alpha_q \mathbb{E}_{s\left(\mathbf{w_{q1}}^{1:L/2}, \mathbf{w_{q2}}^{1:L/2}\right)}\left[\sum_{l=1}^{L/2}\frac{1}{2L}A_q^{(l)}\right], \qquad \text{(linearity of expectation)}
$$
$$\tag{52}$$

$$
= \sum_{q=1}^{Q}\alpha_q\frac{1}{4}\mathbb{E}_{s_q(\mathbf{w}_1, \mathbf{w}_2)}[A_q] \qquad\qquad\qquad \text{(i.i.d. of } \mathbf{w}_l^{(q)})
$$

$$
= \sum_{q=1}^{Q}\alpha_q\frac{1}{4}\mathbb{E}_{s_q(\mathbf{w}_1, \mathbf{w}_2)}\Big[\exp\big(i(\mathbf{w}_1\mathbf{x}_1^\top - \mathbf{w}_2\mathbf{x}_2^\top)\big) + \exp\big(i(\mathbf{w}_2\mathbf{x}_1^\top - \mathbf{w}_1\mathbf{x}_2^\top)\big)
$$

$$
\qquad + \exp\big(i(\mathbf{w}_1\mathbf{x}_1^\top - \mathbf{w}_1\mathbf{x}_2^\top)\big) + \exp\big(i(\mathbf{w}_2\mathbf{x}_1^\top - \mathbf{w}_2\mathbf{x}_2^\top)\big)\Big] \qquad \text{(Euler's identity)}
$$

$$
= \sum_{q=1}^{Q}\alpha_q k_q(\mathbf{x}_1, \mathbf{x}_2)
$$

$$
= k_{\text{ngsm}}(\mathbf{x}_1, \mathbf{x}_2). \qquad\qquad\qquad\qquad \text{(NG-SM kernel definition)}
$$
$$\tag{53}$$

Thus, we conclude that $\varphi\left(\mathbf{x}_1\right)^\top \varphi\left(\mathbf{x}_2\right)$ provides an unbiased estimator for the NG-SM kernel.

$\square$

## C.6 Proof of Theorem 4

*Proof.* We primarily rely on the Matrix Bernstein inequality [47] to establish Theorem 4, with the proof outline depicted in Figure 5.

**Lemma 3** (Matrix Bernstein Inequality). *Consider a finite sequence $\{\mathbf{E}_i\}$ of independent, random, Hermitian matrices with dimension $N$. Assume that*

$$\mathbb{E}[\mathbf{E}_i] = \mathbf{0} \text{ and } \|\mathbf{E}_i\|_2 \leq H \text{ for each index } i,$$

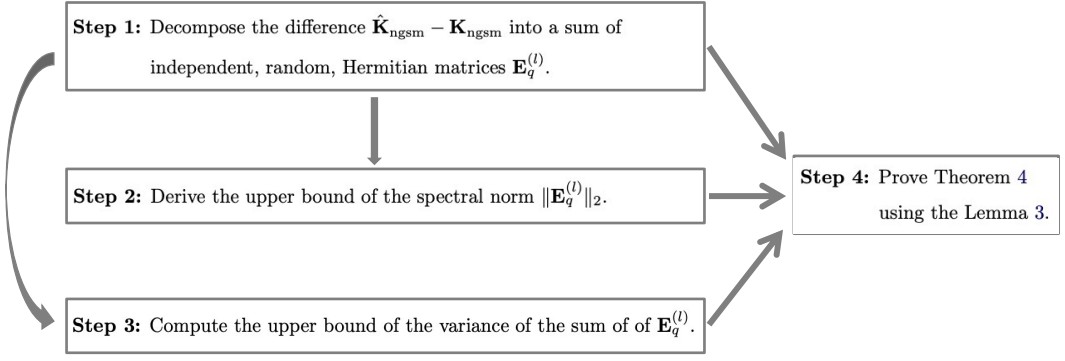

**Figure 5:** Flowchart for proving Theorem 4

where $\|\cdot\|_2$ denotes the matrix spectral norm. Introduce the random matrix $\mathbf{E} = \sum_i \mathbf{E}_i$, and let $v(\mathbf{E})$ be the matrix variance statistic of the sum:

$$v(\mathbf{E}) = \left\| \mathbb{E}[\mathbf{E}^2] \right\| = \left\| \sum_i \mathbb{E}[\mathbf{E}_i^2] \right\|.$$

Then we have

$$\mathbb{E}\left[\|\mathbf{E}\|_2\right] \leq \sqrt{2v(\mathbf{E})\log N} + \frac{1}{3}L\log N. \tag{54}$$

Furthermore, for all $\epsilon \geq 0$.

$$P\left\{\|\mathbf{E}\|_2 \geq \epsilon\right\} \leq N \cdot \exp\left(\frac{-\epsilon^2/2}{v(\mathbf{E}) + H\epsilon/3}\right). \tag{55}$$

*Proof.* The proof of Lemma 3 can be found in Theorem 6.6.1 in [47]. $\qquad\square$

**Step 1:** With the constructed NG-SM kernel matrix approximation, $\hat{\mathbf{K}}_{\text{ngsm}} = \Phi_{\text{ngsm}}(\mathbf{X})\Phi_{\text{ngsm}}(\mathbf{X})^\top$, where the random feature matrix $\Phi_{\text{ngsm}}(\mathbf{X}) = \left[\varphi\left(\mathbf{x}_1\right), \ldots, \varphi\left(\mathbf{x}_N\right)\right]^\top \in \mathbb{R}^{N \times QL}$, we have the following approximation error matrix:

$$\mathbf{E} = \hat{\mathbf{K}}_{\text{ngsm}} - \mathbf{K}_{\text{ngsm}}. \tag{56}$$

We are going to show that $\mathbf{E}$ can be factorized as

$$\mathbf{E} = \sum_{q=1}^{Q}\sum_{l=1}^{L/2}\mathbf{E}_q^{(l)}, \tag{57}$$

where $\mathbf{E}_q^{(l)}$ is a sequence of independent, random, Hermitian matrices with dimension $N$.

Sample $\mathbf{w}_{q1}^{(l)}, \mathbf{w}_{q2}^{(l)}$ from $s_q(\mathbf{w}_1, \mathbf{w}_2)$, and we can show that

$$[\hat{\mathbf{K}}_{\text{ngsm}}]_{h,g} = \sum_{q=1}^{Q}\frac{\alpha_q}{2L}\sum_{l=1}^{L/2}A_q^{(l)}(h,g),$$

where $A_q^{(l)}(h,g) =$

$$\left(\cos\left(\mathbf{w}_{q1}^{(l)\top}\mathbf{x}_h\right)\cos\left(\mathbf{w}_{q1}^{(l)\top}\mathbf{x}_g\right) + \cos\left(\mathbf{w}_{q1}^{(l)\top}\mathbf{x}_h\right)\cos\left(\mathbf{w}_{q2}^{(l)\top}\mathbf{x}_g\right) + \cos\left(\mathbf{w}_{q2}^{(l)\top}\mathbf{x}_h\right)\cos\left(\mathbf{w}_{q1}^{(l)\top}\mathbf{x}_g\right)\right.$$

$$+ \cos\left(\mathbf{w}_{q2}^{(l)\top}\mathbf{x}_h\right)\cos\left(\mathbf{w}_{q2}^{(l)\top}\mathbf{x}_g\right) + \sin\left(\mathbf{w}_{q1}^{(l)\top}\mathbf{x}_h\right)\sin\left(\mathbf{w}_{q1}^{(l)\top}\mathbf{x}_g\right) + \sin\left(\mathbf{w}_{q1}^{(l)\top}\mathbf{x}_h\right)\sin\left(\mathbf{w}_{q2}^{(l)\top}\mathbf{x}_g\right)$$

$$+ \sin\left(\mathbf{w}_{q2}^{(l)\top}\mathbf{x}_h\right)\sin\left(\mathbf{w}_{q1}^{(l)\top}\mathbf{x}_g\right) + \left.\sin\left(\mathbf{w}_{q2}^{(l)\top}\mathbf{x}_h\right)\sin\left(\mathbf{w}_{q2}^{(l)\top}\mathbf{x}_g\right)\right).$$

$$\tag{58}$$

Thus, we have $\hat{\mathbf{K}}_{\text{ngsm}} = \sum_{q=1}^{Q} \sum_{l=1}^{L/2} \frac{\alpha_q}{2L} A_q^{(l)}$. Based on this factorization and Eq. (52) in Proposition 3, we have that

$$\mathbf{K}_{\text{ngsm}} = \sum_{q=1}^{Q} \sum_{l=1}^{L/2} \frac{\alpha_q}{2L} \mathbb{E}[A_q^{(l)}].$$

Therefore, the approximation error matrix $\mathbf{E}$ can be factorized as $\mathbf{E} = \sum_{q=1}^{Q} \sum_{l=1}^{L/2} \mathbf{E}_q^{(l)}$ where

$$\mathbf{E}_q^{(l)} = \frac{\alpha_q}{2L} \left( A_q^{(l)} - \mathbb{E}[A_q^{(l)}] \right) \tag{59}$$

is a sequence of independent, random, Hermitian matrices with dimension $N$ that satisfy the condition of $\mathbb{E}[\mathbf{E}_q^{(l)}] = \mathbf{0}$.

**Step 2:** We can bound the $\|\mathbf{E}_q^{(l)}\|_2$ by following.

$$\|\mathbf{E}_q^{(l)}\|_2 = \frac{\alpha_q}{2L} \left\| A_q^{(l)} - \mathbb{E}\left[ A_q^{(l)} \right] \right\|_2 \tag{60a}$$

$$\leq \frac{\alpha_q}{2L} \left( \left\| A_q^{(l)} \right\|_2 + \left\| \mathbb{E}\left[ A_q^{(l)} \right] \right\|_2 \right) \qquad \text{(triangle inequality)} \tag{60b}$$

$$\leq \frac{\alpha_q}{2L} \left( \left\| A_q^{(l)} \right\|_2 + \mathbb{E}\left[ \left\| A_q^{(l)} \right\|_2 \right] \right) \qquad \text{(Jensen's inequality)} \tag{60c}$$

$$\leq \frac{C}{2L} (8N + 8N) \tag{60d}$$

$$= \frac{C}{2L} 16N = \frac{8CN}{L}, \tag{60e}$$

where $C = \sqrt{\sum_{q=1}^{Q} \alpha_q^2}$ and

$$\mathbf{c}_{qk}^{(l)} = \left[ \cos\left( \mathbf{w}_{qk}^{(l)\top} \mathbf{x}_1 \right), \ldots, \cos\left( \mathbf{w}_{qk}^{(l)\top} \mathbf{x}_N \right) \right]^\top \in \mathbb{R}^N,$$

$$\mathbf{s}_{qk}^{(l)} = \left[ \sin\left( \mathbf{w}_{qk}^{(l)\top} \mathbf{x}_1 \right), \ldots, \sin\left( \mathbf{w}_{qk}^{(l)\top} \mathbf{x}_N \right) \right]^\top \in \mathbb{R}^N,$$

$$A_q^{(l)} = \mathbf{c}_{q1}^{(l)} \mathbf{c}_{q1}^{(l)\top} + \mathbf{s}_{q1}^{(l)} \mathbf{s}_{q1}^{(l)\top} + \mathbf{c}_{q2}^{(l)} \mathbf{c}_{q2}^{(l)\top} + \mathbf{s}_{q2}^{(l)} \mathbf{s}_{q2}^{(l)\top} + \mathbf{c}_{q1}^{(l)} \mathbf{c}_{q2}^{(l)\top} + \mathbf{s}_{q1}^{(l)} \mathbf{s}_{q2}^{(l)\top} + \mathbf{c}_{q2}^{(l)} \mathbf{c}_{q1}^{(l)\top} + \mathbf{s}_{q2}^{(l)} \mathbf{s}_{q1}^{(l)\top}. \tag{61}$$

We are interested in bounding the spectral norm of $A_q^{(l)}$, where

$$A_q^{(l)} = \left( \mathbf{c}_{q1}^{(l)} \mathbf{c}_{q1}^{(l)\top} + \mathbf{s}_{q1}^{(l)} \mathbf{s}_{q1}^{(l)\top} + \mathbf{c}_{q2}^{(l)} \mathbf{c}_{q2}^{(l)\top} + \mathbf{s}_{q2}^{(l)} \mathbf{s}_{q2}^{(l)\top} + \mathbf{c}_{q1}^{(l)} \mathbf{c}_{q2}^{(l)\top} + \mathbf{s}_{q1}^{(l)} \mathbf{s}_{q2}^{(l)\top} + \mathbf{c}_{q2}^{(l)} \mathbf{c}_{q1}^{(l)\top} + \mathbf{s}_{q2}^{(l)} \mathbf{s}_{q1}^{(l)\top} \right).$$

- $\mathbf{c}_{qk}^{(\ell)}$ and $\mathbf{s}_{qk}^{(\ell)}$ are $N$-dimensional vectors, where each element is given by a cosine or sine function. Thus, we have: $\|\mathbf{c}_{qk}^{(\ell)}\|_2 \leq \sqrt{N}, \quad \|\mathbf{s}_{qk}^{(\ell)}\|_2 \leq \sqrt{N}.$

- For any vector $\mathbf{a}, \mathbf{b} \in \mathbb{R}^N$, the spectral norm of $\mathbf{a}\mathbf{b}^\top$ satisfies: $\|\mathbf{a}\mathbf{b}^\top\|_2 = \|\mathbf{a}\|_2 \cdot \|\mathbf{b}\|_2$. Note that each term in $A_q^{(\ell)}$ is of the form $\mathbf{a}\mathbf{b}^\top$. Therefore, since $\|\mathbf{a}\|_2 \leq \sqrt{N}$ and $\|\mathbf{b}\|_2 \leq \sqrt{N}$ as shown earlier, we have: $\|\mathbf{a}\mathbf{b}^\top\|_2 \leq N$, for all terms in $A_q^{(\ell)}$.

Since $A_q^{(\ell)}$ is the sum of 8 such terms, we apply the subadditivity of the spectral norm::

$$\|A_q^{(\ell)}\|_2 \leq \sum_{j=1}^{8} \|\mathbf{a}_j \mathbf{b}_j^\top\|_2 \leq 8N.$$

The inequality (C.6) serves as a valid upper bound. However, it is not necessarily tight, since the spectral norm of a sum of rank-one matrices is typically less than the sum of their individual norms unless the corresponding vectors are perfectly aligned. At last, this upper bound leads directly to the inequality in Eq. (60d).

**Step 3 :** We first have the following bound:

$$\frac{4L^2}{\alpha_q^2}\mathbb{E}\left[\left(\mathbf{E}_q^{(l)}\right)^2\right] = \mathbb{E}\left[(A_q^{(l)})^2\right] - \left(\mathbb{E}\left[A_q^{(l)}\right]\right)^2 \tag{62a}$$

$$\preccurlyeq \mathbb{E}\left[(A_q^{(l)})^2\right] \tag{62b}$$

$$= \mathbb{E}\left[\sum_{\mathbf{k}}\left(\boldsymbol{c}_{qk_1}^{(l)}\boldsymbol{c}_{qk_2}^{(l)\top}\boldsymbol{c}_{qk_3}^{(l)}\boldsymbol{c}_{qk_4}^{(l)\top} + \boldsymbol{c}_{qk_1}^{(l)}\boldsymbol{c}_{qk_2}^{(l)\top}\boldsymbol{s}_{qk_3}^{(l)}\boldsymbol{s}_{qk_4}^{(l)\top} + \boldsymbol{s}_{qk_1}^{(l)}\boldsymbol{s}_{qk_2}^{(l)\top}\boldsymbol{c}_{qk_3}^{(l)}\boldsymbol{c}_{qk_4}^{(l)\top} + \boldsymbol{s}_{qk_1}^{(l)}\boldsymbol{s}_{qk_2}^{(l)\top}\boldsymbol{s}_{qk_3}^{(l)}\boldsymbol{s}_{qk_4}^{(l)\top}\right)\right] \tag{62c}$$

$$= \mathbb{E}\left[\sum_{\mathbf{k}}\left((\boldsymbol{c}_{qk_2}^{(l)\top}\boldsymbol{c}_{qk_3}^{(l)})\boldsymbol{c}_{qk_1}^{(l)}\boldsymbol{c}_{qk_4}^{(l)\top} + (\boldsymbol{s}_{qk_2}^{(l)\top}\boldsymbol{s}_{qk_3}^{(l)})\boldsymbol{s}_{qk_1}^{(l)}\boldsymbol{s}_{qk_4}^{(l)\top} + (\boldsymbol{s}_{qk_2}^{(l)\top}\boldsymbol{c}_{qk_3}^{(l)})(\boldsymbol{s}_{qk_1}^{(l)}\boldsymbol{c}_{qk_4}^{(l)\top} + \boldsymbol{c}_{qk_1}^{(l)}\boldsymbol{s}_{qk_4}^{(l)\top})\right)\right] \tag{62d}$$

$$= \sum_{\mathbf{k}}\left(\mathbb{E}\left[(\boldsymbol{c}_{qk_2}^{(l)\top}\boldsymbol{c}_{qk_3}^{(l)})\boldsymbol{c}_{qk_1}^{(l)}\boldsymbol{c}_{qk_4}^{(l)\top} + (\boldsymbol{s}_{qk_2}^{(l)\top}\boldsymbol{s}_{qk_3}^{(l)})\boldsymbol{s}_{qk_1}^{(l)}\boldsymbol{s}_{qk_4}^{(l)\top} + (\boldsymbol{s}_{qk_2}^{(l)\top}\boldsymbol{c}_{qk_3}^{(l)})(\boldsymbol{s}_{qk_1}^{(l)}\boldsymbol{c}_{qk_4}^{(l)\top} + \boldsymbol{c}_{qk_1}^{(l)}\boldsymbol{s}_{qk_4}^{(l)\top})\right]\right) \tag{62e}$$

$$\preccurlyeq \sum_{\mathbf{k}}\left(N\mathbb{E}\left[\boldsymbol{c}_{qk_1}^{(l)}\boldsymbol{c}_{qk_4}^{(l)\top} + \boldsymbol{s}_{qk_1}^{(l)}\boldsymbol{s}_{qk_4}^{(l)\top}\right] + \mathbb{E}\left[(\boldsymbol{s}_{qk_2}^{(l)\top}\boldsymbol{c}_{qk_3}^{(l)})(\boldsymbol{s}_{qk_1}^{(l)}\boldsymbol{c}_{qk_4}^{(l)\top} + \boldsymbol{c}_{qk_1}^{(l)}\boldsymbol{s}_{qk_4}^{(l)\top})\right]\right) \tag{62f}$$

$$= 4N\mathbb{E}\left[A_q^{(l)}\right] + \sum_{\mathbf{k}}\left(\mathbb{E}\left[(\boldsymbol{s}_{qk_2}^{(l)\top}\boldsymbol{c}_{qk_3}^{(l)})(\boldsymbol{s}_{qk_1}^{(l)}\boldsymbol{c}_{qk_4}^{(l)\top} + \boldsymbol{c}_{qk_1}^{(l)}\boldsymbol{s}_{qk_4}^{(l)\top})\right]\right), \tag{62g}$$

where $\sum_{\mathbf{k}} = \sum_{k_1=1}^2\sum_{k_2=1}^2\sum_{k_3=1}^2\sum_{k_4=1}^2$, and the notation $\mathbf{A} \preccurlyeq \mathbf{B}$ denotes that $\mathbf{B} - \mathbf{A}$ is a PSD matrix. To ensure the validity of the inequality in Eq. (62b), it is necessary to show that $\left(\mathbb{E}[A_q^{(l)}]\right)^2$ is a positive semi-definite (PSD) matrix.

Since $A_q^{(\ell)}$ is a real symmetric matrix, its expectation $\mathbb{E}[A_q^{(\ell)}]$ is also real symmetric. Therefore, we can apply the following result:

**Lemma 4** (Square of a Real Symmetric Matrix is PSD). *Let $M \in \mathbb{R}^{n\times n}$ be a real symmetric matrix. Then its square $M^2 := MM$ is positive semi-definite, i.e., $M^2 \succeq 0$.*

*Proof.* Since $M$ is symmetric, it admits an eigendecomposition of the form $M = U\Lambda U^\top$, where $U \in \mathbb{R}^{n\times n}$ is an orthogonal matrix and $\Lambda \in \mathbb{R}^{n\times n}$ is a real diagonal matrix of eigenvalues. Then,

$$M^2 = MM = U\Lambda U^\top U\Lambda U^\top = U\Lambda^2 U^\top,$$

where $\Lambda^2$ is the diagonal matrix of squared eigenvalues of $M$. Since the square of any real number is non-negative, all eigenvalues of $M^2$ are non-negative. Therefore, $M^2$ is PSD. $\qquad\square$

Applying Lemma 4 to $\mathbb{E}[A_q^{(l)}]$, we conclude that $\left(\mathbb{E}[A_q^{(l)}]\right)^2$ is PSD. This confirms that the inequality in Eq. (62b) holds.

The inequality in Eq. (62f) holds because

$$N\mathbb{E}\left[\boldsymbol{c}_{qk_1}^{(l)}\boldsymbol{c}_{qk_4}^{(l)\top} + \boldsymbol{s}_{qk_1}^{(l)}\boldsymbol{s}_{qk_4}^{(l)\top}\right] - \mathbb{E}\left[(\boldsymbol{c}_{qk_2}^{(l)\top}\boldsymbol{c}_{qk_3}^{(l)})\boldsymbol{c}_{qk_1}^{(l)}\boldsymbol{c}_{qk_4}^{(l)\top} + (\boldsymbol{s}_{qk_2}^{(l)\top}\boldsymbol{s}_{qk_3}^{(l)})\boldsymbol{s}_{qk_1}^{(l)}\boldsymbol{s}_{qk_4}^{(l)\top}\right]$$
$$= \mathbb{E}\left[(\boldsymbol{s}_{qk_2}^{(l)\top}\boldsymbol{s}_{qk_3}^{(l)})\boldsymbol{c}_{qk_1}^{(l)}\boldsymbol{c}_{qk_4}^{(l)\top} + (\boldsymbol{c}_{qk_2}^{(l)\top}\boldsymbol{c}_{qk_3}^{(l)})\boldsymbol{s}_{qk_1}^{(l)}\boldsymbol{s}_{qk_4}^{(l)\top}\right]$$
$$\left(\text{due to } \boldsymbol{c}_{qk_2}^{(l)}\boldsymbol{c}_{qk_3}^{(l)\top} + \boldsymbol{s}_{qk_2}^{(l)}\boldsymbol{s}_{qk_3}^{(l)\top} = \sum_{j=1}^N(\cos\left(\mathbf{w}_{qk_2}^{(l)\top} - \mathbf{w}_{qk_3}^{(l)\top}\right)\mathbf{x}_j) \preccurlyeq N\right) \tag{63}$$

is a PSD matrix.

Then we are able to bound the variance, $\left\| \sum_{q=1}^{Q} \sum_{l=1}^{L/2} \mathbb{E}[(\mathbf{E}_q^{(l)})^2] \right\|_2$, as

$$
\begin{aligned}
& \left\| \sum_{q=1}^{Q} \sum_{l=1}^{L/2} \mathbb{E}[(\mathbf{E}_q^{(l)})^2] \right\|_2 \\
& \leq \left\| \sum_{q=1}^{Q} \sum_{l=1}^{L/2} \frac{\alpha_q^2}{4L^2} \left( 4N\mathbb{E}\left[A_q^{(l)}\right] + \sum_{\mathbf{k}} \left( \mathbb{E}\left[ (\boldsymbol{s}_{qk_2}^{(l)\top} \boldsymbol{c}_{qk_3}^{(l)})(\boldsymbol{s}_{qk_1}^{(l)} \boldsymbol{c}_{qk_4}^{(l)\top} + \boldsymbol{c}_{qk_1}^{(l)} \boldsymbol{s}_{qk_4}^{(l)\top}) \right] \right) \right) \right\|_2 \\
& \leq \frac{2C}{L} \left\| \sum_{q=1}^{Q} \alpha_q \left( \frac{N}{4} \mathbb{E}\left[A_q^{(l)}\right] + \frac{1}{16} \sum_{\mathbf{k}} \left( \mathbb{E}\left[ (\boldsymbol{s}_{qk_2}^{(l)\top} \boldsymbol{c}_{qk_3}^{(l)})(\boldsymbol{s}_{qk_1}^{(l)} \boldsymbol{c}_{qk_4}^{(l)\top} + \boldsymbol{c}_{qk_1}^{(l)} \boldsymbol{s}_{qk_4}^{(l)\top}) \right] \right) \right) \right\|_2 \\
& \leq \frac{2C}{L} \left( N \left\| \mathbf{K}_{\text{ngsm}} \right\|_2 + \frac{1}{16} \sum_{\mathbf{k}} \sum_{q=1}^{Q} \alpha_q \left\| \mathbb{E}\left[ (\boldsymbol{s}_{qk_2}^{(l)\top} \boldsymbol{c}_{qk_3}^{(l)})(\boldsymbol{s}_{qk_1}^{(l)} \boldsymbol{c}_{qk_4}^{(l)\top} + \boldsymbol{c}_{qk_1}^{(l)} \boldsymbol{s}_{qk_4}^{(l)\top}) \right] \right\|_2 \right) \quad \text{(triangle inequality)} \\
& \leq \frac{2C}{L} \left( N \left\| \mathbf{K}_{\text{ngsm}} \right\|_2 + \frac{1}{16} \sum_{\mathbf{k}} \sum_{q=1}^{Q} \alpha_q \mathbb{E}\left[ \left\| (\boldsymbol{s}_{qk_2}^{(l)\top} \boldsymbol{c}_{qk_3}^{(l)})(\boldsymbol{s}_{qk_1}^{(l)} \boldsymbol{c}_{qk_4}^{(l)\top} + \boldsymbol{c}_{qk_1}^{(l)} \boldsymbol{s}_{qk_4}^{(l)\top}) \right\|_2 \right] \right) \quad \text{(Jensen's inequality)} \\
& \leq \frac{2C}{L} \left( N \left\| \mathbf{K}_{\text{ngsm}} \right\|_2 + \frac{4}{16} \sum_{k_1=1}^{2} \sum_{k_4=1}^{2} \frac{N}{2} \sum_{q=1}^{Q} \alpha_q \mathbb{E}\left[ \left\| (\boldsymbol{s}_{qk_1}^{(l)} \boldsymbol{c}_{qk_4}^{(l)\top} + \boldsymbol{c}_{qk_1}^{(l)} \boldsymbol{s}_{qk_4}^{(l)\top}) \right\|_2 \right] \right) \left( |(\boldsymbol{s}_{qk_2}^{(l)\top} \boldsymbol{c}_{qk_3}^{(l)})| \leq \frac{N}{2} \right) \\
& \leq \frac{2CN}{L} \left( \left\| \mathbf{K}_{\text{ngsm}} \right\|_2 + \frac{1}{16} * \frac{N}{2} * 16C\sqrt{Q} \right),
\end{aligned}
$$

(64)

where the last inequality is because that

$$
\mathbb{E}\left[ \left\| (\boldsymbol{s}_{qk_1}^{(l)} \boldsymbol{c}_{qk_4}^{(l)\top} + \boldsymbol{c}_{qk_1}^{(l)} \boldsymbol{s}_{qk_4}^{(l)\top}) \right\|_2 \right] = \sup_{\|\boldsymbol{v}\|_2^2 = 1} \mathbb{E}\left[ \left\| \boldsymbol{v}^\top (\boldsymbol{s}_{qk_1}^{(l)} \boldsymbol{c}_{qk_4}^{(l)\top} + \boldsymbol{c}_{qk_1}^{(l)} \boldsymbol{s}_{qk_4}^{(l)\top}) \boldsymbol{v} \right\|_2 \right] \leq N, \quad (65)
$$

and $\sum_{q=1}^{Q} \alpha_q \leq C\sqrt{Q}$ by the Cauchy–Schwarz inequality.

**Step 4 :**   We can now apply the derived upper bounds, given by Eqs. (60) and (64), to $H$ and $v(\mathbf{E})$ in Lemma 3,

$$
P\left( \left\| \hat{\mathbf{K}}_{\text{ngsm}} - \mathbf{K}_{\text{ngsm}} \right\|_2 \geq \epsilon \right) \leq N \exp\left( \frac{-3\epsilon^2 L}{2NC \left( 6 \left\| \mathbf{K}_{\text{ngsm}} \right\|_2 + 3NC\sqrt{Q} + 8\epsilon \right)} \right), \quad (66)
$$

which completes the proof of Theorem 4.

$\square$

# D   Experimental Details

All experiments were conducted on a cloud server equipped with 2 NVIDIA Tesla V100 GPUs (16GB memory each), a 10-core Intel Xeon Platinum 81xx series CPU, 64GB RAM, and 200GB storage.

## D.1   Dataset Description

This section presents a detailed overview of the datasets used in our experiments, covering the generation process of synthetic data, the description of real-world data, and the applied preprocessing steps.

### D.1.1   Synthetic Data

The datasets are generated using a 2-view MV-GPLVM with an $S$-shaped latent variable, employing two different kernel configurations: (1) both views use the RBF kernel, and (2) one view uses the RBF kernel while the other uses the Gibbs kernel. Detailed descriptions of the kernels are provided below.

- **RBF Kernel**: The kernel function is expressed as:

$$k(\mathbf{x}_1, \mathbf{x}_2) = \ell_o \exp\left(-\frac{\|\mathbf{x}_1 - \mathbf{x}_2\|^2}{2\ell_l^2}\right)$$

where $\ell_o = 1$ denotes the outputscale, and $\ell_l = 1$ represents the lengthscale.

- **Gibbs Kernel**: As a non-stationary kernel, the kernel function is formulated as:

$$k(\mathbf{x}_1, \mathbf{x}_2) = \sqrt{\frac{2\ell_{\mathbf{x}_1}\ell_{\mathbf{x}_2}}{\ell_{\mathbf{x}_1}^2 + \ell_{\mathbf{x}_2}^2}} \exp\left(-\frac{\|\mathbf{x}_1 - \mathbf{x}_2\|^2}{\ell_{\mathbf{x}_1}^2 + \ell_{\mathbf{x}_2}^2}\right)$$

In this context, $\ell_{\mathbf{x}}$ is dynamic length scales derived from the positions of the input point $\mathbf{x}$, specifically defined as:

$$\ell_{\mathbf{x}} = \exp\left(-0.5 \cdot \|\mathbf{x}\|\right).$$

### D.1.2   Real-World Data

**Table 4:** Description of real-world datasets.

| DATASET | # SAMPLES (N) | # DIMENSIONS (M) | # LABELS |
|---------|---------------|------------------|----------|
| BRIDGES | 214 | 4 | 2 |
| CIFAR | 60,000 | $20 \times 20$ | 10 |
| R-CIFAR | 2,000 | $20 \times 20$ | 5 |
| MNIST | 70,000 | $28 \times 28$ | 10 |
| R-MNIST | 1,000 | $28 \times 28$ | 10 |
| NEWSGROUPS | 2752 | 19 | 3 |
| YALE | 165 | 1850 | 15 |
| CHANNEL | 1,000 | 64 | 10 |
| BRENDAN | 2,000 | $20 \times 28$ | - |

The detailed preprocessing steps and descriptions of the real-world datasets are provided below. For convenience, key information about the datasets is summarized in Table 4.

- 1) BRIDGES: This dataset is a collection of data documenting the daily bicycle counts crossing four East River bridges in New York City[10] (Brooklyn, Williamsburg, Manhattan, and Queensboro). We classify the data into weekdays and weekends, treating these as binary labels. This classification aims to examine the variations in bicycle counts on weekdays versus weekends, exploring whether significant differences exist in the traffic patterns between the two.

- 2) CIFAR: This dataset comprises 60,000 color images with a resolution of 32×32 pixels, categorized into 10 distinct classes: airplane, automobile, bird, cat, deer, dog, frog, horse, ship, and truck. These images were further resized from $32 \times 32$ pixels to $20 \times 20$ pixels and converted to grayscale for training.

---

[10] https://data.cityofnewyork.us/Transportation/Bicycle-Counts-for-East-River-Bridges/gua4-p9wg

- 3) R-CIFAR: We sample this dataset from the CIFAR dataset, specifically selecting five categories: airplane, automobile, bird, cat, and deer. For each category, 400 images are sampled, and each 32×32 pixel image is converted into a 20×20 pixel image.

- 4) MNIST: It is a classic handwritten digit recognition dataset. It consists of 70,000 grayscale images with a resolution of $28 \times 28$ pixels, divided into 60,000 training samples and 10,000 testing samples. Each image represents a handwritten digit ranging from 0 to 9.

- 5) R-MNIST: We select 1,000 randomly handwritten digit images from the classic MNIST dataset.

- 6) NEWSGROUPS: It is a dataset for text classification, containing articles from multiple newsgroups[11]. We restrict the vocabulary to words that appear within a document frequency range of 10% to 90%. For our analysis, we specifically select text from three classes: comp.sys.mac.hardware, sci.med, and alt.atheism.

- 7) YALE: The Yale Faces Dataset[12] consists of face images from 15 different individuals, captured under various lighting conditions, facial expressions, and viewing angles.

- 8) BRENDAN: The dataset contains 2,000 photos of Brendan's face[13].

- 9) MNIST–SVHN: Two paired views: MNIST digits (784 dimensions) and SVHN digits (3072 dimensions), each belonging to one of 10 classes.

- 10) MOVIES: Extracted from IMDb, where each movie is represented by a keyword vector (1878 dimensions) and an actor vector (1398 dimensions)[14]. The dataset consists of 617 movies categorized into 17 genre classes.

- 11) CORA: A citation network dataset in which each document is described by two views: a bag-of-words content vector (1433 dimensions) and a citation structure vector (2708 dimensions). It contains 2708 documents spanning 7 research topic classes[15].

### D.1.3  Channel Data

To evaluate model performance under realistic wireless environments, we simulate channel data using the QUAsi Deterministic RadIo channel GenerAtor (QuaDRiGa)[16], a widely-used geometry-based stochastic channel simulator. The parameter settings used in our experiments are listed in Table 5.

The simulation scenario emulates a typical urban communication setting, where the user equipment (UE) moves at a constant speed of $30\,km/h$. QuaDRiGa accounts for both the speed and the direction of movement, which significantly influence the channel behavior. Specifically, the relative motion between the UE and the base station (BS) introduces Doppler shifts and time-varying path loss. As the UE moves through the environment, the geometry between the transmitter, receiver, and scatterers changes over time, leading to dynamic variations in multipath components such as delays, angles, and Doppler frequencies.

It is worth noting that no explicit additive noise (e.g., Gaussian noise) is introduced in the simulation. Instead, the randomness in the channel arises naturally from QuaDRiGa's stochastic modeling of multipath fading, spatial non-stationarity, and time dynamics. These controlled yet realistic variations ensure consistency across different simulation runs while providing sufficient complexity to evaluate the robustness of the proposed model.

### D.2  Benchmark Methods

We provide a description of the benchmark methods as follows:

- 1) PCA: A classical linear dimensionality reduction method that projects data to orthogonal components maximizing variance.

- 2) LDA: A supervised linear method that finds projections maximizing class separability.

---

[11] http://qwone.com/~jason/20Newsgroups/

[12] http://vision.ucsd.edu/content/yale-face-database

[13] https://cs.nyu.edu/~roweis/data/frey_rawface.mat

[14] https://lig-membres.imag.fr/grimal/data.html

[15] https://lig-membres.imag.fr/grimal/data.html

[16] https://quadriga-channel-model.de

**Table 5:** Parameters settings of QUADRIGA.

| PARAMETER DESCRIPTION | VALUES |
|---|---|
| **Overall Setup** | |
| # samples | 1000 |
| # user equipment (UE) | 10 |
| # receive antennas | 1 |
| Moving speed (km/h) | 30 |
| Proportion of indoor UEs | 1 |
| Time sampling interval (seconds) | 5e-3 |
| Total duration of sampling (seconds) | 5 |
| Channel type | 3GPP_3D_UMa |
| **Channel Configuration** | |
| Center frequency (Hz) | 1.84e9 |
| Use random initial phase | False |
| Use geometric polarization | False |
| Use spherical waves | False |
| Show progress bars | False |
| **Base Station (BS) Antenna Configuration** | |
| # vertical elements per antenna | 4 |
| # horizontal elements per antenna | 1 |
| # rows in the antenna array | 2 |
| # columns in the antenna array | 8 |
| Electrical tilt angle (degrees) | 7 |
| # carriers | 1 |
| # transmit antennas | 32 |
| **UE Antenna Configuration** | |
| UE antenna array | omni |
| # subcarriers | 1 |
| Subcarrier spacing (Hz) | 1e6 |
| # loops (simulations) | 1 |
| Minimum UE distance from BS (meters) | 35 |
| Maximum UE distance from BS (meters) | 300 |
| **Layout Configuration** | |
| # base stations | 1 |
| Base station position (x, y, z) (meters) | (0, 0, 30) |
| UE movement path length (meters) | 41.667 |

- 3) ISOMAP: A nonlinear manifold learning method that preserves geodesic distances between data points.

- 4) HPF: A hierarchical poisson factorization model for probabilistic matrix factorization, commonly used in recommendation systems.

- 5) BGPLVM: A bayesian formulation of GPLVM that places a prior over latent variables and infers their posterior using variational inference.

- 6) GPLVM-SVI: A scalable GPLVM variant using stochastic variational inference to handle large datasets.

- 7) VAE: A deep generative model that learns latent representations via amortized variational inference.

- 8) NBVAE: A probabilistic model for sparse, overdispersed count data, combining a Gaussian process prior with a negative binomial likelihood for flexible, non-linear modeling.

- 9) DCA: The deep count autoencoder is designed to denoise single-cell RNA sequencing data by accounting for count distribution, overdispersion, and sparsity, using a zero-inflated negative binomial noise model.

- 10) CVQ-VAE: The clustering vector quantised variational aautoencoder is an enhanced VQ-VAE model designed to prevent codebook collapse by utilizing an online clustered codebook.

- 11) RFLVM: GPLVM extension using random feature approximation for scalable inference.

- 12) DGPLVM: A deep extension of GPLVM that stacks multiple GP layers to learn hierarchical nonlinear mappings from latent space, typically trained via variational inference.

- 13) ARFLVM: An advised GPLVM using kernel learning and latent regularization for flexible representation learning.

- 14) MVAE: Multi-modal VAE using a product-of-experts (PoE) to infer shared latent representations from multiple modalities.

**Table 6:** Descriptions of benchmark methods.

| METHOD | REFERENCE | IMPLEMENTATION CODE |
|---|---|---|
| PCA | [48] | Using the scikit-learn library [49]. |
| LDA | [50] | Using the scikit-learn library [49]. |
| ISOMAP | [51] | Using the scikit-learn library [49]. |
| HPF | [52] | https://github.com/david-cortes/hpfrec |
| BGPLVM | [53] | https://github.com/SheffieldML/GPy |
| GPLVM-SVI | [23] | https://github.com/vr308/Generalised-GPLVM |
| VAE | [15] | https://github.com/pytorch/examples/blob/main/vae/main.py |
| NBVAE | [54] | https://github.com/ethanheathcote/NBVAE |
| DCA | [55] | https://github.com/theislab/dca |
| CVQ-VAE | [56] | https://github.com/lyndonzheng/CVQ-VAE |
| RFLVM | [24, 57] | https://github.com/gwgundersen/rflvm |
| DGPLVM | [58] | https://github.com/UCL-SML/Doubly-Stochastic-DGP |
| ARFLVM | [21] | https://github.com/zhidilin/advisedGPLVM |
| MVAE | [11] | https://github.com/mhw32/multimodal-vae-public |
| MMVAE | [12, 16] | https://github.com/OpenNLPLab/MMVAE-AVS |

**Table 7:** Default Hyperparameter Settings.

| PARAMETER DESCRIPTION | VALUES |
|---|---|
| **NG-SM Kernel Setup** | |
| # Mixture densities ($Q$) | 2 |
| Dim. of random feature ($L/2$) | 50 |
| Dim. of latent space ($D$) | 2 |
| **Optimizer Setup (Adam)** | |
| Learning rate | 0.01 |
| Beta | (0.9, 0.99) |
| # Iterations | 10000 |

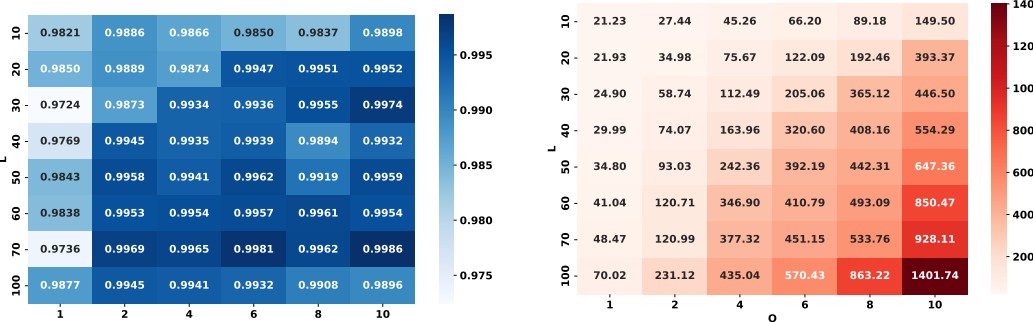

**Figure 6:** Heatmap of $R^2$ in latent manifold learning.

**Figure 7:** Heatmap of wall-time (seconds) in latent manifold learning.

- 15) MMVAE: Mixture-of-experts (MoE) based VAE that balances shared and private representations for multi-modal learning.

Table 6 provides the references and implementation details for each benchmark method, aiming to enhance reproducibility.

### D.3 Hyperparameter Settings

Figure 11 depicts the latent manifold learning outcomes of NG-RFLVM for varying values of $Q$ and $L$ on synthetic single-view data. Figure 6 shows a heatmap of $R^2$ scores, quantifying the similarity between the learned and ground truth latent variables, with values closer to 1 indicating better alignment. Figure 7 displays a heatmap of wall-time across varying values of $Q$ and $L$. These results demonstrate that larger values of $Q$ and $L$ typically improve model performance, albeit at the cost of higher computational complexity. To balance computational efficiency with latent representation quality, we select $Q = 2$ and $L = 50$. The default hyperparameter settings are summarized in Table 7.

**Table 8:** Classification accuracy (%) is evaluated by fitting a KNN classifier ($k = 1$) with five-fold cross-validation. **Mean and standard deviation** are computed over five experiments, and the top performance is in bold.

| DATASET | PCA | LDA | ISOMAP | HPF | BGPLVM | GPLVM-SVI | DGPLVM |
|---|---|---|---|---|---|---|---|
| BRIDGES | $84.10 \pm 0.76$ | $66.81 \pm 5.31$ | $79.77 \pm 2.58$ | $54.42 \pm 10.96$ | $81.85 \pm 3.71$ | $79.61 \pm 1.93$ | $64.75 \pm 4.80$ |
| R-CIFAR | $26.78 \pm 0.22$ | $22.78 \pm 0.66$ | $27.23 \pm 0.66$ | $20.80 \pm 0.64$ | $27.13 \pm 1.46$ | $25.12 \pm 1.24$ | $28.16 \pm 1.86$ |
| R-MNIST | $36.56 \pm 1.23$ | $23.38 \pm 2.69$ | $44.45 \pm 2.14$ | $31.49 \pm 4.02$ | $56.75 \pm 3.37$ | $34.41 \pm 5.48$ | $74.82 \pm 2.24$ |
| NEWSGROUPS | $39.24 \pm 0.52$ | $39.14 \pm 1.89$ | $39.79 \pm 1.04$ | $33.44 \pm 1.91$ | $38.52 \pm 1.05$ | $37.84 \pm 1.88$ | $37.63 \pm 2.48$ |
| YALE | $54.37 \pm 0.87$ | $33.86 \pm 2.38$ | $58.84 \pm 1.72$ | $51.17 \pm 1.96$ | $55.35 \pm 3.65$ | $52.17 \pm 1.59$ | $76.06 \pm 2.60$ |

| DATASET | VAE | NBVAE | DCA | CVQ-VAE | RFLVM | ARFLVM | **OURS** |
|---|---|---|---|---|---|---|---|
| BRIDGES | $75.15 \pm 1.63$ | $75.85 \pm 3.88$ | $70.21 \pm 3.63$ | $68.86 \pm 1.38$ | $84.61 \pm 3.95$ | $84.64 \pm 1.54$ | $\mathbf{85.30 \pm 1.27}$ |
| R-CIFAR | $26.65 \pm 0.25$ | $25.97 \pm 0.50$ | $25.51 \pm 1.90$ | $22.45 \pm 1.23$ | $28.47 \pm 10.34$ | $29.02 \pm 0.62$ | $\mathbf{31.44 \pm 0.68}$ |
| R-MNIST | $64.37 \pm 2.16$ | $28.18 \pm 1.20$ | $17.19 \pm 7.54$ | $12.87 \pm 0.53$ | $60.20 \pm 5.53$ | $79.59 \pm 1.52$ | $\mathbf{80.99 \pm 0.59}$ |
| NEWSGROUPS | $38.52 \pm 0.29$ | $39.87 \pm 1.00$ | $39.97 \pm 3.41$ | $35.60 \pm 1.96$ | $41.35 \pm 0.95$ | $\mathbf{41.82 \pm 0.75}$ | $40.10 \pm 1.48$ |
| YALE | $61.16 \pm 2.04$ | $45.60 \pm 4.68$ | $28.49 \pm 5.40$ | $33.89 \pm 0.28$ | $65.37 \pm 6.79$ | $76.56 \pm 1.02$ | $\mathbf{76.60 \pm 1.96}$ |

## D.4 Additional Experiments

To explore more general scenarios, we first conduct single-view experiments to validate our model's representation capability (App. D.4.1), and then extend the evaluation to more complex multi-view settings, including synthetic data (App. D.1.1), multi-view MNIST with a large number of views (App. D.4.3) and more challenging multi-view datasets (App. D.4.4). Finally, we provide qualitative visualization of the learned representations (App. D.4.5).

### D.4.1 Single-View Data

In this section, we first examine the following single-view data types: images (R-MNIST, YALE, R-CIFAR), text (NEWSGROUPS), and structured data (BRIDGES). To accommodate the high computational costs of RFLVM, the dataset sizes for CIFAR and MNIST are reduced and denoted with the prefix 'R-' (see details in App. D.1.2).

The results[17] in Table 8 present the mean and standard deviation of classification accuracy from a five-fold cross-validated K-nearest neighbor (KNN) classification, performed on the learned latent variables for each dataset and method.

Based on this experiment, our method is capable of capturing informative latent representations across various datasets due to its superior modeling and learning capacities. The inferior performance of classic approaches is primarily due to their insufficient learning capacity. The GPLVM variants perform slightly inferior to our approach, primarily due to the assumption of kernel stationarity.

While DGPLVM addresses the modeling limitation by incorporating deep structures, its performance is hindered by more complex model optimization processes [43]. Similarly, VAE-based methods inherently suffer from posterior collapse, making them prone to generating uninformative latent representations, partly due to overfitting [19, 20].

### D.4.2 Multi-View Synthetic Data

As the number of views increases, Figures 8a and 8b present the unified latent representations generated by our method and the corresponding $R^2$ scores, respectively. These results show that with more views, the unified latent representations learned by our model progressively align more closely with the ground truth.

### D.4.3 Multi-View MNIST

We generated a four-view dataset derived from the MNIST dataset using a rotation operation, as illustrated on the left-hand side of Figure 9, alongside reconstruction results obtained with our method. The right-hand side of Figure 9 displays the KNN accuracy evaluated using the latent variables learned by various methods on MV-MNIST. These results highlight that our approach not only achieves superior performance in downstream classification tasks but also effectively reconstructs data form each view through the shared latent space.

---

[17]Comparison benchmarks include various classic dimensionality reduction approaches [9, 48, 50, 52], GPLVM-based approaches [21, 23, 24, 53], and VAE-based models [15, 54–56]. See further details in Table 6.

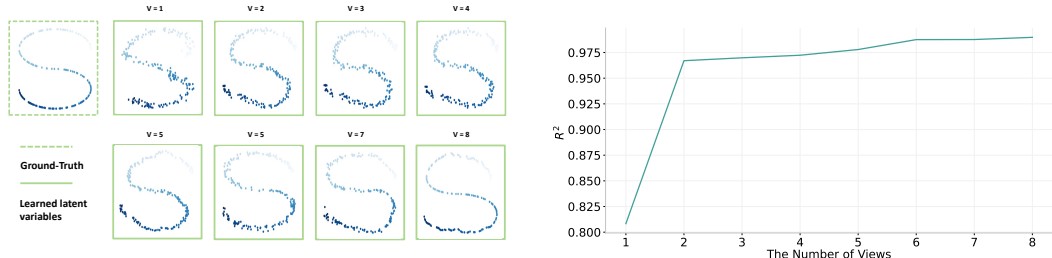

**(a)** Latent manifold learning results with multiple views.

**(b)** $R^2$ against the number of views.

**Figure 8:** Comparison of latent manifold learning and $R^2$ against the number of views.

**Figure 9:** (**Left**) MV-MNIST reconstruction task. (**Right**) Classification accuracy (%) evaluated using KNN classifier with five-fold cross-validation. Mean and standard deviation of the accuracy is computed over five experiments.

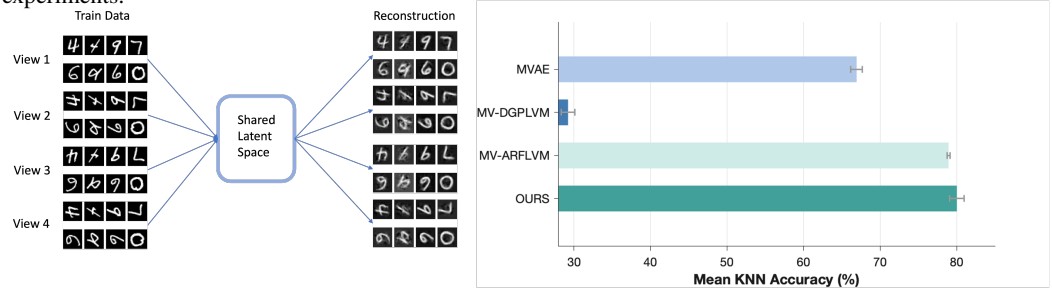

**Table 9:** Classification accuracy (%) evaluated by KNN ($k = 1$) and NN classifiers with five-fold cross-validation. **Mean and standard deviation** are computed over five experiments; the top performance is in bold.

| MODEL | MNIST-SVHN | | MOVIES | | CORA | |
|---|---|---|---|---|---|---|
| | KNN | NN | KNN | NN | KNN | NN |
| **OURS** | **93.13 ± 0.84** | **96.86 ± 0.69** | **20.64 ± 0.57** | **43.44 ± 2.35** | **46.13 ± 0.51** | **57.81 ± 0.58** |
| MV-ARFLVM | 91.16 ± 1.16 | 94.62 ± 1.56 | 19.44 ± 1.00 | 41.71 ± 0.49 | 43.61 ± 0.58 | 56.16 ± 0.74 |
| MV-DGPLVM | 85.67 ± 0.72 | 72.72 ± 1.18 | 15.15 ± 0.55 | 38.70 ± 0.21 | 25.17 ± 2.87 | 30.65 ± 1.95 |
| MVAE | 80.23 ± 1.02 | 88.21 ± 0.94 | 14.32 ± 0.68 | 40.78 ± 1.26 | 38.85 ± 0.23 | 34.29 ± 1.63 |

### D.4.4 Experiments on More Challenging Multi-View Datasets

In this section, we evaluate our method on more challenging multi-view datasets, including MNIST–SVHN, MOVIES, and CORA[18].

We evaluate the models through five-fold cross-validation with two types of classifiers: KNN and a neural network (NN). The averaged classification accuracy and its standard deviation across different datasets are reported in Table 9. The results show that our method learns high-quality latent representations. MVAE variants perform worse due to large parameterization and posterior collapse, which lead to uninformative latents and overfitting. MV-GPLVM models alleviate these issues and generally surpass MVAE, though often at higher cost. Our approach mitigates this overhead while maintaining accuracy, with gains over MV-NGPLVM stemming from the NG-SM kernel. In contrast, MV-DGPLVM is unstable and computationally prohibitive due to its heavy parameterization.

### D.4.5 Visualization of Learned Representation

We visualize the clustering structure of the learned representation to enhance interpretability. Since the latent space is set to 2D by default, we directly visualize it for BRIDGES, MNIST, and NEWS-GROUPS in multi-view setting in Figure 10, and observe that a clearer cluster structure corresponds to higher classification accuracy.

---

[18]See App. D.1.2 for detailed dataset descriptions.

**Figure 10:** Visualization of learned representations of BRIDGES, MNIST, and NEWSGROUPS datasets in a multi-view setting.

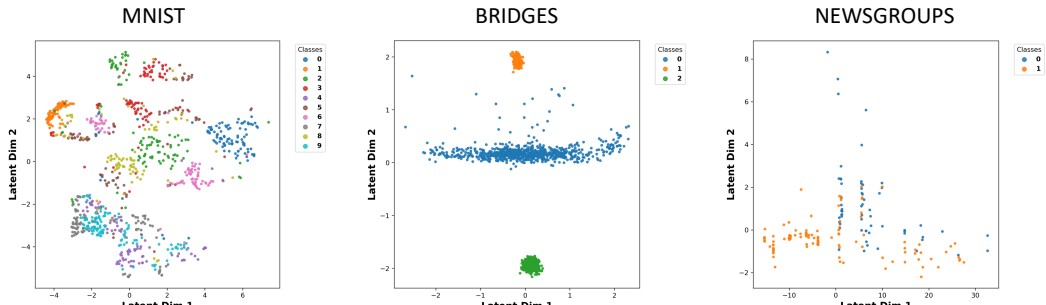

**Table 10:** Quantitative comparison of reconstruction quality on image-based datasets, measured by MSE. Each entry reports the mean and standard deviation over five runs. The best performance is highlighted in **bold**.

| MODEL | MNIST | CIFAR | MNIST–SVHN (MNIST view) | MNIST–SVHN (SVHN view) |
|---|---|---|---|---|
| **OURS** | **0.023 ± 0.0004** | **0.021 ± 0.0002** | **0.019 ± 0.0002** | **0.040 ± 0.0003** |
| MV-ARFLVM | 0.025 ± 0.0002 | 0.022 ± 0.0008 | 0.020 ± 0.0016 | 0.041 ± 0.0016 |
| MV-DGPLVM | 0.064 ± 0.0031 | 0.044 ± 0.0009 | 0.024 ± 0.0013 | 0.057 ± 0.0054 |
| MVAE | 0.077 ± 0.0040 | 0.031 ± 0.0010 | 0.031 ± 0.0021 | 0.044 ± 0.0015 |

## D.5 Data Reconstruction

In this section, we evaluate our model on data reconstruction tasks, including image reconstruction (App. D.5.1) and missing data imputation (App. D.5.2).

### D.5.1 Image Data Reconstruction

We evaluate the reconstruction capability of our model on image-based datasets. We first assess single-view reconstruction quality on MNIST and CIFAR, where each model reconstructs images from their respective latent representations. We then extend the evaluation to the multi-view setting using the MNIST–SVHN dataset: from the learned shared latent variables, we reconstruct both the MNIST and SVHN views. Reconstruction quality is measured using the MSE. Each model is trained and evaluated over five independent runs, and we report the averaged MSE along with the corresponding standard deviation in Table 10.

Our method consistently achieves the lowest MSE across all datasets, indicating superior reconstruction quality compared with existing baselines. These results demonstrate the effectiveness of our approach in capturing complex image structures while maintaining robustness across both single- and multi-view settings.

### D.5.2 Missing Data Imputation

In this section, we evaluate our model's capability for missing data imputation using the single-view datasets MNIST and BRENDAN. We randomly set various proportions of the observed data $\mathbf{Y}$ to zero (denoted as $\mathbf{Y}_{\text{obs}}$), ranging from 0% to 60%. Using the incomplete datasets $\mathbf{Y}_{\text{obs}}$, our model estimates the underlying latent variable $\mathbf{X}$, and the missing values are imputed as $\hat{\mathbf{Y}}_{\text{miss}} = \mathbb{E}[\mathbf{Y}_{\text{miss}} \mid \mathbf{X}, \mathbf{Y}_{\text{obs}}]$. Figures 12 and 13 illustrate the reconstruction tasks on the MNIST and BRENDAN datasets with varying proportions of missing values, showing superior ability of our model to restore missing pixels.[19]

## E Limitation

---

[19]In the future, we particularly interested in expressing a factorized latent space where each view is paired with an additional private space, alongside a shared space to capture unaligned variations across different views [59, 60].

**Figure 11:** Latent manifold learning results with different Q and L.

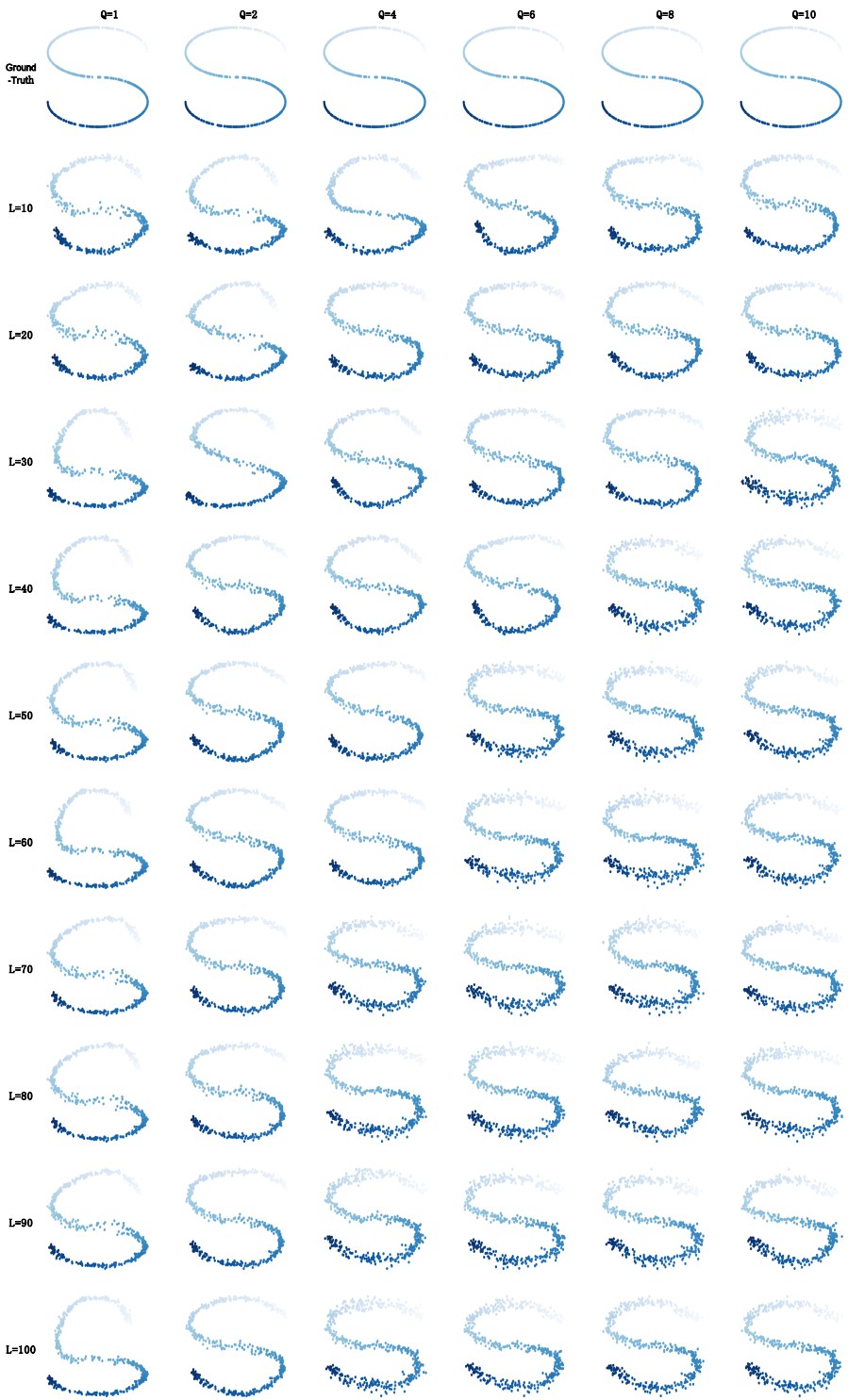

While the proposed NG-MVLVM framework effectively captures informative unified latent representations, there remains room for further enhancement. In particular, the current model does not explicitly account for cross-view dependencies, which may limit its performance in scenarios with structured dependencies across views. As a future direction, we plan to incorporate cross-view interaction terms to further enhance the model's capacity for multi-view representation learning.

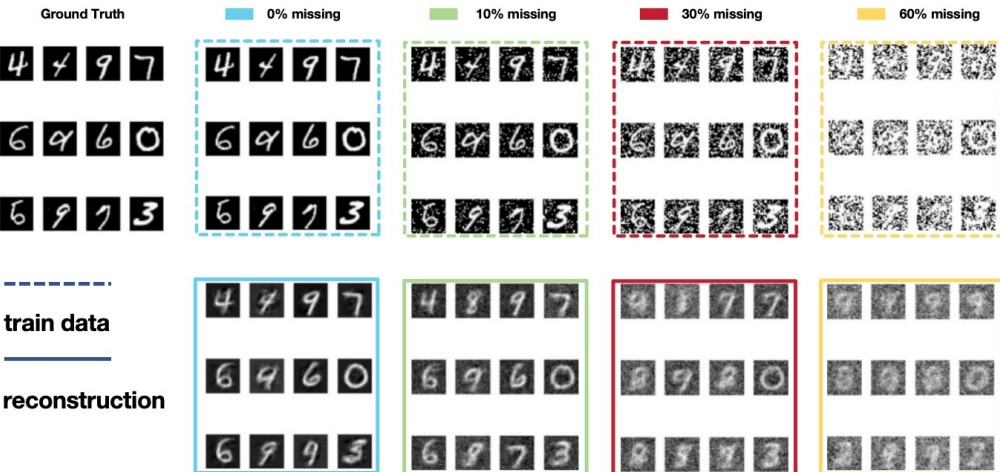

**Figure 12:** MNIST reconstruction task.

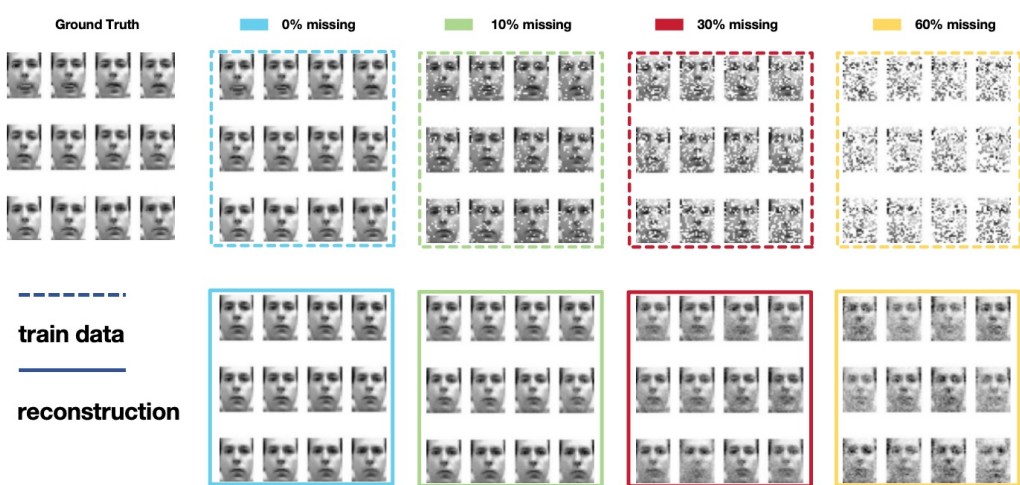

**Figure 13:** BRENDAN reconstruction task.

