# OpenReview forum: "Multi-View Oriented GPLVM: Expressiveness and Efficiency"
_NeurIPS.cc/2025/Conference — NeurIPS 2025 poster_

### Official Review · Reviewer_oW4K · 2025-07-03

**Clarity:** 3
**Significance:** 2
**Originality:** 2
**Rating:** 4
**Confidence:** 3

**Summary:**

This paper proposes a novel NG-MVLVM model aimed at enhancing kernel expressiveness and scalable inference. The key contributions include a new duality between spectral density and kernel function, and a scalable random Fourier feature approximation combined with a reparameterization trick. Empirical validation showing superior performance over state-of-the-art multi-view learning methods.

**Questions:**

1. Is the bound of Theorem 4 tight and does it play an important role?
2. Are there guidences or requirements when choosing the kernel?

**Ethical Concerns:**

["NO or VERY MINOR ethics concerns only"]

**Limitations:**

yes

**Paper Formatting Concerns:**

The full phrase of the abbreviation RFF should appear at its first use on line 48

**Quality:**

3

**Strengths And Weaknesses:**

Strengths:
1. The proposed NG-SM kernel addresses limited kernel expressiveness, lack of flexibility, and high computational complexity of previous BSM kernels.
2. The duality between spectral density and kernel function provides a principled foundation for kernel design.
3. Experiments are conducted on diverse datasets to demonstrate the effectiveness of the proposed method.

Weaknesses:
1. It's hard to identify the contributions over previous work. The authors are suggested to summarize how each component differs from them.
2. Despite the huge amount of experiments, he paper does not fully analyze the individual contributions of the NG-SM kernel and the random Fourier feature approximation through ablation studies. The terms "kernel expressiveness", "time-varying correlations" are abstract and hard to quantify.
3. The claimed efficiency is not empirically validated with runtime comparisons against other MVC methods.

---

> ### Author Rebuttal · Authors · 2025-07-30
>
> We acknowledge and appreciate the reviewer’s detailed feedback. Please see our responses addressing each comment below.
>
> >C1: It's hard to identify the contributions over previous work. The authors are suggested to summarize how each component differs from them.
>
> We have summarized the comparison with previous work in **Table 1 (page 2 of the paper)**. To highlight our contributions more clearly, we extract the key content of Table 1 below:
>
>
>  | Model          | Scalable model | Highly expressive kernel  |
>  |----------------|----------------|---------------------------|
>  | MVAE           | √              | -                         |
>  | MM-VAE         | √              | -                         |
>  | MV-GPLVM       | ×              | ×                         |
>  | MV-GPLVM-SVI   | √              | ×                         |
>  | MV-RFLVM       | ×              | ×                         |
>  | MV-DGPLVM      | ×              | ×                         |
>  | MV-ARFLVM      | √              | ×                         |
>  | **NG-MVLVM**   | **√**          | **√**                     |
>
> From this table, our contributions can be summarized as:
>
> 1. **Expressiveness:** We propose a **Next-Gen Spectral Mixture (NG-SM) kernel** that models the spectral density as a Gaussian mixture, making it a highly expressive kernel capable of capturing complex data structures. Building on this, we design **NG-MVLVM**, which learns a more informative shared latent representation across views.
>
> 2. **Efficiency:** We develop an **unbiased RFF-based approximation** for the NG-SM kernel that is differentiable with respect to kernel parameters, allowing us to use variational inference while reducing training cost. Combined with a two-step reparameterization trick, our approach further lowers computational complexity and enables efficient, scalable learning for multi-view data.
>
> >C2: Despite the huge amount of experiments, the paper does not fully analyze the individual contributions of the NG-SM kernel and the random Fourier feature approximation through ablation studies. The terms "kernel expressiveness", "time-varying correlations" are abstract and hard to quantify.
>
> We appreciate the reviewer’s comment. First, we note that the NG-SM kernel and the RFF approximation target different aspects of our model: **NG-SM** focuses on enhancing **expressiveness**, aiming to learn a more informative shared latent representation across views, while **RFF** improves **efficiency** by reducing computational complexity.
>
> Furthermore, we demonstrate  the individual contributions of each component in our model through our experiments. In Section 4.2, we compare our model against baselines using various kernels (e.g., SM kernel in MV-ARFLVM, Gibbs kernel in MV-NGPLVM), where our model with NG-SM kernel consistently achieves better results, demonstrating its stronger expressiveness and improved manifold learning.
> Figure 2 in Section 4.2 illustrates the model fitting wall-time on the MNIST dataset as the dataset size increases, highlighting the efficiency gains achieved by the RFF-based approximation.
>
> Finally, we will provide clearer definitions of key terms in the revision. For example:
> - **Kernel expressiveness:** Refers to how flexible a kernel function is in modeling various data patterns. A highly expressive kernel can approximate complex, nonlinear relationships, periodic behaviors, or multi-scale structures in the data.
>
> - **Time-varying correlations:** Means the similarity between two points is influenced not only by their distance, e.g., $\|x_1 - x_2\|$, but also by their specific locations $x_1$ and $x_2$ in the input domain. For example, two points with the same distance apart may have high correlation in one region but low correlation in another, depending on the local data characteristics.
>
>
>
> >C3: The claimed efficiency is not empirically validated with runtime comparisons against other MVC methods.
>
> Thank you for your suggestion. We note that Figure 2 in Section 4.2 have presented the runtime on the MNIST dataset, where we compare our method against other multi-view baselines (e.g., MV-DGPLVM, MVAE). The results clearly demonstrate the efficiency of our model relative to the other Gaussian process-based models. And we assume that "MVC" here refers to **multi-view learning methods**, which include the baselines we have compared against in our experiments.
>
>
> >C4: Is the bound of Theorem 4 tight and does it play an important role?
>
> **Tightness**: The bound in Theorem 4 is considered reasonably tight because it correctly captures the convergence behavior of the RFF approximation.
>
>    - Specifically, the term $L$ appears in the numerator of the exponent, $\exp \left(- \frac{3c^2 L}{\cdots}\right),$
> which implies that the probability of the approximation error exceeding $\epsilon$ decays exponentially as the number of random features $L$ increases.
>   - This aligns with the result that the expected RFF approximation error decreases at a rate of $O(1/\sqrt{L})$ [1], because the exponential tail bound guarantees that if we set $\epsilon \propto 1/\sqrt{L}$, the probability of the error exceeding $\epsilon$ decreases rapidly with L, which is consistent with the same convergence rate.
>
> **Importance**: The bound provides a crucial theoretical guarantee for the unbiased and differentiable RFF estimator, ensuring that the error between the RFF-based kernel and the true NG-SM kernel matrix diminishes as $L$ increases.
>
> [1] Rahimi, A., and Recht, B. (2008). *Random Features for Large-Scale Kernel Machines*. In *Advances in Neural Information Processing Systems (NeurIPS)*, 1177–1184.
>
>
>
> >C5: Are there guidences or requirements when choosing the kernel?
>
> Our NG-SM kernel is a universal kernel, meaning it can theoretically approximate any continuous kernel function. Unlike traditional GP models, our approach does not require selecting a predefined kernel, as the kernel is learned through inference of the model parameters. This is particularly advantageous since, as suggested by the question, the ``correct'' kernel is often unknown a priori.
>
> By learning the kernel directly from the data, our method can adaptively capture the underlying structure and dependencies without manual specification. As demonstrated in our experiments, this flexibility allows the NG-SM kernel to model diverse and complex data patterns effectively. Thus, we recommend the NG-SM kernel as a default choice in most applications where kernel flexibility is desired.
>
> >C6: The full phrase of the abbreviation RFF should appear at its first use on line 48
>
> Thank you for pointing this out. We have revised the manuscript to include the full phrase **“Random Fourier Features (RFF)”** at its first occurrence on line 48.

---

### Official Review · Reviewer_7AbS · 2025-07-03

**Clarity:** 2
**Significance:** 3
**Originality:** 3
**Rating:** 5
**Confidence:** 3

**Summary:**

In this work, the authors introduce a new multi-view Gaussian process variable model that incorporates a novel Next-Gen Spectral Mixture (NG-SM) kernel. This approach aims to address the limitations of kernel expressiveness and reduce the high computational complexity associated with the previous bivariate Spectral Mixture (BSM) kernel. The authors conduct experiments using synthetic data, (image, label) pairs, and wireless communication data. Their proposed method demonstrates state-of-the-art performance in terms of classification accuracy, training time, and data reconstruction.

**Questions:**

- Can the authors utilize neural network-based classifiers for evaluating classification accuracy results?
- Why did the authors decrease the image size of the original datasets? How does the approach adjust to larger images?
- How do the latent space sizes of the different models compare? Do they have similar dimensions?
- For the wireless communication data, what do the ten different classes represent?

**Ethical Concerns:**

["NO or VERY MINOR ethics concerns only"]

**Final Justification:**

After reviewing the authors' response along with the other reviews, they addressed my concerns by providing additional experiments and evaluations, stating they would update the submitted manuscript accordingly.

For the above reasons the reviewer increases its pre-rebuttal score.

**Limitations:**

Yes.

**Paper Formatting Concerns:**

The reviewer did not notice major formatting issues in the paper.

**Quality:**

3

**Strengths And Weaknesses:**

**Strengths**
- The authors introduce a novel duality between the spectral density and the kernel function in the context of the multi-view Gaussian process latent variable model. The new kernel enhances both the expressiveness and computational efficiency compared to previous work.

- The paper is well-written, accompanied by detailed supplementary material and shared code. The authors conduct experiments on a variety of datasets, reporting a range of results that include classification accuracy, training times, and data reconstruction capabilities.

- The proposed approach achieves state-of-the-art performance when compared to prior Gaussian process latent variable models.

**Weaknesses**
- While the approach advocates for learning a model using multimodal data, the chosen experiments do not include sufficiently challenging multimodal datasets. In the (image, label) experiments, the use of multi-view techniques primarily serves as dataset augmentation since the new views are just rotations of the original images. Additionally, it is unclear to the reviewer whether the model is supposed to reconstruct the class labels for the images. Examples of more challenging multimodal datasets can be found in reference [1].

- **Clarity** The reviewer believes that including diagrams illustrating the dependencies of the random variables, both overall and for each experimental setup, would enhance the understanding of your work. Although the authors provide experimental details in the supplementary materials, it remains unclear to the reviewer what Y represents for each dataset.

- For the image-based datasets, a quantitative analysis of reconstruction quality is missing. How do the different approaches compare concerning data reconstruction?

[1] Shi, Yuge, Brooks Paige, and Philip Torr. "Variational mixture-of-experts autoencoders for multi-modal deep generative models." Advances in neural information processing systems 32 (2019).

---

> ### Author Rebuttal · Authors · 2025-07-30
>
> We thank the reviewer for the valuable feedback. We have addressed the raised questions and suggestions, with most of the changes made in the experimental section. Below, we summarize the key additions to the experiments:
>
> 1. **New Datasets:**
>    Added experiments on more challenging multimodal datasets, including MNIST–SVHN, MOVIES, and CORA.
>
> 2. **New Metrics:**
>    Used neural network-based classifiers (3-layer MLP) for evaluating classification accuracy.
>
> 3. **Image Reconstruction Experiments:**
>   Reported MSE-based reconstruction results and added cross-view reconstruction tasks.
>
> Below, we provide point-by-point responses to each of your comments.
>
> >C1: While the approach advocates for learning a model using multimodal data, the chosen experiments do not include sufficiently challenging multimodal datasets. In the (image, label) experiments, the use of multi-view techniques primarily serves as dataset augmentation since the new views are just rotations of the original images. Additionally, it is unclear to the reviewer whether the model is supposed to reconstruct the class labels for the images. Examples of more challenging multimodal datasets can be found in reference [1].
>
> >C4: Can the authors utilize neural network-based classifiers for evaluating classification accuracy results?
>
> We thank the reviewer for the constructive feedback. To address the concerns about multimodal challenges and evaluation, we have:
> - Evaluated on more challenging datasets (MNIST-SVHN [1], MOVIES, and CORA), where our method maintains strong performance (see table below);
> - Added neural network-based classifiers (3-layer MLP) as an additional evaluation metric, further validating our approach (see table below);
> - Included cross-view reconstruction tasks (e.g., MNIST → SVHN) to better assess representation learning. Due to image upload limitations, the visual results cannot be shown here but will be included in the revised version.
>
>
> | Metric | Dataset     | MNIST-SVHN     | MOVIES         | CORA           |
> |--------|-------------|----------------|----------------|----------------|
> | KNN    | **OURS**    | **93.13 ± 0.84** | **20.64 ± 0.57** | **46.13 ± 0.51** |
> | KNN    | MV-ARFLVM   | 91.16 ± 1.16    | 19.44 ± 1.00   | 43.61 ± 0.58   |
> | KNN    | MV-DGPLVM   | 85.67 ± 0.72    | 15.15 ± 0.55   | 25.17 ± 2.87   |
> | KNN    | MVAE        | 80.23 ± 1.02    | 14.32 ± 0.68   | 38.85 ± 0.23   |
> | NN     | **OURS**    | **96.86 ± 0.69** | **43.44 ± 2.35** | **57.81 ± 0.58** |
> | NN     | MV-ARFLVM   | 94.62 ± 1.56    | 41.71 ± 0.49   | 56.16 ± 0.74   |
> | NN     | MV-DGPLVM   | 72.72 ± 1.18    | 38.70 ± 0.21   | 30.65 ± 1.95   |
> | NN     | MVAE        | 88.21 ± 0.94    | 40.78 ± 1.26   | 34.29 ± 1.63   |
>
> **Remark**:  We also provide a quantitative analysis of reconstruction quality on MNIST-SVHN in our response to Comment C3.
>
> **The new datasets used here are:**
>
>    - *MNIST–SVHN:*  Two views: Paired MNIST (784-dim) and SVHN (3072-dim) digits (10 classes).
>
>    - *MOVIES:*  Extracted from IMDb, each movie is represented by a keyword vector (1878-dim) and an actor vector (1398-dim). The dataset contains 617 movies categorized into 17 genre classes.
>
>    - *CORA:*  Each document has two views: a bag-of-words content vector (1433-dim) and a citation structure vector (2708-dim). The dataset includes 2708 documents across 7 research topic classes.
>
> **Clarification**:
>
>    - Section 4.2 evaluates classification on learned latent variables, not label reconstruction. These additions will be incorporated into the revision.
>
> >C2: Clarity. The reviewer believes that including diagrams illustrating the dependencies of the random variables, both overall and for each experimental setup, would enhance the understanding of your work. Although the authors provide experimental details in the supplementary materials, it remains unclear to the reviewer what Y represents for each dataset.
>
> We appreciate the reviewer’s constructive suggestion. To improve clarity, we will add a **probabilistic graphical model** of NG-MVLVM in the revision to explicitly illustrate the dependencies between random variables.
>
> **Clarify the role of $Y$:**  Across all experiments, $Y$ represents the observed multi-view data (e.g., images, text, or graphs), while our model infers a unified latent representation from these inputs.
>
> >C3: For the image-based datasets, a quantitative analysis of reconstruction quality is missing. How do the different approaches compare concerning data reconstruction?
>
> We appreciate the reviewer’s suggestions. We now include a quantitative comparison of reconstruction quality on image-based datasets, using **MSE** as the reconstruction metric.
>
>
>  | Model        | MNIST            | CIFAR            | MNIST–SVHN (MNIST view) | MNIST–SVHN (SVHN view) |
>  |--------------|------------------|------------------|--------------------------|-------------------------|
>  | **OURS**     | **0.023 ± 0.0004**   | **0.021 ± 0.0002**   | **0.019 ± 0.0002**           | **0.040 ± 0.0003**          |
>  | MV-ARFLVM    | 0.025 ± 0.0002   | 0.022 ± 0.0008   | 0.020 ± 0.0016           | 0.041 ± 0.0016          |
>  | MV-DGPLVM    | 0.064 ± 0.0031   | 0.044 ± 0.0009   | 0.024 ± 0.0013           | 0.057 ± 0.0054          |
>  | MVAE         | 0.077 ± 0.0040   | 0.031 ± 0.0010   | 0.031 ± 0.0021           | 0.044 ± 0.0015          |
>
> From the table, our method consistently achieves lower MSE values compared to other baselines on all image-based datasets, demonstrating better reconstruction quality.
>
> >C5: Why did the authors decrease the image size of the original datasets? How does the approach adjust to larger images?
>
> We appreciate the reviewer's question regarding image size adjustments.
>
> To clarify, the size reduction was applied only in the single-view experiments (Appendix 4.1) when comparing with RFLVM. This was necessary because RFLVM's MCMC sampling becomes computationally prohibitive for full-size images, as noted in our paper.
>
> However, in the main experiments (Section 4.2), we used full datasets, and all benchmarks reported in Section 4.2 can handle the full-size data without reduction.
>
> >C6: How do the latent space sizes of the different models compare? Do they have similar dimensions?
>
> All models are evaluated with the same latent space dimensionality, which is set to 2 by default to ensure a fair comparison.
>
> >C7: For the wireless communication data, what do the ten different classes represent?
>
> The ten classes correspond to ten different user equipments (UEs), each following a distinct trajectory. Within each class, the samples are the channel matrices observed at different time steps as the UE moves.

---

> > ### Comment · Reviewer_7AbS · 2025-08-08
> > **Reviewer Response**
> >
> > The reviewer thanks the authors for their detailed response.
> >
> > The authors addressed all of the reviewer’s concerns by incorporating additional experiments on the MNIST-SVHN, MOVIES, and CORA datasets. They also used a neural network-based classifier for further evaluations. The proposed approach achieved state-of-the-art (SOTA) classification accuracy. Additionally, the additional reconstruction loss table indicates that the proposed method has the smallest mean squared error (MSE) for the image-based datasets.

---

> > > ### Author Response · Authors · 2025-08-09
> > >
> > > Dear Reviewer 7AbS,
> > >
> > > Thank you for your constructive feedback, which has helped improve the quality of our work. We also appreciate your recognition of our rebuttal efforts.

---

### Official Review · Reviewer_ATeZ · 2025-07-04

**Clarity:** 2
**Significance:** 2
**Originality:** 2
**Rating:** 4
**Confidence:** 4

**Summary:**

This paper proposes a novel and well-motivated multi-view representation learning method by addressing two key limitations of the existing MV-GPLVM model: the lack of kernel expressiveness and poor scalability.

**Questions:**

1)	While this paper claims that the proposed NG-SM kernel is capable of approximating any continuous kernel via Bochner duality and Gaussian mixture spectral density modeling, the theoretical guarantee is only loosely discussed.
2)	Although the method aims to learn informative shared latent variables across views, the paper does not offer enough interpretability or visualization analysis (e.g., t-SNE, clustering structure, or downstream task results).
3)	The paper mentions that the complexity is reduced to O(NL²) using RFF-based kernel approximation, but it lacks empirical runtime comparisons with other methods.

**Ethical Concerns:**

["NO or VERY MINOR ethics concerns only"]

**Limitations:**

1)	While this paper claims that the proposed NG-SM kernel is capable of approximating any continuous kernel via Bochner duality and Gaussian mixture spectral density modeling, the theoretical guarantee is only loosely discussed.
2)	Although the method aims to learn informative shared latent variables across views, the paper does not offer enough interpretability or visualization analysis (e.g., t-SNE, clustering structure, or downstream task results).
3)	The paper mentions that the complexity is reduced to O(NL²) using RFF-based kernel approximation, but it lacks empirical runtime comparisons with other methods.

**Paper Formatting Concerns:**

The language needs further polishing.

**Quality:**

3

**Strengths And Weaknesses:**

The authors propose a new Next-Gen Spectral Mixture (NG-SM) kernel, derived through a duality between the kernel and its spectral density modeled via bivariate Gaussian mixtures. To make the model scalable, they further design an efficient random Fourier feature approximation combined with a reparameterization strategy. The resulting model enables efficient variational inference and achieves strong performance across multiple heterogeneous multi-view datasets.

---

> ### Author Rebuttal · Authors · 2025-07-30
>
> We thank the reviewer for the invaluable feedback and comments. Below are our detailed responses.
>
> >C1: While this paper claims that the proposed NG-SM kernel is capable of approximating any continuous kernel via Bochner duality and Gaussian mixture spectral density modeling, the theoretical guarantee is only loosely discussed.
>
> Thank you for raising this point and we agree that a rigorous proof is necessary. Accordingly, we have included a complete proof in the revised paper and have provided a brief sketch below for your reference.
>
> **Step 1:**
>
> As established by our universal Bochner’s theorem, any continuous  kernel $k_{t}(x_1,x_2)$ can be represented as:
>    $$
>    k_{t}(x_1, x_2) = \frac{1}{4} \int \Gamma(x_1, x_2; w_1, w_2) \, p_{t}(w_1, w_2) \, dw_1 dw_2,
>    $$
>    where $p_{t}(w_1,w_2)$ is the  spectral density of the target kernel and $\Gamma$ is a bounded sum of exponential terms.
>
> **Step 2:**
>
>    It is known that Gaussian mixtures are dense in $L_1(\mathbb{R}^{2d})$ [1]. Thus, for any $\varepsilon > 0$, there exists a $Q$-component Gaussian mixture $p_{ngsm}$ such that
>    $$
>    \| p_{ngsm} - p_{t} \|_{L_1} < \varepsilon,
>    $$
>    provided $Q$ is sufficiently large.
>
> **Step 3:**
>
>    We define the approximate kernel:
>    $$
>    k_{ngsm}(x_1, x_2) = \frac{1}{4} \int \Gamma(x_1, x_2; w_1, w_2) \, p_{ngsm}(w_1, w_2) \, dw_1 dw_2.
>    $$
>    The difference between the target and approximated kernels is:
>    $$
>    k_{t}(x_1, x_2) - k_{ngsm}(x_1, x_2)
>    = \frac{1}{4} \int \Gamma(x_1, x_2; w_1, w_2) \, [p_{t}(w_1, w_2) - p_{ngsm}(w_1, w_2)] \, dw_1 dw_2.
>    $$
>    Taking the absolute value and applying the triangle inequality:
>    $$
>    |k_{t}(x_1, x_2) - k_{ngsm}(x_1, x_2)|
>    \le \frac{1}{4} \int |\Gamma(x_1, x_2; w_1, w_2)| \, |p_{t} - p_{ngsm}| \, dw_1 dw_2.
>    $$
>    Since each exponential term in $\Gamma$ has modulus 1 (i.e., $|e^{i\theta}| = 1$), and $\Gamma$ is a sum of 4 such terms, we have $|\Gamma(x_1, x_2; w_1, w_2)| \le 4$. Thus,
>
>    $$
>    |k_{t}(x_1, x_2) - k_{ngsm}(x_1, x_2)| \le 1/4 \cdot 4  \| p_{t} - p_{ngsm} \|_{L_1}
>    $$
>
>    This implies that the kernel approximation error is bounded by the $L_1$ distance between $p_t$ and $p_{ngsm}$, uniformly over all input $(x_1, x_2)$.
>
> **Conclusion**
>
> As $Q \to \infty$, $p_{ngsm} \to p_{t}$ in $L_1$, implying $k_{ngsm} \to k_{t}$ uniformly. Therefore, NG-SM can approximate any continuous kernel arbitrarily well.
>
> [1] N. Kostantinos. *Gaussian mixtures and their applications to signal processing*. In *Advanced Signal Processing Handbook: Theory and Implementation for Radar, Sonar, and Medical Imaging Real Time Systems*, 2000.
>
> >C2: Although the method aims to learn informative shared latent variables across views, the paper does not offer enough interpretability or visualization analysis (e.g., t-SNE, clustering structure, or downstream task results).
>
> We agree that visualization and interpretability analysis is important. Thus, we  already included both qualitative and quantitative evidence in the original submission: Section 4.1 (Figure 1) visualizes the learned 2D latent variables and compares them with the ground-truth S-curve; Sections 4.2 and 4.3 evaluate downstream tasks using KNN, SVM, and other classifiers on multiple datasets.
>
> Following your suggestion, we further visualize the clustering structure of the learned representation to enhance interpretability. Since the latent space is 2D, we directly visualize it for MNIST and CIFAR, and observe that the clearer cluster structure corresponds to higher classification accuracy - consistent with the results in Section 4.2.
>
> >C3: The paper mentions that the complexity is reduced to O(NL²) using RFF-based kernel approximation, but it lacks empirical runtime comparisons with other methods.
>
> We appreciate the reviewer’s suggestion. As shown in Figure 2 (Section 4.2), we have included runtime evaluations on the MNIST dataset, demonstrating the efficiency of our RFF-based kernel approximation. However, since the analysis is embedded within the overall performance discussion and Figure 2 is not explicitly referenced in the runtime-related text, this result may appear unclear. To address this, we will revise the experimental section to explicitly refer to Figure 2 and better highlight the runtime advantage in practice.

---

### Author Response · Authors · 2025-08-06

**Dear Reviewers,**

We sincerely thank you for your thoughtful and constructive feedback. We have carefully addressed each of your comments in detail. For your convenience, we summarize our response below:

---

### 1. **Improved Novelty** *(ATeZ, oW4K)*

In response to the concerns raised by *(ATeZ, oW4K)* regarding the theoretical guarantee of Theorem 4 and the role of the RFF-based kernel approximation bound, we have:

- Provided a **proof** of Theorem 4, demonstrating that our NG-SM kernel can universally approximate any continuous kernel via Bochner duality and Gaussian mixture spectral density.

- Clarified the **tightness and role** of the RFF-based kernel approximation bound, showing that the approximation error decays exponentially with the number of features, aligning with established convergence results.

---

### 2. **Improved Clarity and Presentation** *(7AbS, ATeZ, oW4K)*

To address concerns regarding the clarity of our presentation and the experimental setup, we have:

- Clarified the **contributions** of our work, highlighting its distinction from previous methods in both expressiveness and efficiency. We also added intuitive explanations for key technical terms to improve accessibility.

- Added a **probabilistic graphical model** (to appear in the revision) to better illustrate the dependencies between random variables in our model.

- Expanded and clarified **experimental details**, including the meaning of \(Y\) across datasets, the use of 2D latent spaces for fair comparison, and the rationale for image size reduction in certain baselines. We also clarified the class labels used in the wireless communication dataset.

---

### 3. **Expanded Experiments and Practical Insights** *(7AbS, ATeZ, oW4K)*

In response to comments from *(7AbS, ATeZ, oW4K)* regarding the inclusion of new datasets and evaluation metrics, we have made the following improvements:

- Introduced **new challenging multimodal datasets** (MNIST–SVHN, MOVIES, CORA) and added **cross-view reconstruction tasks** to better evaluate the quality of learned representations.

- Incorporated **neural network-based classifiers** (3-layer MLP) as additional evaluation metrics, and reported **quantitative reconstruction results** using MSE, where our method consistently achieves superior performance over baselines.

- Highlighted the **runtime comparison** results in **Section 4.2** to emphasize the **computational efficiency** of our model in practice.


---

We believe that these clarifications and revisions have significantly strengthened our paper and addressed the concerns raised by the reviewers. If you have any further questions or concerns, we are more than willing to engage in further discussions.

Sincerely,
**Authors of submission 10221**

---

### Note · Authors · 2025-08-12

**Dear AC, SAC, PC, and Reviewers,**

Thank you all for your valuable time and thoughtful feedback during the review and rebuttal process.

We would like to emphasize that we have carefully addressed all reviewer comments in our rebuttal. We refer you to our last response summary for an overview, and the initial response for point-by-point clarifications.

We are especially grateful to **Reviewer 7AbS** for the **positive and constructive feedback** and the engagement in the rebuttal. We also appreciate that the **initial positive assessments** from **Reviewers ATeZ and oW4K**, who expressed **high confidence** in their evaluation. Although they did not further engage during the rebuttal phase, we have fully addressed their original concerns.

With all feedback now incorporated, we believe the paper has been significantly strengthened and presents a solid contribution that aligns well with the goals and standards of NeurIPS. We sincerely hope to share this work with the community through acceptance at the conference.

Thank you again for your time and consideration.

Sincerely,

**Authors of submission 10221**

---

### Decision · Program_Chairs · 2025-09-17

**Decision:**

Accept (poster)

**Comment:**

This paper proposes a multi-view GPLVM with a new Next-Gen Spectral Mixture kernel and scalable RFF-based inference. It tackles the issues of limited expressiveness and high cost, showing strong results across several datasets. All the reviewers agree that the paper is technically solid. Overall, the work is solid and would make a useful contribution.